# CL-DiffPhyCon: Closed-loop Diffusion Control of Complex Physical Systems

**Long Wei**[1][*]    **Haodong Feng**[1][*]    **Yuchen Yang**[2][§]    **Ruiqi Feng**[1]    **Peiyan Hu**[3][§]
**Xiang Zheng**[4][§]    **Tao Zhang**[1]    **Dixia Fan**[1]    **Tailin Wu**[1][†]
[1]Department of Artificial Intelligence, Westlake University,
[2]School of Statistics and Data Science, Nankai University,
[3]Academy of Mathematics and Systems Science, Chinese Academy of Sciences,
[4]School of Future Technology, South China University of Technology
{weilong,fenghaodong,wutailin}@westlake.edu.cn

## Abstract

The control problems of complex physical systems have broad applications in science and engineering. Previous studies have shown that generative control methods based on diffusion models offer significant advantages for solving these problems. However, existing generative control approaches face challenges in both performance and efficiency when extended to the closed-loop setting, which is essential for effective control. In this paper, we propose an efficient Closed-Loop Diffusion method for Physical systems Control (CL-DiffPhyCon). By employing an asynchronous denoising framework for different physical time steps, CL-DiffPhyCon generates control signals conditioned on real-time feedback from the system with significantly reduced computational cost during sampling. Additionally, the control process could be further accelerated by incorporating fast sampling techniques, such as DDIM. We evaluate CL-DiffPhyCon on two tasks: 1D Burgers' equation control and 2D incompressible fluid control. The results demonstrate that CL-DiffPhyCon achieves superior control performance with significant improvements in sampling efficiency. The code can be found at https://github.com/AI4Science-WestlakeU/CL_DiffPhyCon.

## 1 Introduction

The control problem of complex physical systems is a critical research area for optimizing a sequence of control actions to achieve specific objectives. It has wide applications in science and engineering fields, including fluid control (Verma et al., 2018), plasma control (Degrave et al., 2022), and particle dynamics control (Reyes Garza et al., 2023). The challenge in controlling such systems arises from their high-dimensional and highly nonlinear characteristics. Therefore, to achieve good control performance, there is an inherent requirement of *closed-loop* control, which is particularly necessary for control tasks that involve extra challenges, such as stochastic dynamics. Specifically, each control decision should be based on the latest state feedback from the system dynamics, allowing for continuous adaptation of the control inputs in response to any changes.

Over recent decades, several methods have been developed to address this problem, including classical control methods, recent reinforcement learning approaches, and the latest generative methods. Among them, diffusion models, such as DiffPhyCon (Wei et al., 2024), have demonstrated competitive performance, often outperforming both classical control and reinforcement learning methods in complex physical systems control. The superiority of diffusion models for general decision-making problems has also been widely demonstrated recently (Ajay et al., 2022; Janner et al., 2022a).

However, these diffusion control approaches encounter significant challenges in handling the closed-loop control problems, due to their reliance on a synchronous denoising strategy. The diffusion models start from pure noise to a denoised sample for all physical time steps within the model horizon. Applying a full sampling process at each physical time step can realize the closed-loop control, but

---

[*]Equal contribution. [§]Work done as an intern at Westlake University. [†]Corresponding author.

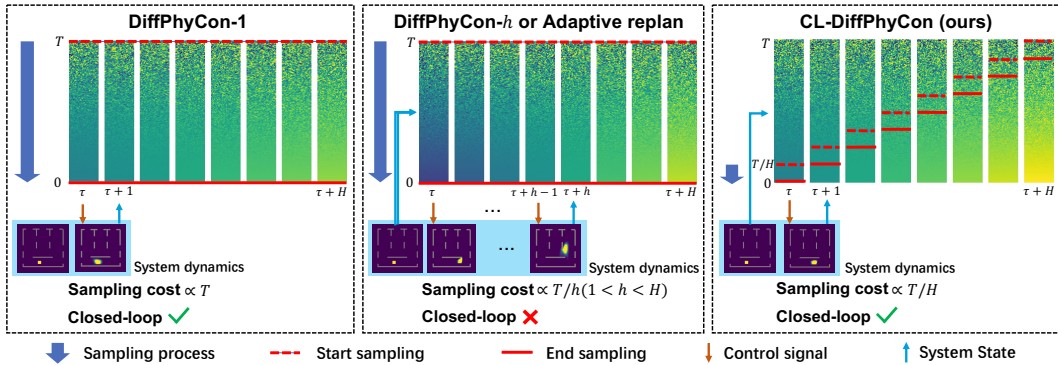

Figure 1: **Advantages of our CL-DiffPhyCon (right) over previous diffusion control methods (left and middle).** The diffusion model horizon is denoted as $H$ and the total number of diffusion steps is $T$. By employing an asynchronous denoising framework, our method could achieve closed-loop control and accelerate the sampling process significantly. The notation DiffPhyCon-$h$ means conducting a full sampling process including $T$ denoising steps every $h$ physical time steps.

incurs high sampling cost. Moreover, it may disrupt the consistency of the control signals, thereby affecting overall performance (Kaelbling & Lozano-Pérez, 2011). On the other hand, conducting a full sampling process every several physical time steps improves sampling efficiency but no longer conforms to the closed-loop requirement, leading to inferior control decisions due to outdated system states. Although an online replanning strategy has been proposed recently to determine when to replan adaptively (Zhou et al., 2024), they do not establish a fully closed-loop framework. In addition, it involves extra computation of likelihood estimation or sampling from scratch, with a dependence on thresholding hyperparameters, which may vary across different tasks and require experiments or a significant amount of tuning to determine.

In this paper, we propose a novel Closed-Loop Diffusion method for Physical systems Control, named as CL-DiffPhyCon. The key idea is to decouple the synchronous denoising within the model horizon, allowing different physical time steps to exhibit different noise levels. In this way, closed-loop generation of control sequences is naturally realized: the asynchronous diffusion model outputs control signals sequentially with increasing levels of noise along physical time steps, which enables utilization of real-time feedback state for control signal sampling in each horizon without waiting for all the following control signals in the same horizon to be denoised completely. Then, the feedback serves as the initial condition for sampling subsequent control signals, ensuring they are generated based on this reliable state. Our method can also be seen as a seamless replanning approach that leverages fresh observations with minimal sampling costs. Therefore, compared to existing diffusion-based control methods (Wei et al., 2024; Zhou et al., 2024), our approach not only realizes closed-loop control but also achieves significant sampling acceleration. These advantages of our method are illustrated in Figure 1.

In summary, we make the following contributions: (1) We propose CL-DiffPhyCon, a novel closed-loop diffusion control method for complex physical systems. The core of this method is an asynchronous diffusion model derived from a theoretical analysis of the target distribution. This model enables parallel denoising across physical time steps, allowing earlier control actions to be sampled sooner, thereby accelerating the sampling process. Additionally, the control process could be further accelerated by incorporating fast sampling techniques, such as DDIM (Song et al., 2020). (2) We evaluate CL-DiffPhyCon on the 1D Burgers' equation control and 2D incompressible fluid control tasks. The results demonstrate that CL-DiffPhyCon achieves notable control performance with significant sampling acceleration.

## 2 RELATED WORK

Classical control methods like PID (Li et al., 2006) and MPC (Schwenzer et al., 2021) are known for their high efficiency, steady performance, and good interpretability, but they face significant

challenges in both performance and efficiency when applied to control high-dimensional complex physical systems. Recently, imitation learning and reinforcement learning has shown good performance on a wide range of physical systems (Pomerleau, 1988; Zhuang et al., 2023), including drag reduction (Rabault et al., 2019; Elhawary, 2020; Feng et al., 2023; Wang et al., 2024), heat transfer (Beintema et al., 2020; Hachem et al., 2021), and fish swimming (Novati et al., 2017; Verma et al., 2018; Feng et al., 2024). Another category of supervised learning (SL) methods (Holl et al., 2020; Hwang et al., 2022) plan control signals by utilizing backpropagation through a neural surrogate model. In contrast, our approach does not depend on surrogate models; instead, it simultaneously learns the dynamics of physical systems and the corresponding control sequences. Additionally, physics-informed neural networks (PINNs) (Willard et al., 2020) have recently been utilized for control (Mowlavi & Nabi, 2023), but they necessitate explicit PDEs. In contrast, our method is data-driven and has broader applicability.

Diffusion models (Ho et al., 2020) excel at learning high-dimensional distributions and have achieved significant success in image and video generation (Dhariwal & Nichol, 2021; Ho et al., 2022), weather forecasting (Price et al., 2023), and inverse design (Wu et al., 2024), to name a few. They have also demonstrated superior capabilities on decision making tasks, such as robot control (Janner et al., 2022b; Ajay et al., 2022) and high-dimensional nonlinear systems control (Wei et al., 2024), compared with widely used reinforcement learning and imitation learning approaches. However, they struggle to balance the conflicting goals of achieving closed-loop control and maintaining efficient sampling for long trajectories. Some previous work has focused on improving the adaptability of diffusion generation (Zhou et al., 2024) on decision-making tasks, but it is not closed-loop and needs extra hyperparameters and computations for the decision of replanning. In contrast, our method is a closed-loop approach with an efficient sampling strategy without extra hyperparameter. Some recent works also assign varying noise levels to different frames within a model horizon (Wu et al., 2023; Ruhe et al., 2024). The key differences between our work and these approaches are twofold: (1) Our method focuses on the closed-loop control task of complex physical systems, whereas their methods are geared towards sequential content generation that does not involve interaction with the external world. (2) The diffusion models we need to learn are derived from the target distribution we aim to sample from (see Section 4.1 for details), while theirs are heuristic.

# 3 BACKGROUND

## 3.1 PROBLEM SETUP

Given an initial state $\mathbf{u}_0$, a system dynamics $G$, and a specified control objective $\mathcal{J}$, We consider the following complex physical systems control problem:

$$\min_{\pi} \mathbb{E}_{\mathbf{w}_{\tau+1} \sim \pi(\mathbf{w}_{\tau+1}|\mathbf{u}_\tau)}[\mathcal{J}(\mathbf{u}_0, \mathbf{w}_1, \mathbf{u}_1, \ldots, \mathbf{w}_N, \mathbf{u}_N)] \quad \text{s.t.} \quad \mathbf{u}_{\tau+1} = G(\mathbf{u}_\tau, \mathbf{w}_{\tau+1}, \xi_\tau). \quad (1)$$

Here $\mathbf{u}_\tau \in \mathbb{R}^{d_\mathbf{u}}$ and $\mathbf{w}_\tau \in \mathbb{R}^{d_\mathbf{w}}$ are the **system state** and external **control signal** at physical time step $\tau$, respectively. The **system dynamics** $G$ represents the transition of states over time under external control in the system, typically determined by implicit PDEs. $G$ could be *stochastic* with nonzero random noise $\xi_\tau$, or *deterministic* with $\xi_\tau = 0$. The evolution of states can only be observed through state measurement. The **control objective** $\mathcal{J}$ is defined over a trajectory of length $N$, representing the performance of the control strategy. For example, $\mathcal{J}$ can be designed to measure the deviation from a target state $\mathbf{u}^*$, subject to cost constraints: $\mathcal{J} = \|\mathbf{u}_N - \mathbf{u}^*\|^2 + \sum_{\tau=1}^{N} \|\mathbf{w}_\tau\|^2$. In this paper, we focus on *closed-loop* control, which means that the control signal $\mathbf{w}_{\tau+1}$ in each time step is sampled from a distribution conditioned on the current state $\mathbf{u}_\tau$. Unlike open-loop control, which determines all actions in advance, closed-loop control continuously incorporates real-time feedback to adjust control decisions dynamically, making it particularly effective for complex and evolving systems where evolution of states can only be observed through measurement of the state. To simplify notation, we introduce a variable $\mathbf{z}_\tau = [\mathbf{w}_\tau, \mathbf{u}_\tau]$ to represent the concatenation of $\mathbf{w}_\tau$ and $\mathbf{u}_\tau$. The transition probabilities in the training trajectories collected offline are assumed to satisfy the *Markov property*:

$$p(\mathbf{z}_{\tau+1}|\mathbf{u}_0, \mathbf{z}_{1:\tau}) = p(\mathbf{z}_{\tau+1}|\mathbf{u}_\tau) = p(\mathbf{w}_{\tau+1}|\mathbf{u}_\tau)p(\mathbf{u}_{\tau+1}|\mathbf{u}_\tau, \mathbf{w}_{\tau+1}). \quad (2)$$

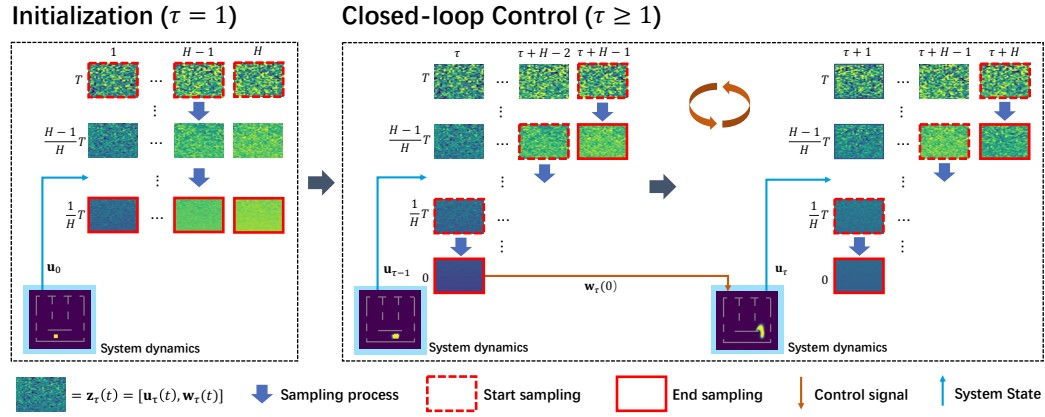

Figure 2: **CL-DiffPhyCon for closed-loop control**. First, it uses the synchronous diffusion model for initialization. Then, it uses the asynchronous diffusion model for iterative control. Sampling of each control signal is based on the latest state feedback from the system dynamics.

## 3.2 PRELIMINARY: DIFFUSION CONTROL MODELS

DiffPhyCon (Wei et al., 2024) is a recent diffusion generative method to solve the problem Eq. 1 for small $N$ in the open-loop manner. In this section, we briefly review this framework of its light version. Suppose that we have a training set $\mathcal{D}_{\text{train}}$ containing $M$ trajectories $\{\mathbf{u}_0^{(i)}, \mathbf{z}_{1:N}^{(i)}\}_{i=1}^M$ collected offline. We define the following forward diffusion SDE (Song et al., 2021) on the $\mathcal{D}_{\text{train}}$,

$$d\mathbf{z}_\tau(t) = f(t)\mathbf{z}_\tau(t)dt + g(t)d\omega_\tau(t), \quad \tau \in [1:N], \quad t \in [0,T], \tag{3}$$

where $\mathbf{z}_\tau(0) = \mathbf{z}_\tau$, $f(t)$ and $g(t)$ are scalar functions, and $\omega_\tau(t)$ is Wiener process [1]. Through Eq. 3, we augment the distribution of $\mathcal{D}_{\text{train}}$, denoted as $p_{\text{train}}(\mathbf{u}_0, \mathbf{z}_{1:N}) = p_{\text{train}}(\mathbf{u}_0)p_{\text{train}}(\mathbf{z}_{1:N}|\mathbf{u}_0)$, to the distribution of the diffusion process $p_t(\mathbf{u}_0, \mathbf{z}_{1:N}(t)) = p_{\text{train}}(\mathbf{u}_0)p_t(\mathbf{z}_{1:N}(t)|\mathbf{u}_0)$. We have two terminal conditions, $p_0(\mathbf{z}_{1:N}(0)|\mathbf{u}_0) = p_{\text{train}}(\mathbf{z}_{1:N}(0)|\mathbf{u}_0)$ and $p_T(\mathbf{z}_{1:N}(T)|\mathbf{u}_0) \approx \mathcal{N}(\mathbf{z}_{1:N}(T)|\mathbf{0}, \sigma_T^2\mathbf{I})$. The reverse-time SDE (the denoising/sampling process) of Eq. 3 has the following formula:

$$d\mathbf{z}_{1:N}(t) = [f(t)\mathbf{z}_{1:N}(t) - g(t)^2\nabla_{\mathbf{z}_{1:N}(t)}\log p_t(\mathbf{z}_{1:N}(t)|\mathbf{u}_0)]dt + g(t)d\omega_{1:N}(t), \quad t \in [T,0]. \tag{4}$$

Once we have learned a diffusion model $\boldsymbol{\epsilon}_\phi$ which approximates the score function $\nabla \log p_t(\mathbf{z}_{1:N}(t)|\mathbf{u}_0)$, we can sample $\mathbf{z}_{1:N}(0) \sim p_{\text{train}}(\mathbf{z}_{1:N}(0)|\mathbf{u}_0)$ through Eq. 4 by sustracting predicded noise from $\mathbf{z}_{1:N}(t)$ gradually from $t = T$ (where $\mathbf{z}_{1:N}(T) \sim \mathcal{N}(\mathbf{0},\mathbf{I})$) to $t = 0$. The control objective $\mathcal{J}(\mathbf{z}_{1:N}(0))$ is optimized by the following denoising process (Song et al., 2021; Chung et al., 2023):

$$
\begin{aligned}
d\mathbf{z}_{1:N}(t) =& [f(t)\mathbf{z}_{1:N}(t) - \underbrace{g(t)^2\nabla_{\mathbf{z}_{1:N}(t)}\log p_t(\mathbf{z}_{1:N}(t)|\mathbf{u}_0)}_{\text{denoise by the score function}} \\
& + \underbrace{g(t)^2\lambda \cdot \nabla_{\mathbf{z}_{1:N}(t)}\mathcal{J}(\hat{\mathbf{z}}_{1:N}(0))}_{\text{guided sampling by the control objective } \mathcal{J}}]dt + g(t)d\omega_{1:N}(t), \quad t \in [T,0].
\end{aligned}
\tag{5}
$$

Here, $\hat{\mathbf{z}}_{1:N}(0)$ is the approximate noise-free $\mathbf{z}_{1:N}(0)$ from $\mathbf{z}_{1:N}(t)$ given by Tweedie's estimate. The implementation is specialized by VP SDE (Song et al., 2021), i.e. DDPM (Ho et al., 2020), and the diffusion model is implemented by a parameterized denoising network $\boldsymbol{\epsilon}_\phi$ of horizon $H$, which equals $N$. Please refer to Appendices A, B and C.1 for more details.

## 4 METHOD

In this section, we detail our method CL-DiffPhyCon. In Section 4.1, we illustrate our idea and derive the two distributions we need to learn. To sample from them, we present the synchronous

---

[1] In $\mathbf{z}_\tau(t)$, the subscript $\tau$ denotes physical time step, and $(t)$ in the parentheses indicates SDE step.

and asynchronous diffusion models in Section 4.2 and Section 4.3, respectively. In Section 4.4, we introduce closed-loop control, which is illustrated in Figure 2. The efficiency of CL-DiffPhyCon is analyzed in Section 4.5. For ease of theoretical analysis, we adopt the SDE formulation of our method. For the DDPM implementation of its sampling process, please refer to Appendix C.

## 4.1 ASYNCHRONOUSLY DENOISING FRAMEWORK

Recall that DiffPhyCon (Wei et al., 2024) requires latent variables, e.g., system state and control signal, during the reverse diffusion process of horizon $H$ (which is typically much shorter than $N$) being denoised synchronously. As a result, to sample a control signal, a full denoising process of length $T$ over the whole horizon is performed, introducing a large amount of computation over all the latent variables within this horizon. To address this issue, we propose CL-DiffPhyCon, an *asynchronous denoising process* scheme such that the latent variables of the early physical time step are denoised in advance of the latter ones. At each physical time step, the sampled control signal is input to the system dynamics and the output state serves as the initial condition for the following denoising process. Thus, the control signal is planned based on the current system state and closed-loop control is achieved. Meanwhile, the computational cost between two successive times steps is significantly reduced due to the parallel denoising nature, compared with synchronous sampling.

Formally, we aim to model the joint distribution $p\big(\mathbf{z}_1(0), \mathbf{z}_2(0), \cdots, \mathbf{z}_N(0)|\mathbf{u}_0\big)$ in a *Markov* complex physical system, where each $\mathbf{z}_\tau(0) = [\mathbf{w}_\tau, \mathbf{u}_\tau]$ is a pair of noise-free control signal and system state, and $\mathbf{u}_0$ is the initial state. Denote $\mathbf{z}_{\tau:\tau+H-1}(t) = [\mathbf{z}_\tau(t), \mathbf{z}_{\tau+1}(t), \cdots, \mathbf{z}_{\tau+H-1}(t)]$ as the sequence of hidden variables with *synchronous* noise levels in the horizontal interval $[\tau : \tau+H-1]$ and $\tilde{\mathbf{z}}_{\tau:\tau+H-1}(t) = [\mathbf{z}_\tau(t), \mathbf{z}_{\tau+1}(t+\frac{1}{H}T), \cdots, \mathbf{z}_{\tau+H-1}(t+\frac{H-1}{H}T)]$ as its counterpart with *asynchronous* noise levels. Let's consider the augmented joint distribution

$$p\big(\mathbf{z}_{1:N}(0), \tilde{\mathbf{z}}_{1:H}(\tfrac{1}{H}T), \cdots, \tilde{\mathbf{z}}_{N+1:N+H}(\tfrac{1}{H}T)|\mathbf{u}_0\big). \tag{6}$$

Through sampling from this augmented joint distribution, we can obtain the desired control signals and states sequence $\mathbf{z}_{1:N}(0) \sim p_{\text{train}}\big(\mathbf{z}_{1:N}(0)|\mathbf{u}_0\big)$. By conditioning on previous variables sequentially, we obtain the following decomposition theorem:

**Theorem 1.** *Assume that the joint distribution $p\big(\mathbf{z}_1(0), \mathbf{z}_2(0), \cdots, \mathbf{z}_N(0)|\mathbf{u}_0\big)$ has Markov property. For any $\tau > 0$, we assume $\mathbf{z}_\tau(T)$ is independently normally distributed with density $\mathcal{N}(\mathbf{z}_\tau(T)|\mathbf{0}, \sigma_T^2\mathbf{I})$. The augmented joint distribution can be decomposed as:*

$$p\big(\mathbf{z}_{1:N}(0), \tilde{\mathbf{z}}_{1:H}(\tfrac{1}{H}T), \cdots, \tilde{\mathbf{z}}_{N+1:N+H}(\tfrac{1}{H}T)|\mathbf{u}_0\big)$$

$$= \underbrace{p\big(\tilde{\mathbf{z}}_{1:H}(\tfrac{1}{H}T)|\mathbf{u}_0\big)}_{\text{initializing distribution}} \prod_{\tau=1}^{N} \underbrace{p\big(\tilde{\mathbf{z}}_{\tau:\tau+H-1}(0)|\mathbf{u}_{\tau-1}(0), \tilde{\mathbf{z}}_{\tau:\tau+H-1}(\tfrac{1}{H}T)\big)}_{\text{transition distribution}} \mathcal{N}(\mathbf{z}_{\tau+H}(T); \mathbf{0}, \sigma_T^2\mathbf{I}). \tag{7}$$

The proofs of this theorem and the subsequent propositions are provided in the Appendix D. Note that $\tilde{\mathbf{z}}_{1:H}(\frac{1}{H}T)$ serves as a condition for $p\big(\tilde{\mathbf{z}}_{\tau:\tau+H-1}(0)|\mathbf{u}_{\tau-1}(0), \tilde{\mathbf{z}}_{\tau:\tau+H-1}(\frac{1}{H}T)\big)$ when $\tau = 1$. This theorem implies that to sample from the augmented joint distribution, we only need to specify two kinds of distributions for ancestral sampling: the *initializing distribution* $p\big(\tilde{\mathbf{z}}_{1:H}(\frac{1}{H}T)|\mathbf{u}_0\big)$, and the *transition distribution* $p\big(\tilde{\mathbf{z}}_{\tau:\tau+H-1}(0)|\mathbf{u}_{\tau-1}(0), \tilde{\mathbf{z}}_{\tau:\tau+H-1}(\frac{1}{H}T)\big)$ for $\tau > 0$.

## 4.2 SAMPLING FROM INITIALIZING DISTRIBUTION

To sample from $p\big(\tilde{\mathbf{z}}_{1:H}(\frac{1}{H}T)|\mathbf{u}_0\big)$, we train a *synchronous diffusion model* $\boldsymbol{\epsilon}_\phi$ by the following DDPM loss (Ho et al., 2020):

$$\mathcal{L}_{\text{synch}} = \mathbb{E}_{t, (\mathbf{u}_0, \mathbf{z}_{1:H}), \boldsymbol{\epsilon}}[\|\boldsymbol{\epsilon} - \boldsymbol{\epsilon}_\phi\big(\mathbf{z}_{1:H}(t), \mathbf{u}_0, t\big)\|_2^2]. \tag{8}$$

Here, $\mathbf{z}_{1:H}(t) = s(t)\mathbf{z}_{1:H} + s(t)^2\sigma(t)^2\boldsymbol{\epsilon}$ involves the same level of noise for all $t \in [1 : H]$, and the expectation is about $t \sim U(0, T)$, $(\mathbf{u}_0, \mathbf{z}_{1:H}) \sim \mathcal{D}_{\text{train}}$ and $\boldsymbol{\epsilon} \sim \mathcal{N}(\mathbf{0}, \mathbf{I})$. Intuitively, this model learns to predict the noise in its input $\mathbf{z}_{1:H}(t)$. After training, we can compute the score function by $\nabla_{\mathbf{z}_{1:H}(t)} \log p_t\big(\mathbf{z}_{1:H}(t)|\mathbf{u}_0\big) \approx -\boldsymbol{\epsilon}_\phi\big(\mathbf{z}_{1:H}(t), \mathbf{u}_0, t\big)$. Then we apply Eq. 4 to sample a sequence of $\{\mathbf{z}_{1:H}(t)\}$, where $t$ goes from $T$ to $\frac{1}{H}T$, from which we select the *diagonal* latent variables $\tilde{\mathbf{z}}_{1:H}(\frac{1}{H}T) = [\mathbf{z}_1(\frac{1}{H}T), \cdots, \mathbf{z}_H(T)]$ following the desired distribution $p\big(\tilde{\mathbf{z}}_{1:H}(\frac{1}{H}T)|\mathbf{u}_0\big)$.

### 4.3 SAMPLING FROM TRANSITION DISTRIBUTION

The transition distribution $p\big(\tilde{\mathbf{z}}_{\tau:\tau+H-1}(0)|\mathbf{u}_{\tau-1}(0),\tilde{\mathbf{z}}_{\tau:\tau+H-1}(\frac{1}{H}T)\big)$ converts $\tilde{\mathbf{z}}_{\tau:\tau+H-1}(\frac{1}{H}T)$ to $\tilde{\mathbf{z}}_{\tau:\tau+H-1}(0)$ by denoising over $t$ where $t$ goes from $T/H$ to $0$, given the state condition $\mathbf{u}_{\tau-1}$. To learn such denoising process, we first define a component-wise asynchronous forward diffusion SDE, which characterizes the diffusion dynamics of $\tilde{\mathbf{z}}_{\tau:\tau+H-1}(t)$ with an increasingly higher level of noise as $t$ increases from $0$ to $T/H$:

$$d\tilde{\mathbf{z}}_{\tau:\tau+H-1}(t) = \tilde{f}_{\tau:\tau+H-1}(t)\tilde{\mathbf{z}}_{\tau:\tau+H-1}(t)dt + \tilde{g}_{\tau:\tau+H-1}(t)d\omega_{\tau:\tau+H-1}(t), \qquad (9)$$

where $\tilde{f}_{\tau:\tau+H-1}(t) = [f(t), f(t+\frac{1}{H}T), \cdots, f(t+\frac{H-1}{H}T)]$ and $\tilde{g}_{\tau:\tau+H-1}(t) = [g(t), g(t+\frac{1}{H}T), \cdots, g(t+\frac{H-1}{H}T)]$ are vectors of scalar functions that are component-wisely applied. Similar to Eq. 4, the reverse-time SDE (the denoising/sampling process) of Eq. 9 has the following formula:

$$\begin{aligned} d\tilde{\mathbf{z}}_{\tau:\tau+H-1}(t) = [&\tilde{f}_{\tau:\tau+H-1}(t)\tilde{\mathbf{z}}_{\tau:\tau+H-1}(t) \\ &- \underbrace{\tilde{g}_{\tau:\tau+H-1}(t)^2 \nabla_{\tilde{\mathbf{z}}_{\tau:\tau+H-1}(t)} \log p_t\big(\tilde{\mathbf{z}}_{\tau:\tau+H-1}(t)|\mathbf{u}_{\tau-1}(0)\big)}_{\text{denoise by the score function}} \end{aligned} \qquad (10)$$

$$+ \tilde{g}_{\tau:\tau+H-1}(t)d\omega_{\tau:\tau+H-1}(t).$$

Thus, to sample $\tilde{\mathbf{z}}_{\tau:\tau+H-1}(0)$ from the transition distribution via Eq. 10, the key problem is to estimate the score function $\nabla_{\tilde{\mathbf{z}}_{\tau:\tau+H-1}(t)} \log p_t(\tilde{\mathbf{z}}_{\tau:\tau+H-1}(t)|\mathbf{u}_{\tau-1}(0))$. We introduce a novel *asynchronous diffusion model* $\boldsymbol{\epsilon}_\theta$ such that $\boldsymbol{\epsilon}_\theta\big(\tilde{\mathbf{z}}_{\tau:\tau+H-1}(t),\mathbf{u}_{\tau-1}(0),t\big) \approx -\nabla_{\tilde{\mathbf{z}}_{\tau:\tau+H-1}(t)} \log p_t(\tilde{\mathbf{z}}_{\tau:\tau+H-1}(t)|\mathbf{u}_{\tau-1}(0))$. This $\boldsymbol{\epsilon}_\theta$ could also be interpreted as predicting the noise in each component of its input $\tilde{\mathbf{z}}_{\tau:\tau+H-1}(t)$. To train $\boldsymbol{\epsilon}_\theta$, we need training data of the form $\tilde{\mathbf{z}}_{\tau:\tau+H-1}$. The following proposition presents a way to sample $\tilde{\mathbf{z}}_{\tau:\tau+H-1}$ from the training trajectories given the initial state condition $\mathbf{u}_{\tau-1}(0)$.

**Proposition 1.** *Assume that the joint distribution* $p\big(\mathbf{z}_1(0),\mathbf{z}_2(0),\cdots,\mathbf{z}_N(0)|\mathbf{u}_0\big)$ *has* Markov *property. For any* $t \in [0,\frac{1}{H}T]$*, we have:*

$$p_t\big(\tilde{\mathbf{z}}_{\tau:\tau+H-1}(t)|\mathbf{u}_{\tau-1}(0)\big) = \mathbb{E}_{\mathbf{z}_{\tau:\tau+H-1}(0)}\big[\prod_{i=0}^{H-1} p\big(\mathbf{z}_{\tau+i}(t+\frac{i}{H}T)|\mathbf{z}_{\tau+i}(0)\big)\big], \qquad (11)$$

*where the expectation is about* $\mathbf{z}_{\tau:\tau+H-1}(0) \sim p_{\text{train}}\big(\mathbf{z}_{\tau:\tau+H-1}(0)|\mathbf{u}_{\tau-1}(0)\big)$*, and each*

$$p\big(\mathbf{z}_{\tau+i}(t+\frac{i}{H}T)|\mathbf{z}_{\tau+i}(0)\big) = \mathcal{N}\big(\mathbf{z}_{\tau+i}(t+\frac{i}{H}T); s(t+\frac{i}{H}T)\mathbf{z}_{\tau+i}(0), s(t+\frac{i}{H}T)^2\sigma(t+\frac{i}{H}T)^2\mathbf{I}\big).$$

From this proposition, to sample $\tilde{\mathbf{z}}_{\tau:\tau+H-1}$, we first sample a subsequence $\mathbf{z}_{\tau:\tau+H-1}(0)$ of trajectory following the state $\mathbf{u}_{\tau-1}(0)$, and then independently sample each component $\mathbf{z}_{\tau+i}(t+\frac{i}{H}T)$ of $\tilde{\mathbf{z}}_{\tau:\tau+H-1}$ by adding noise to the corresponding $\mathbf{z}_{\tau+i}(0)$. Then, we train $\boldsymbol{\epsilon}_\theta$ by the following loss:

$$\mathcal{L}_{\text{asynch}} = \mathbb{E}_{\tau,t,(\mathbf{u}_{\tau-1}(0),\tilde{\mathbf{z}}_{\tau:\tau+H-1}(t)),\boldsymbol{\epsilon}}\big[\|\boldsymbol{\epsilon} - \boldsymbol{\epsilon}_\theta\big(\tilde{\mathbf{z}}_{\tau:\tau+H-1}(t),\mathbf{u}_{\tau-1},t\big)\|_2^2\big]. \qquad (12)$$

where the expectation is about $\tau \sim U(1, N-H+1)$, $t \sim U(0,\frac{1}{H}T)$, $(\mathbf{u}_{\tau-1}(0),\tilde{\mathbf{z}}_{\tau:\tau+H-1}(t)) \sim p_{\text{train}}\big(\mathbf{u}_{\tau-1}(0)\big)p_t\big(\tilde{\mathbf{z}}_{\tau:\tau+H-1}(t)|\mathbf{u}_{\tau-1}(0)\big)$, $\boldsymbol{\epsilon} \sim \mathcal{N}(\mathbf{0},\mathbf{I})$. After training, we can use Eq. 10, where $t$ goes from $\frac{1}{H}T$ to $0$, to sample $\tilde{\mathbf{z}}_{\tau:\tau+H-1}(0) \sim p\big(\tilde{\mathbf{z}}_{\tau:\tau+H-1}(0)|\mathbf{u}_{\tau-1}(0),\tilde{\mathbf{z}}_{\tau:\tau+H-1}(\frac{1}{H}T)\big)$.

To sample $\tilde{\mathbf{z}}_{\tau:\tau+H-1}(0)$ that optimizes the control objective $\mathcal{J}(\mathbf{z}_{1:N}(0))$, the denoising process can be performed with an extra term representing the guidance of the control objective $\mathcal{J}$:

$$\begin{aligned} d\tilde{\mathbf{z}}_{\tau:\tau+H-1}(t) = [&\tilde{f}_{\tau:\tau+H-1}(t)\tilde{\mathbf{z}}_{\tau:\tau+H-1}(t) \\ &-\tilde{g}_{\tau:\tau+H-1}(t)^2 \nabla_{\tilde{\mathbf{z}}_{\tau:\tau+H-1}(t)} \log p_t\big(\tilde{\mathbf{z}}_{\tau:\tau+H-1}(t)|\mathbf{u}_{\tau-1}(0)\big)]dt \\ &+\underbrace{\tilde{g}_{\tau:\tau+H-1}(t)^2 \lambda \nabla_{\tilde{\mathbf{z}}_{\tau:\tau+H-1}(t)} \mathcal{J}(\hat{\mathbf{z}}_{\tau:\tau+H-1}(0))]dt}_{\text{guided sampling by the control objective } \mathcal{J}} \end{aligned} \qquad (13)$$

$$+\tilde{g}_{\tau:\tau+H-1}(t)d\omega_{\tau:\tau+H-1}(t).$$

Here, $\hat{\mathbf{z}}_{\tau:\tau+H-1}(0) \triangleq \mathbb{E}[\mathbf{z}_{\tau:\tau+H-1}(0)|\tilde{\mathbf{z}}_{\tau:\tau+H-1}(t)]$ is the noise-free approximation of $\mathbf{z}_{\tau:\tau+H-1}(0)$ given by an asynchronous Tweedie's estimate (implemented in Eq. 41):

$$\hat{\mathbf{z}}_{\tau+i}(0) = s(t+\frac{i}{H}T)^{-1}\mathbf{z}_{\tau+i}(t+\frac{i}{H}T) + s(t+\frac{i}{H}T)\sigma(t+\frac{i}{H}T)^2 \frac{\partial \log p_t\big(\tilde{\mathbf{z}}_{\tau:\tau+H-1}(t)|\mathbf{u}_{\tau-1}(0)\big)}{\partial \mathbf{z}_{\tau+i}(t+\frac{i}{H}T)}.$$

$$(14)$$

---

**Algorithm 1** Closed-loop Control of CL-DiffPhyCon

---
1: **Require** synchronous model $\epsilon_\phi$, asynchronous model $\epsilon_\theta$, control objective $\mathcal{J}(\cdot)$, initial state $\mathbf{u}_{\text{env},0}$, episode length $N$, model horizon $H$, full denoising steps $T$, hyperparameters $\lambda$.
2: **Initialize** $\tilde{\mathbf{z}}_{1:H}(\frac{1}{H}T)$ using $\mathbf{u}_{\text{env},0}$, $\epsilon_\phi$, and $\mathbf{z}_{1:H}(T) \sim \mathcal{N}(\mathbf{0}, \sigma_T^2\mathbf{I})$ by Eq. 5
3: **for** $\tau = 1, \cdots, N$ **do**
4:     **for** $t = T/H, T/H - 1, \cdots, 1$ **do**
5:         update $\tilde{\mathbf{z}}_{\tau:\tau+H-1}(t-1)$ using $\mathbf{u}_{\text{env},\tau-1}(0)$, $\epsilon_\theta$, and $\tilde{\mathbf{z}}_{\tau:\tau+H-1}(t)$ by Eq. 13
6:     **end for**
7:     $[\mathbf{u}_\tau(0), \mathbf{w}_\tau(0)] = \mathbf{z}_\tau(0)$
8:     input $\mathbf{u}_{\text{env},\tau-1}(0)$ and $\mathbf{w}_\tau(0)$ into the system dynamics $G$, which outputs $\mathbf{u}_{\text{env},\tau}$ // closed-loop feedback
9:     sample $\mathbf{z}_{\tau+H}(T) \sim \mathcal{N}(\mathbf{0}, \sigma_T^2\mathbf{I})$ // append the end of the horizon with noise
10:     $\tilde{\mathbf{z}}_{\tau+1:\tau+H}(\frac{1}{H}T) = [\mathbf{z}_{\tau+1}(\frac{1}{H}T), \cdots, \mathbf{z}_{\tau+H}(T)]$
11: **end for**

---

## 4.4 CLOSED-LOOP CONTROL

Based on the two learned diffusion models, we now introduce the closed-loop control procedure, which together realize the distribution $\pi$ in Eq. 1, under the guidance of $\mathcal{J}$. We first use the synchronous model $\epsilon_\phi$ to produce the initial asynchronous variable $\tilde{\mathbf{z}}_{1:H}(\frac{1}{H}T)$ conditioned on the initial state $\mathbf{u}_{\text{env},0}$[2] by applying Eq. 5 in the horizon $[1, H]$. Then, the control process starts. At each physical time step $\tau \geq 1$, we sample $\tilde{\mathbf{z}}_{\tau:\tau+H-1}(0)$ using $\mathbf{u}_{\text{env},\tau-1}$ and $\tilde{\mathbf{z}}_{\tau:\tau+H-1}(\frac{1}{H}T)$ through Eq. 13. We extract the control signal $\mathbf{w}_\tau(0)$ from the sampled $\tilde{\mathbf{z}}_{\tau:\tau+H-1}(0)$, and input the pair $(\mathbf{u}_{\text{env},\tau-1}, \mathbf{w}_\tau(0))$ to the system dynamics $G$, which outputs the next state $\mathbf{u}_{\text{env},\tau}$. Then we sample a noise $\mathbf{z}_{\tau+H}(T) \sim \mathcal{N}(\mathbf{0}, \sigma_T^2\mathbf{I})$, and append it to the last $H - 1$ components of $\tilde{\mathbf{z}}_{\tau:\tau+H-1}(0)$ to compose $\tilde{\mathbf{z}}_{\tau+1:\tau+H}(\frac{1}{H}T)$. Now we take $\mathbf{u}_{\text{env},\tau-1}$, instead of the sampled $\mathbf{u}_{\tau-1}(0)$, and $\tilde{\mathbf{z}}_{\tau+1:\tau+H}(\frac{1}{H}T)$ to the next loop. The whole procedure is presented in Algorithm 1 and illustrated in Figure 2. The closed-loop property of this inference process is analyzed in Appendix E.

## 4.5 EFFICIENCY OF CL-DIFFPHYCON

It is clear that CL-DiffPhyCon is $H/h$ times faster than DiffPhyCon-$h$, the control method that conducts a full sampling process of DiffPhyCon with horizon $H$ every $h$ physical time steps, as illustrated in Figure 1. Even compared with the adaptive replaning diffusion method (Zhou et al., 2024), CL-DiffPhyCon is still more efficient because each latent variable is sampled only once and it does not involve extra computation such as likelihood estimation.

Additionally, although some fast sampling methods were proposed recently, such as DDIM Song et al. (2020), to reduce the sampling cost of diffusion models, our CL-DiffPhyCon still has an independent acceleration effect beyond them. The key insight is that CL-DiffPhyCon adopts the same number of sampling steps $T$ in each physical time step compared to DiffPhyCon (Wei et al., 2024), which brings the opportunity to incorporate fast sampling methods. Hence, by applying them to each physical time step separately in both the sampling processes of the synchronous and asynchronous models, the control efficiency of CL-DiffPhyCon could be further enhanced.

## 5 EXPERIMENT

In the experiments, we aim to answer three questions: (1) Can CL-DiffPhyCon outperform the classical and state-of-the-art baselines? (2) Can CL-DiffPhyCon achieve the desired acceleration of inference as we analyzed in Section 4.5, and obtain further acceleration by involving DDIM (Song et al., 2020)? (3) Can CL-DiffPhyCon address the challenges of noise, partial observation, partial/boundary control, and high dimensional indirect control? To answer these questions, we conduct experiments on two control tasks: 1D Burgers' equation control and 2D incompressible fluid control, both of which have important applications in science and engineering.

---

[2]In Section 4.4 and Algorithm 1, the subscript "env" in $\mathbf{u}_{\text{env},\tau}$ denotes the state feedback from the system, to distinguish $\mathbf{u}_{\text{env},\tau}$ from the sampled state $\mathbf{u}_\tau(0)$ by diffusion models.

Table 1: Comparison results on 1D Burgers' equation control. The average control objective ( $\mathcal{J}$ ) and inference time (averaged across all settings) are reported, using a single NVIDIA A100 80GB GPU with 16 CPU cores. Bold font is the best model and the runner-up is underlined.

| | noise-free ↓ | physical constraint ↓ | system noise ↓ | measure noise ↓ | FOPC ↓ | POPC ↓ | average time (s) ↓ |
|---|---|---|---|---|---|---|---|
| BC | 0.4708 | 0.4704 | 0.4138 | 0.3981 | 0.1093 | 0.0987 | 0.8356 |
| BPPO | 0.4686 | 0.4507 | 0.4088 | 0.3979 | 0.1079 | 0.0984 | 0.8231 |
| PID | 0.3250 | 0.3585 | 0.2323 | 0.2911 | - | 0.0827 | **0.7717** |
| DiffPhyCon-1 | 0.0210 | 0.0214 | 0.0222 | 0.0232 | 0.0330 | 0.0332 | 49.2347 |
| DiffPhyCon-5 | 0.0252 | 0.0257 | 0.0271 | 0.0272 | 0.0482 | 0.0484 | 10.8308 |
| DiffPhyCon-15 | 0.0361 | 0.0365 | 0.0382 | 0.0377 | 0.1128 | 0.1132 | 4.9833 |
| RDM | 0.0296 | 0.0296 | 0.0310 | 0.0336 | 0.1124 | 0.1121 | 6.9130 |
| **CL-DiffPhyCon (ours)** | **0.0096** | **0.0110** | **0.0095** | **0.0127** | **0.0291** | **0.0295** | 4.5474 |
| **CL-DiffPhyCon (DDIM, ours)** | 0.0112 | 0.0123 | 0.0114 | 0.0146 | 0.0311 | 0.0313 | 0.8257 |

## 5.1 BASELINES

The following classical, imitation learning, reinforcement learning, and diffusion control methods are selected as baselines: the classical control algorithm PID (Li et al., 2006); an imitation learning method Behaviour Cloning (BC) (Pomerleau, 1988); a recent reinforcement learning method Behavior Proximal Policy Optimization (BPPO) (Zhuang et al., 2023); two diffusion control methods, including RDM (Zhou et al., 2024), which adaptively decides when to conduct a full sampling process of control sequence, and DiffPhyCon (Wei et al., 2024), whose original version plans the control sequences of the whole trajectories in one sampling process. Since the trajectory length is much longer than the diffusion model horizon $H$ ($H = 16$ and $H = 15$ in 1D and 2D tasks, respectively), we extend DiffPhyCon to its three variants DiffPhyCon-$h$ ($h \in \{1, 5, H - 1\}$) as our baselines, based on the predefined interval $h$ of physical time steps to conduct a full sampling process. All the diffusion control baselines use the trained synchronous diffusion model of CL-DiffPhyCon for sampling, with steps $T = 900$ and $T = 600$ on the 1D and 2D tasks, respectively. PID is inapplicable to the complex 2D task (Åström & Hägglund, 2000). RDM is reproduced following the official code. However, the default values of two thresholding hyperparameters in RDM do not perform well on our tasks. Therefore, we select a pair of values that perform best (see Appendix K for details). For other baselines, we follow the implementations in DiffPhyCon Wei et al. (2024).

## 5.2 1D BURGERS' EQUATION CONTROL

**Experiment settings.** The Burgers' equation is a widely used equation to describe a variety of physical systems. We follow the works in Hwang et al. (2022); Mowlavi & Nabi (2023) and consider the 1D Burgers' equation with the Dirichlet boundary condition and external force $w(\tau, x)$ as follows:

$$\begin{cases} \frac{\partial u}{\partial \tau} = -u \cdot \frac{\partial u}{\partial x} + \nu \frac{\partial^2 u}{\partial x^2} + w(\tau, x) & \text{in } [0, \mathcal{T}] \times \Omega, \\ u(\tau, x) = 0 & \text{in } [0, \mathcal{T}] \times \partial\Omega, \\ u(0, x) = u_0(x) & \text{in } \{\tau = 0\} \times \Omega, \end{cases} \tag{15}$$

where $\nu$ is the viscosity, and $u_0(x)$ is the initial condition. Given a target state $u_d(\tau, x)$ defined in range $[0, \mathcal{T}] \times \Omega$, the control objective $\mathcal{J}$ is to minimize the error between $u(\tau, x)$ and $u_d(\tau, x)$:

$$\mathcal{J} := \int_{\mathcal{T}} \int_{\Omega} |u(\tau, x) - u_d(\tau, x)|^2 \mathrm{d}x \mathrm{d}\tau, \tag{16}$$

subject to Eq. 15. We explore four kinds of settings from real-world considerations: (1) noise-free control; (2) control under physical constraint with a limited range of allowance for control actions; (3) control under random system and measurement noise, respectively; (4) partial control (PC) where actuators are limited to control approximately $1/8$ of the full spatial domain, which is further divided to full observation (FOPC) and partial observation (POPC) with sensors in $1/8$ of the full spatial domain. Details of these settings are provided in Appendix F.2.

**Results.** In Table 1, we present the results of our proposed CL-DiffPhyCon and baselines. Note that the reported metrics in different settings are not directly comparable. It can be seen that CL-DiffPhyCon delivers the best results compared to all baselines in all settings. BC and BPPO perform

Table 2: Comparison results on 2D incompressible fluid control. The average control objective ($\mathcal{J}$) and inference time (averaged across all settings) are reported, using a single NVIDIA A6000 48GB GPU with 16 CPU cores. Bold font is the best model and the runner-up is underlined.

| | Large domain control | | Boundary control | | Average |
| | Fixed map↓ | Random map↓ | Fixed map↓ | Random map↓ | time (s)↓ |
|---|---|---|---|---|---|
| BC | 0.6722 | 0.7046 | 0.8861 | 0.8871 | 4.67 |
| BPPO | 0.6343 | 0.6524 | 0.8830 | 0.8844 | 4.65 |
| DiffPhyCon-1 | 0.5454 | 0.3754 | 0.7517 | 0.7955 | 1666.50 |
| DiffPhyCon-5 | 0.5051 | 0.5458 | 0.6703 | 0.7451 | 357.66 |
| DiffPhyCon-14 | 0.5823 | 0.5621 | 0.6498 | 0.7221 | 141.88 |
| RDM | 0.4074 | 0.4356 | 0.6553 | 0.7087 | 238.43 |
| **CL-DiffPhyCon (ours)** | **0.3371** | **0.3485** | **0.6169** | **0.7003** | 144.04 |
| **CL-DiffPhyCon (DDIM, ours)** | 0.4100 | 0.4254 | 0.6671 | 0.7109 | 26.01 |

poorly on this task as they rely heavily on the quality of training control sequences, which are far from optimal solutions (see Appendix F.1). The diffusion control baselines perform better, compared with BC and BPPO, because diffusion models conduct global optimization over each horizon through the conditional generation and excel in sampling from high dimensional space (Wei et al., 2024). Specifically, CL-DiffPhyCon decreases $\mathcal{J}$ by $54.3\%$ and $48.6\%$ compared with the best baselines (DiffPhyCon-1) in the noise-free and physical constraint settings, respectively. Furthermore, when the system or measurement is perturbed by noise, our method achieves improvements comparable to those in the noise-free case, which decreases $\mathcal{J}$ by $48.6\%$ and $57.2\%$, respectively. Moreover, despite the reduced controllable range in FOPC and POPC settings typically diminishes the effectiveness of diffusion-based control methods, CL-DiffPhyCon maintains superior performance over all baselines with at least $11.8\%$ and $11.1\%$ decreasing on $\mathcal{J}$. BC, BPPO, and PID demonstrate improved performance because only a small part of actions need to be optimized, making it easier to fit compared to high-dimensional actions, while they still show significantly lower performance compared to diffusion-based methods. Visualizations are presented in Appendix I; they show that CL-DiffPhyCon achieves lower error regarding the target state compared with baselines. In terms of inference time besides the enhanced control performance, CL-DiffPhyCon brings significant acceleration of the sampling process, about $H/h$ times faster than DiffPhyCon-$h$ and two times faster than RDM. Furthermore, by combining with DDIM of 30 sampling steps, CL-DiffPhyCon achieves an additional $5\times$ speedup, demonstrating the independent acceleration effect of CL-DiffPhyCon from existing fast sampling methods of diffusion models. As a result, the reduced time cost of CL-DiffPhyCon is comparable with those non-diffusion control baselines BC, BPPO, and PID, while the performance is much superior.

### 5.3 2D INCOMPRESSIBLE FLUID CONTROL

**Experiment settings.** Our experimental setup is based on previous studies of Holl et al. (2020); Wei et al. (2024). Given an initial cloud of smoke in a $64 \times 64$ incompressible fluid field, this task aims to minimize the volume of smoke failing to pass through the top middle exit (seven exits in total) over $N = 64$ physical time steps, by applying a sequence of 2D forces outside the outermost obstacles. This task represents a simplified scenario of real-life applications such as indoor air quality control (Nair et al., 2022). We consider two settings from real-world consideration: **large domain control**, where control signals are applied to all peripheral regions consisting of 1,792 cells outside the outermost obstacles, and **boundary control**, where control signals are restricted to only the $4 \times 8$ cells inside the four exits. It is very challenging since the control forces can only be exerted indirectly in the peripheral regions, which necessitates the model to plan ahead to prevent the smoke from entering the wrong exits or getting stuck in corners. The boundary control setting is even more challenging due to the significantly reduced range of influence of the controllable cells. During inference, for both settings, we follow Zhou et al. (2024) and add $p = 0.1$ probability of random control in the execution of control signals. Random controls cause unexpected changes in the system state, which may render previously planned control sequences no longer applicable, thus necessitating re-planning and adding additional challenge to the task. For both settings, besides a fixed map (FM) evaluation mode where test samples use the same obstacles' configuration with training trajectories, we also introduce a random map (RM) mode where the obstacles' configuration varies. For details of experimental settings and implementation, please refer to Appendix G.

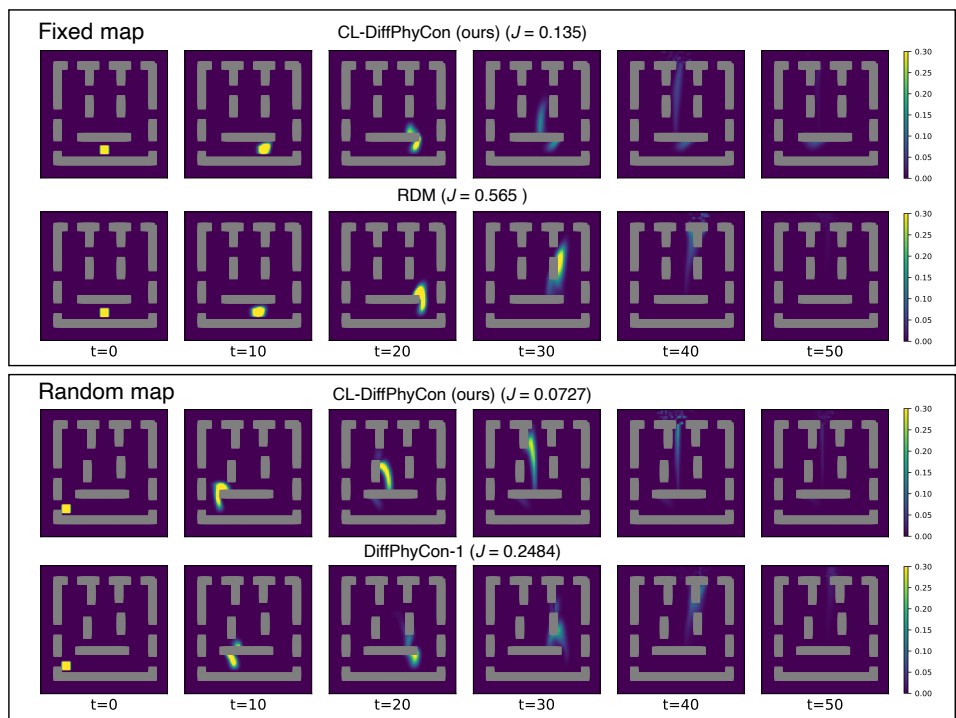

Figure 3: Visual comparisons between CL-DiffPhyCon and best baselines on 2D fluid control under fixed map (top) and random map (bottom) evaluation modes, respectively.

**Results.** Table 2 shows the control performance of CL-DiffPhyCon and baselines. The results indicate that CL-DiffPhyCon outperforms the baselines in both large domain control and boundary control settings, under both fixed map and random map evaluation modes. This validates that our asynchronous diffusion model could effectively sample appropriate subsequent control sequences by conditioning on the changed system state in a closed-loop manner, showcasing strong generalizability. The results also demonstrate the advantage of diffusion-based control methods over BC and BPPO. In Figure 3, we illustrate randomly selected test samples controlled by CL-DiffPhyCon and the best baselines in the large domain control setting. CL-DiffPhyCon exhibits a stronger capability of adapting to changed fluid and maps. More visualization results are presented in Appendix J. We also present the comparison of average inference time in Table 2. Aligning with our observation on the 1D task, CL-DiffPhyCon achieves approximately $H/h$ times speedup compared to DiffPhyCon-$h$. Besides, CL-DiffPhyCon is much more efficient than RDM by avoiding frequent replanning. By adopting DDIM, the inference is further accelerated, over $5\times$ faster than strong diffusion-based control baseline RDM, with comparable control performance. For a detailed study of the effect of the model horizon $H$, please refer to Appendix G.3.

## 6    CONCLUSION AND LIMITATION

In this paper, we propose CL-DiffPhyCon, a novel diffusion-based method for closed-loop control of complex physical systems, grounded in a theoretical analysis of the target distribution. Experiments on two physical control tasks demonstrate its superior performance and efficiency. Still, it has several limitations that offer opportunities for future work. First, CL-DiffPhyCon is currently trained offline without interacting with the system dynamics. Incorporating real-time feedback during training could enable dynamic adaptation and the discovery of new strategies. Second, while the two diffusion models in CL-DiffPhyCon are theoretically derived, a formal bound on its optimization performance under guidance sampling remains an open question, providing a promising direction for further theoretical research. Finally, the effectiveness of CL-DiffPhyCon in more domains is worth exploring. Although our method is designed for the control of complex physical systems, it holds promise for broader applications, such as robot control and drone control.

ACKNOWLEDGMENTS

We gratefully acknowledge the support of Westlake University Research Center for Industries of the Future and Westlake University Center for High-performance Computing. This research is funded by the Outstanding Postdoctoral Program Funding Project of Zhejiang Province. The content is solely the responsibility of the authors and does not necessarily represent the official views of the funding entities.

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

## A   BASIC PROPERTIES OF STOCHASTIC DIFFERENTIAL EQUATION (SDE)

In this work, we use the formulas of stochastic differential equation (SDE) to express diffusion models. Specifically, the SDEs that we use can be written to the following unified form:

$$d\mathbf{x}(t) = f(t)\mathbf{x}(t)dt + g(t)d\omega(t). \tag{17}$$

The reverse-time SDE of Eq. 17 is:

$$d\mathbf{x}(t) = [f(t)\mathbf{x}(t) - g(t)^2 \nabla_{\mathbf{x}(t)} \log p_t(\mathbf{x}(t))]dt + g(t)d\omega(t). \tag{18}$$

In this section, we summarize some basic properties of Eq. 17, which can also be found in existing SDE works Song et al. (2021); Chung et al. (2023); Karras et al. (2022).

First, such a linear SDE has the closed solution Särkkä & Solin (2019)

$$\mathbf{x}(t) = \mathbf{x}(0) \exp\left(\int_0^t f(\eta)d\eta\right) + \int_0^t \exp\left(\int_\zeta^t f(\eta)d\eta\right)g(\zeta)d\omega(\zeta), \tag{19}$$

where the second term is an Itô integral. And the mean $\mathbf{m}(t)$ and covariance $\mathbf{V}(t)$ satisfies the following ordinary differential equations:

$$\frac{d\mathbf{m}(t)}{dt} = f(t)\mathbf{m}(t),$$
$$\frac{d\mathbf{V}(t)}{dt} = 2f(t)\mathbf{V}(t) + g(t)^2\mathbf{I}. \tag{20}$$

Solving these equations, we get:

$$\mathbf{m}(t) = s(t)\mathbf{m}(0),$$
$$\mathbf{V}(t) = s(t)^2(\sigma(t)^2\mathbf{I} + \mathbf{V}(0)), \tag{21}$$

where $s(t) = \exp\left(\int_0^t f(\eta)d\eta\right)$ and $\sigma(t) = \sqrt{\int_0^t \frac{g(\eta)^2}{s(\eta)^2}d\eta}$.

**Tweedie's estimate**. From Eq. 19 and Eq. 21, we know

$$\mathbf{x}(t)|\mathbf{x}(0) \sim \mathcal{N}\left(s(t)\mathbf{x}(0), s(t)^2\sigma(t)^2\mathbf{I}\right). \tag{22}$$

According to the density function of normal distribution, we have:

$$\nabla_{\mathbf{x}(t)} \log p\left(\mathbf{x}(t)|\mathbf{x}(0)\right) = s(t)^{-2}\sigma(t)^{-2}(s(t)\mathbf{x}(0) - \mathbf{x}(t)). \tag{23}$$

Take the expectation over $\mathbf{x}(0)$ conditional on $\mathbf{x}(t)$ on both sides:

$$
\begin{aligned}
& s(t)^{-2}\sigma(t)^{-2}(s(t)\mathbb{E}[\mathbf{x}(0)|\mathbf{x}(t)] - \mathbf{x}(t)) \\
=& \mathbb{E}_{\mathbf{x}(0)}[\nabla_{\mathbf{x}(t)} \log p\left(\mathbf{x}(t)|\mathbf{x}(0)\right)|\mathbf{x}(t)] \\
=& \int \nabla_{\mathbf{x}(t)} \log p\left(\mathbf{x}(t)|\mathbf{x}(0)\right) p\left(\mathbf{x}(0)|\mathbf{x}(t)\right) d\mathbf{x}(0) \\
=& \int \nabla_{\mathbf{x}(t)} p\left(\mathbf{x}(t), \mathbf{x}(0)\right)/p\left(\mathbf{x}(t)\right) d\mathbf{x}(0) \\
=& \nabla_{\mathbf{x}(t)} p\left(\mathbf{x}(t)\right)/p\left(\mathbf{x}(t)\right) \\
=& \nabla_{\mathbf{x}(t)} \log p\left(\mathbf{x}(t)\right).
\end{aligned} \tag{24}
$$

Rearranging the equation, we get Tweedie's estimate:

$$\hat{\mathbf{x}}(0) \triangleq \mathbb{E}[\mathbf{x}(0)|\mathbf{x}(t)] = s(t)^{-1}\mathbf{x}(t) + s(t)\sigma(t)^2\nabla_{\mathbf{x}(t)} \log p\left(\mathbf{x}(t)\right). \tag{25}$$

Intuitively, Eq. 25 estimate the noise-free $\mathbf{x}_0$ given $\mathbf{x}_t$.

**Approximate score function by diffusion models**. Now we consider the following loss function of DDPM (Ho et al., 2020) to train the diffusion model $\epsilon_\phi$:

$$
\begin{aligned}
& \mathbb{E}_{t,\mathbf{x}(0),\epsilon}[\|\epsilon - \epsilon_\phi\left(s(t)\mathbf{x}(0) + s(t)^2\sigma(t)^2\epsilon, t\right)\|_2^2] \\
=& \mathbb{E}_{t,\mathbf{x}(0),\mathbf{x}(t)}[\|\frac{\mathbf{x}(t) - \mathbf{x}(0)s(t)}{s(t)^2\sigma(t)^2} - \epsilon_\phi\left(\mathbf{x}(t), t\right)\|_2^2].
\end{aligned} \tag{26}
$$

The best prediction $\epsilon_{\phi^*}(\cdot, t)$ in this loss is the following conditional expectation:

$$
\begin{aligned}
\epsilon_{\phi^*}\big(\mathbf{x}(t), t\big) &= \mathbb{E}\big[\frac{\mathbf{x}(t) - \mathbf{x}(0)s(t)}{s(t)^2\sigma(t)^2} - \epsilon_\phi\big(\mathbf{x}(t), t\big)|\mathbf{x}(t)\big] \\
&= \frac{\mathbf{x}(t) - \mathbb{E}[\mathbf{x}(0)|\mathbf{x}(t)]s(t)}{s(t)^2\sigma(t)^2}.
\end{aligned}
\tag{27}
$$

Plugging Eq. 25 in it, we get the score function:

$$
\nabla_{\mathbf{x}(t)} \log p\big(\mathbf{x}(t)\big) = -\epsilon_{\phi^*}\big(\mathbf{x}(t), t\big).
\tag{28}
$$

## B  VARIANCE PRESERVING (VP) SDE

In the practical training and inference, we use Variance Preserving (VP) SDE Song et al. (2021), which is actually the continous version of DDPM Ho et al. (2020).

VP SDE specializes Eq. 17 to

$$
\mathrm{d}\mathbf{x}(t) = -\frac{1}{2}\beta(t)\mathbf{x}(t)\mathrm{d}t + \sqrt{\beta(t)}\mathrm{d}\omega(t),
\tag{29}
$$

where $\beta(t) > 0$ for $t \in [0, T]$. And its reverse-time SDE is:

$$
\mathrm{d}\mathbf{x}(t) = [-\frac{1}{2}\beta(t)\mathbf{x}(t) - \beta(t)\nabla_{\mathbf{x}(t)} \log p_t\big(\mathbf{x}(t)\big)]\mathrm{d}t + \sqrt{\beta(t)}\mathrm{d}\omega(t),
\tag{30}
$$

In algorithm implementation, we use $K$ time steps[3], $t_i = \frac{i}{K}T$ for $i = 0, \cdots, K-1$, to discretize Eq. 29. Using $\Delta t = \frac{1}{K}T$, we have:

$$
\begin{aligned}
\mathbf{x}(t_i + \Delta t) &\approx \mathbf{x}(t_i) - \frac{1}{2}\beta(t_i)\Delta t\mathbf{x}(t_i) + \sqrt{\beta(t_i)\Delta t}\xi, \\
&\approx \sqrt{1 - \beta(t_i)\Delta t}\mathbf{x}(t_i) + \sqrt{\beta(t_i)\Delta t}\xi,
\end{aligned}
\tag{31}
$$

where $\xi \sim \mathcal{N}(\mathbf{0}, \mathbf{I})$.

Let $\beta_i = \beta(t_i)\Delta t$, $\mathbf{x}_i = \mathbf{x}(t_i)$. We get the discrete forward iterative equation, which is exact DDPM:

$$
\mathbf{x}_{i+1} = \sqrt{1 - \beta_i}\mathbf{x}_i + \sqrt{\beta_i}\xi.
\tag{32}
$$

Similarly, Eq. 30 can be discretize to

$$
\begin{aligned}
\mathbf{x}(t_i - \Delta t) &\approx \mathbf{x}(t_i) + \frac{1}{2}\beta(t_i)\Delta t\mathbf{x}(t_i) + \beta(t_i)\Delta t\nabla_{\mathbf{x}(t_i)} \log p_{t_i}\big(\mathbf{x}(t_i)\big) + \sqrt{\beta(t_i)\Delta t}\xi \\
&\approx \frac{1}{\sqrt{1 - \beta(t_i)\Delta t}}\mathbf{x}(t_i) + \beta(t_i)\Delta t\nabla_{\mathbf{x}(t_i)} \log p_{t_i}\big(\mathbf{x}(t_i)\big) + \sqrt{\beta(t_i)\Delta t}\xi.
\end{aligned}
\tag{33}
$$

So, we get the discrete backward iterative equation:

$$
\mathbf{x}_{i-1} = \frac{1}{\sqrt{1 - \beta_i}}\mathbf{x}_i + \beta_i\nabla_{\mathbf{x}_i} \log p\big(\mathbf{x}_i\big) + \sqrt{\beta_i}\xi.
\tag{34}
$$

---

[3]In this section, *time step* means diffusion step, denoted by the subscript $i$, rather than physical time step as in the main text.

Additionally, we have:

$$
\begin{aligned}
s(t_i)^2 &= \exp\Big(\int_{t_0}^{t_i+\Delta t} -\beta(\eta)\mathrm{d}\eta\Big) \\
&= \exp\Big(\sum_{j=0}^{i} -\beta(t_j)\Delta t\Big) \\
&= \prod_{j=0}^{i} \exp\big(-\beta(t_j)\Delta t\big) \\
&\approx \prod_{j=0}^{i} (1-\beta(t_j)\Delta t) \\
&= \prod_{j=0}^{i} (1-\beta_j).
\end{aligned}
\tag{35}
$$

And, we also have:

$$
\begin{aligned}
s(t_i)^2\sigma(t_i)^2 &= \exp\Big(\int_{t_0}^{t_i+\Delta t} -\beta(\eta)\mathrm{d}\eta\Big)\int_{t_0}^{t_i+\Delta t}\beta(\eta)\exp\Big(\int_{t_0}^{\eta}\beta(\zeta)\mathrm{d}\zeta\Big)\mathrm{d}\eta \\
&= \int_{t_0}^{t_i+\Delta t}\beta(\eta)\exp\Big(\int_{\eta}^{t_i+\Delta t}-\beta(\zeta)\mathrm{d}\zeta\Big)\mathrm{d}\eta \\
&\approx 1-\exp\Big(\int_{t_0}^{t_i+\Delta t}-\beta(\eta)\exp\Big(\int_{\eta}^{t_i+\Delta t}-\beta(\zeta)\mathrm{d}\zeta\Big)\mathrm{d}\eta\Big) \\
&\approx 1-\exp\Big(\int_{t_0}^{t_i+\Delta t}-\beta(\eta)\mathrm{d}\eta\Big) \\
&= 1-s(t_i)^2 \\
&\approx 1-\prod_{j=0}^{i}(1-\beta_j).
\end{aligned}
\tag{36}
$$

We define $\alpha_i = 1-\beta_i$ and $\bar{\alpha}_i = \prod_{j=0}^{i}\alpha_i$. Then, we get $s(t_i)^2 \approx \bar{\alpha}_i$, $s(t_i)^2\sigma(t_i)^2 \approx 1-\bar{\alpha}_i$, and $\mathbf{x}_i|\mathbf{x}_0 \sim \mathcal{N}\big(\sqrt{\bar{\alpha}_i}\mathbf{x}_0, (1-\bar{\alpha}_i)\mathbf{I}\big)$. Therefore, Tweedie's estimate becomes

$$
\hat{\mathbf{x}}(0) = \frac{1}{\sqrt{\bar{\alpha}_i}}(\mathbf{x}_i + (1-\bar{\alpha}_i)\nabla_{\mathbf{x}_i}\log p(\mathbf{x}_i)).
\tag{37}
$$

According to Eq. 21, the covariance of $\mathbf{x}_i$ is $(1-\bar{\alpha}_i)\mathbf{I} + \bar{\alpha}_i\mathbf{V}_0$. Assuming the clear data has covariance $\mathbf{I}$, we can start from $\mathbf{x}_K \sim \mathcal{N}(\mathbf{0}, \mathbf{I})$ during inference.

## C  DDPM IMPLEMENTATION OF CL-DIFFPHYCON

DDPM (Ho et al., 2020) is a widely adopted implementation of diffusion models. Based on the general DDPM derivations in Appendix B and SDE derivations of CL-DiffPhyCon in Section 4, here we also present the DDPM implementation of our CL-DiffPhyCon. By choosing $f(t) = -\frac{1}{2}\beta(t)$, $g(t) = \sqrt{\beta(t)}$ and following the values of $\alpha_t$'s and $\beta_t$'s specified in Appendix B, we have the following implementations.

### C.1  DDPM SAMPLING FROM THE SYNCHRONOUS DIFFUSION MODEL

By using the approximation $\nabla_{\mathbf{z}_{1:N}(t)}\log p_t\big(\mathbf{z}_{1:N}(t)|\mathbf{u}_0\big) \approx -\boldsymbol{\epsilon}_\phi\big(\mathbf{z}_{1:N}(t), \mathbf{u}_0, t\big)$ and Eq. 34, the DDPM version implementation of the sampling process in Eq. 5 (from DiffPhyCon(Wei et al.,

2024)) is

$$
\begin{aligned}
\mathbf{z}_{1:N}(t-1) =& \frac{1}{\sqrt{1-\beta_t}}\mathbf{z}_{1:N}(t) + \beta_t \nabla_{\mathbf{z}_{1:N}(t)} \log p_t\big(\mathbf{z}_{1:N}(t)|\mathbf{u}_0\big) \\
& - \beta_t \lambda \cdot \nabla_{\mathbf{z}_{1:N}(t)} \mathcal{J}(\hat{\mathbf{z}}_{1:N}(0)) + \sqrt{\beta_t}\xi \\
\approx& \frac{1}{\sqrt{1-\beta_t}}\mathbf{z}_{1:N}(t) - \beta_t \boldsymbol{\epsilon}_\phi\big(\mathbf{z}_{1:N}(t), \mathbf{u}_0, t\big) \\
& - \beta_t \lambda \cdot \nabla_{\mathbf{z}_{1:N}(t)} \mathcal{J}(\hat{\mathbf{z}}_{1:N}(0)) + \sqrt{\beta_t}\xi,
\end{aligned}
\tag{38}
$$

where $\hat{\mathbf{z}}_{1:N}(0) \triangleq \mathbb{E}[\mathbf{z}_{1:N}(0)|\mathbf{z}_{1:N}(t)]$ can be computed by the Tweedie's estimate according to Eq. 37 as follows:

$$
\hat{\mathbf{z}}_{1:N}(0) = \frac{1}{\sqrt{\bar{\alpha}_t}}\big(\mathbf{z}_{1:N}(t) - (1-\bar{\alpha}_i)\boldsymbol{\epsilon}_\phi(\mathbf{z}_{1:N}(t), \mathbf{u}_0, t)\big)
\tag{39}
$$

In implementation, to sample $\tilde{\mathbf{z}}_{1:H}(\frac{1}{H}T)$ as in Line 2 of Algorithm 1, we run Eq. 38 iteratively from $t = T$ to $t = T/H$ and use $H$ to replace $N$.

## C.2 DDPM SAMPLING FROM THE ASYNCHRONOUS DIFFUSION MODEL

Similarly, the DDPM version implementation of the sampling process in Eq. 13 is

$$
\begin{aligned}
\tilde{\mathbf{z}}_{\tau:\tau+H-1}(t-1) =& \frac{1}{\sqrt{1-\beta_t}}\tilde{\mathbf{z}}_{\tau:\tau+H-1}(t) + \beta_t \nabla_{\tilde{\mathbf{z}}_{\tau:\tau+H-1}(t)} \log p_t\big(\tilde{\mathbf{z}}_{\tau:\tau+H-1}(t)|\mathbf{u}_0\big) \\
& - \beta_t \lambda \cdot \nabla_{\mathbf{z}_{\tau:\tau+H-1}(t)} \mathcal{J}(\hat{\mathbf{z}}_{\tau:\tau+H-1}(0)) + \sqrt{\beta_t}\xi \\
\approx& \frac{1}{\sqrt{1-\beta_t}}\tilde{\mathbf{z}}_{\tau:\tau+H-1}(t) - \beta_t \boldsymbol{\epsilon}_\theta\big(\tilde{\mathbf{z}}_{\tau:\tau+H-1}(t), \mathbf{u}_0, t\big) \\
& - \beta_t \lambda \cdot \nabla_{\mathbf{z}_{\tau:\tau+H-1}(t)} \mathcal{J}(\hat{\mathbf{z}}_{\tau:\tau+H-1}(0)) + \sqrt{\beta_t}\xi,
\end{aligned}
\tag{40}
$$

where each component $\hat{\mathbf{z}}_i(0)$ of $\hat{\mathbf{z}}_{\tau:\tau+H-1}(0) \triangleq \mathbb{E}[\mathbf{z}_{\tau:\tau+H-1}(0)|\tilde{\mathbf{z}}_{\tau:\tau+H-1}(t)] = [\mathbf{z}_\tau(0), \mathbf{z}_{\tau+1}(0), \cdots, \mathbf{z}_{\tau+H-1}(0)]$ can be computed by the Tweedie's estimate according to Eq. 14 and Eq. 37 as follows:

$$
\hat{\mathbf{z}}_{\tau+i}(0) = \frac{1}{\sqrt{\bar{\alpha}_{t+i\frac{H}{T}}}}\big(\tilde{\mathbf{z}}_i(t) - (1-\bar{\alpha}_{t+i\frac{H}{T}})\boldsymbol{\epsilon}_i\big)
\tag{41}
$$

for $i = 0, \cdots, H-1$. Here $\boldsymbol{\epsilon}_i$ denotes the $i$-th component of the predicted sequence of noises $\boldsymbol{\epsilon}_\theta(\tilde{\mathbf{z}}_{\tau:\tau+H-1}(t), \mathbf{u}_{\tau-1}, t) = [\boldsymbol{\epsilon}_0, \cdots, \boldsymbol{\epsilon}_{H-1}]$ by the asynchronous diffusion model $\boldsymbol{\epsilon}_\theta$. The reason why each component of $\hat{\mathbf{z}}_i(0)$ of $\hat{\mathbf{z}}_{\tau:\tau+H-1}(0)$ is estimated separately is that different components of $\tilde{\mathbf{z}}_{\tau:\tau+H-1}(t)$ have different levels of noise, each corresponding to a different scalar coefficient $\bar{\alpha}_{t+i\frac{H}{T}}$.

# D DERIVATIONS

*Proof of Theorem 1.* We first conduct the following decomposition:

$$
\begin{aligned}
&p\big(\mathbf{z}_{1:N}(0), \tilde{\mathbf{z}}_{1:H}(\tfrac{1}{H}T), \cdots, \tilde{\mathbf{z}}_{N+1:N+H}(\tfrac{1}{H}T)|\mathbf{u}_0\big)\\
=&p\big(\tilde{\mathbf{z}}_{1:H}(\tfrac{1}{H}T), \mathbf{z}_1(0), \tilde{\mathbf{z}}_{2:H+1}(\tfrac{1}{H}T), \cdots, \mathbf{z}_N(0), \tilde{\mathbf{z}}_{N+1:N+H}(\tfrac{1}{H}T)|\mathbf{u}_0\big)\\
=&p\big(\tilde{\mathbf{z}}_{1:H}(\tfrac{1}{H}T)|\mathbf{u}_0\big)\\
&\prod_{\tau=1}^{N} p\big(\mathbf{z}_\tau(0), \tilde{\mathbf{z}}_{\tau+1:\tau+H}(\tfrac{1}{H}T)|\mathbf{u}_0, \tilde{\mathbf{z}}_{1:H}(\tfrac{1}{H}T), \mathbf{z}_1(0), \tilde{\mathbf{z}}_{2:H+1}(\tfrac{1}{H}T), \cdots, \mathbf{z}_{\tau-1}(0), \tilde{\mathbf{z}}_{\tau:\tau+H-1}(\tfrac{1}{H}T)\big)\\
=&p\big(\tilde{\mathbf{z}}_{1:H}(\tfrac{1}{H}T)|\mathbf{u}_0\big)\\
&\prod_{\tau=1}^{N} p\big(\tilde{\mathbf{z}}_{\tau:\tau+H-1}(0), \mathbf{z}_{\tau+H}(T)|\mathbf{u}_0, \tilde{\mathbf{z}}_{1:H}(\tfrac{1}{H}T), \mathbf{z}_1(0), \tilde{\mathbf{z}}_{2:H+1}(\tfrac{1}{H}T), \cdots, \mathbf{z}_{\tau-1}(0), \tilde{\mathbf{z}}_{\tau:\tau+H-1}(\tfrac{1}{H}T)\big)\\
=&p\big(\tilde{\mathbf{z}}_{1:H}(\tfrac{1}{H}T)|\mathbf{u}_0\big) \prod_{\tau=1}^{N} \mathcal{N}(\mathbf{z}_{\tau+H}(T)|\mathbf{0}, \sigma_T^2\mathbf{I})\\
&\prod_{\tau=1}^{N} p\big(\tilde{\mathbf{z}}_{\tau:\tau+H-1}(0)|\mathbf{u}_0, \tilde{\mathbf{z}}_{1:H}(\tfrac{1}{H}T), \mathbf{z}_1(0), \tilde{\mathbf{z}}_{2:H+1}(\tfrac{1}{H}T), \cdots, \mathbf{z}_{\tau-1}(0), \tilde{\mathbf{z}}_{\tau:\tau+H-1}(\tfrac{1}{H}T)\big).
\end{aligned}
\tag{42}
$$

The third equation holds since $[\mathbf{z}_\tau(0), \tilde{\mathbf{z}}_{\tau+1:\tau+H}(\tfrac{1}{H}T)]$ and $[\tilde{\mathbf{z}}_{\tau:\tau+H-1}(0), \mathbf{z}_{\tau+H}(T)]$ are two different arrangements of a same vector $[\mathbf{z}_\tau(0), \tilde{\mathbf{z}}_{\tau+1}(\tfrac{1}{H}T), \cdots, \tilde{\mathbf{z}}_{\tau+H-1}(\tfrac{H-1}{H}T), \mathbf{z}_{\tau+H}(T)]$. The last equation holds since $\mathbf{z}_\tau(T)$ is independently normally distributed with density $\mathcal{N}(\mathbf{z}_\tau(T)|\mathbf{0}, \sigma_T^2\mathbf{I})$ for any $\tau$. Then, we analyze the conditional probability $p\big(\tilde{\mathbf{z}}_{\tau:\tau+H-1}(0)|\mathbf{u}_0, \tilde{\mathbf{z}}_{1:H}(\tfrac{1}{H}T), \mathbf{z}_1(0), \tilde{\mathbf{z}}_{2:H+1}(\tfrac{1}{H}T), \cdots, \mathbf{z}_{\tau-1}(0), \tilde{\mathbf{z}}_{\tau:\tau+H-1}(\tfrac{1}{H}T)\big)$ in detail.

Due to Markov property of physical systems, when $\mathbf{u}_{\tau-1}(0)$ appears, other variables with temporal subscripts less than $\tau$ can not affect those with subscripts greater than or equal to $\tau$. Therefore, we can remove those variables with subscripts less than $\tau$, except for $\mathbf{u}_{\tau-1}(0)$. Thus, we have:

$$
\begin{aligned}
&p\big(\tilde{\mathbf{z}}_{\tau:\tau+H-1}(0)|\mathbf{u}_0, \tilde{\mathbf{z}}_{1:H}(\tfrac{1}{H}T), \mathbf{z}_1(0), \tilde{\mathbf{z}}_{2:H+1}(\tfrac{1}{H}T), \cdots, \mathbf{z}_{\tau-1}(0), \tilde{\mathbf{z}}_{\tau:\tau+H-1}(\tfrac{1}{H}T)\big)\\
=&p\big(\tilde{\mathbf{z}}_{\tau:\tau+H-1}(0)|\mathbf{u}_{\tau-1}(0), \mathbf{c}, \tilde{\mathbf{z}}_{\tau:\tau+H-1}(\tfrac{1}{H}T)\big),
\end{aligned}
\tag{43}
$$

where $\mathbf{c}$ denotes the subset of latent variables $[\mathbf{z}_\tau(T), \tilde{\mathbf{z}}_{\tau:\tau+1}(\tfrac{H-1}{H}T), \cdots, \tilde{\mathbf{z}}_{\tau:\tau+H-2}(\tfrac{2}{H}T)]$ extracted from $[\tilde{\mathbf{z}}_{1:H}(\tfrac{1}{H}T), \cdots, \mathbf{z}_{\tau-1}(0), \tilde{\mathbf{z}}_{\tau:\tau+H-1}(\tfrac{1}{H}T)]$. Further, by adding new variables $\mathbf{z}_{\tau:\tau+H-1}(0)$ to this probability, we have

$$
\begin{aligned}
&p\big(\tilde{\mathbf{z}}_{\tau:\tau+H-1}(0)|\mathbf{u}_{\tau-1}(0), \mathbf{c}, \tilde{\mathbf{z}}_{\tau:\tau+H-1}(\tfrac{1}{H}T)\big)\\
=&\int p\big(\tilde{\mathbf{z}}_{\tau:\tau+H-1}(0)|\mathbf{u}_{\tau-1}(0), \mathbf{c}, \tilde{\mathbf{z}}_{\tau:\tau+H-1}(\tfrac{1}{H}T), \mathbf{z}_{\tau:\tau+H-1}(0)\big)\\
&p\big(\mathbf{z}_{\tau:\tau+H-1}(0)|\mathbf{u}_{\tau-1}(0), \mathbf{c}, \tilde{\mathbf{z}}_{\tau:\tau+H-1}(\tfrac{1}{H}T)\big)\mathrm{d}\mathbf{z}_{\tau:\tau+H-1}(0).
\end{aligned}
\tag{44}
$$

The first distribution inside the integral in Eq. 44 can be simplified as follows:

$$
p\big(\tilde{\mathbf{z}}_{\tau:\tau+H-1}(0)|\mathbf{u}_{\tau-1}(0),\mathbf{c},\tilde{\mathbf{z}}_{\tau:\tau+H-1}(\tfrac{1}{H}T),\mathbf{z}_{\tau:\tau+H-1}(0)\big)
$$

$$
=\frac{p\big(\tilde{\mathbf{z}}_{\tau:\tau+H-1}(0),\mathbf{c},\tilde{\mathbf{z}}_{\tau:\tau+H-1}(\tfrac{1}{H}T),\mathbf{z}_{\tau:\tau+H-1}(0)|\mathbf{u}_{\tau-1}(0)\big)}{p\big(\mathbf{c},\tilde{\mathbf{z}}_{\tau:\tau+H-1}(\tfrac{1}{H}T),\mathbf{z}_{\tau:\tau+H-1}(0)|\mathbf{u}_{\tau-1}(0)\big)}
$$

$$
=\frac{p\big(\tilde{\mathbf{z}}_{\tau:\tau+H-1}(0),\tilde{\mathbf{z}}_{\tau:\tau+H-1}(\tfrac{1}{H}T),\mathbf{z}_{\tau:\tau+H-1}(0)|\mathbf{u}_{\tau-1}(0)\big)p\big(\mathbf{c}|\tilde{\mathbf{z}}_{\tau:\tau+H-1}(\tfrac{1}{H}T)\big)}{p\big(\tilde{\mathbf{z}}_{\tau:\tau+H-1}(\tfrac{1}{H}T),\mathbf{z}_{\tau:\tau+H-1}(0)|\mathbf{u}_{\tau-1}(0)\big)p\big(\mathbf{c}|\tilde{\mathbf{z}}_{\tau:\tau+H-1}(\tfrac{1}{H}T)\big)}
\qquad(45)
$$

$$
=\frac{p\big(\tilde{\mathbf{z}}_{\tau:\tau+H-1}(0),\tilde{\mathbf{z}}_{\tau:\tau+H-1}(\tfrac{1}{H}T),\mathbf{z}_{\tau:\tau+H-1}(0)|\mathbf{u}_{\tau-1}(0)\big)}{p\big(\tilde{\mathbf{z}}_{\tau:\tau+H-1}(\tfrac{1}{H}T),\mathbf{z}_{\tau:\tau+H-1}(0)|\mathbf{u}_{\tau-1}(0)\big)}
$$

$$
=p\big(\tilde{\mathbf{z}}_{\tau:\tau+H-1}(0)|\mathbf{u}_{\tau-1}(0),\tilde{\mathbf{z}}_{\tau:\tau+H-1}(\tfrac{1}{H}T),\mathbf{z}_{\tau:\tau+H-1}(0)\big).
$$

In the second equation, we use the Markov property of the diffusion process of $\mathbf{z}_\tau(t)$ over $t$, and skip a step including the following two equations

$$
p\big(\mathbf{c}|\mathbf{u}_{\tau-1}(0),\mathbf{z}_{\tau:\tau+H-1}(0),\tilde{\mathbf{z}}_{\tau:\tau+H-1}(0),\tilde{\mathbf{z}}_{\tau:\tau+H-1}(\tfrac{1}{H}T)\big)=p\big(\mathbf{c}|\tilde{\mathbf{z}}_{\tau:\tau+H-1}(\tfrac{1}{H}T)\big)
$$

$$
p\big(\mathbf{c}|\mathbf{u}_{\tau-1}(0),\mathbf{z}_{\tau:\tau+H-1}(0),\tilde{\mathbf{z}}_{\tau:\tau+H-1}(\tfrac{1}{H}T)\big)=p\big(\mathbf{c}|\tilde{\mathbf{z}}_{\tau:\tau+H-1}(\tfrac{1}{H}T)\big)
$$

in the numerator and denominator respectively of the right side of the second equation. Similarly, we show that we can also simplify the other distribution inside the integral in Eq. 44.

$$
p\big(\mathbf{z}_{\tau:\tau+H-1}(0)|\mathbf{u}_{\tau-1}(0),\mathbf{c},\tilde{\mathbf{z}}_{\tau:\tau+H-1}(\tfrac{1}{H}T)\big)
$$

$$
=\frac{p\big(\mathbf{z}_{\tau:\tau+H-1}(0),\mathbf{c},\tilde{\mathbf{z}}_{\tau:\tau+H-1}(\tfrac{1}{H}T)|\mathbf{u}_{\tau-1}(0)\big)}{p\big(\mathbf{c},\tilde{\mathbf{z}}_{\tau:\tau+H-1}(\tfrac{1}{H}T)|\mathbf{u}_{\tau-1}(0)\big)}
$$

$$
=\frac{p\big(\mathbf{z}_{\tau:\tau+H-1}(0)|\mathbf{u}_{\tau-1}(0)\big)p\big(\mathbf{c},\tilde{\mathbf{z}}_{\tau:\tau+H-1}(\tfrac{1}{H}T)|\mathbf{u}_{\tau-1}(0),\mathbf{z}_{\tau:\tau+H-1}(0)\big)}{p\big(\mathbf{c},\tilde{\mathbf{z}}_{\tau:\tau+H-1}(\tfrac{1}{H}T)|\mathbf{u}_{\tau-1}(0)\big)}
$$

$$
=\frac{p\big(\mathbf{z}_{\tau:\tau+H-1}(0)|\mathbf{u}_{\tau-1}(0)\big)p\big(\tilde{\mathbf{z}}_{\tau:\tau+H-1}(\tfrac{1}{H}T)|\mathbf{u}_{\tau-1}(0),\mathbf{z}_{\tau:\tau+H-1}(0)\big)p\big(\mathbf{c}|\tilde{\mathbf{z}}_{\tau:\tau+H-1}(\tfrac{1}{H}T)\big)}{p\big(\tilde{\mathbf{z}}_{\tau:\tau+H-1}(\tfrac{1}{H}T)|\mathbf{u}_{\tau-1}(0)\big)p\big(\mathbf{c}|\tilde{\mathbf{z}}_{\tau:\tau+H-1}(\tfrac{1}{H}T)\big)}
$$

$$
=\frac{p\big(\mathbf{z}_{\tau:\tau+H-1}(0)|\mathbf{u}_{\tau-1}(0)\big)p\big(\tilde{\mathbf{z}}_{\tau:\tau+H-1}(\tfrac{1}{H}T)|\mathbf{u}_{\tau-1}(0),\mathbf{z}_{\tau:\tau+H-1}(0)\big)}{p\big(\tilde{\mathbf{z}}_{\tau:\tau+H-1}(\tfrac{1}{H}T)|\mathbf{u}_{\tau-1}(0)\big)}
$$

$$
=p\big(\mathbf{z}_{\tau:\tau+H-1}(0)|\mathbf{u}_{\tau-1}(0),\tilde{\mathbf{z}}_{\tau:\tau+H-1}(\tfrac{1}{H}T)\big).
$$

$$
(46)
$$

Combining Eq. 43 to Eq. 46, we get the following result:

$$
p\big(\tilde{\mathbf{z}}_{\tau:\tau+H-1}(0)|\mathbf{u}_0,\tilde{\mathbf{z}}_{1:H}(\tfrac{1}{H}T),\mathbf{z}_1(0),\tilde{\mathbf{z}}_{2:H+1}(\tfrac{1}{H}T),\cdots,\mathbf{z}_{\tau-1}(0),\tilde{\mathbf{z}}_{\tau:\tau+H-1}(\tfrac{1}{H}T)\big)
$$

$$
=\int p\big(\tilde{\mathbf{z}}_{\tau:\tau+H-1}(0)|\mathbf{u}_{\tau-1}(0),\tilde{\mathbf{z}}_{\tau:\tau+H-1}(\tfrac{1}{H}T),\mathbf{z}_{\tau:\tau+H-1}(0)\big)
$$

$$
p\big(\mathbf{z}_{\tau:\tau+H-1}(0)|\mathbf{u}_{\tau-1}(0),\tilde{\mathbf{z}}_{\tau:\tau+H-1}(\tfrac{1}{H}T)\big)\mathrm{d}\mathbf{z}_{\tau:\tau+H-1}(0)
\qquad(47)
$$

$$
=\int p\big(\tilde{\mathbf{z}}_{\tau:\tau+H-1}(0),\mathbf{z}_{\tau:\tau+H-1}(0)|\mathbf{u}_{\tau-1}(0),\tilde{\mathbf{z}}_{\tau:\tau+H-1}(\tfrac{1}{H}T)\big)\mathrm{d}\mathbf{z}_{\tau:\tau+H-1}(0)
$$

$$
=p\big(\tilde{\mathbf{z}}_{\tau:\tau+H-1}(0)|\mathbf{u}_{\tau-1}(0),\tilde{\mathbf{z}}_{\tau:\tau+H-1}(\tfrac{1}{H}T)\big).
$$

Plugging this to Eq. 42, we prove the conclusion. $\qquad\square$

*Proof of Proposition 1.* Adding new variables $\mathbf{z}_{\tau:\tau+H-1}(0)$, we have:

$$p_t\big(\tilde{\mathbf{z}}_{\tau:\tau+H-1}(t)|\mathbf{u}_{\tau-1}(0)\big)$$
$$= \int p\big(\tilde{\mathbf{z}}_{\tau:\tau+H-1}(t)|\mathbf{u}_{\tau-1}(0), \mathbf{z}_{\tau:\tau+H-1}(0)\big) p\big(\mathbf{z}_{\tau:\tau+H-1}(0)|\mathbf{u}_{\tau-1}(0)\big) \mathrm{d}\mathbf{z}_{\tau:\tau+H-1}(0). \tag{48}$$

Notice that the distributions of $\{\mathbf{z}_{\tau+i}(t + \frac{i}{H}T)\}_{i=0}^{H-1}$ are conditional independent. Therefore, we can reformulate the above equation to:

$$p_t\big(\tilde{\mathbf{z}}_{\tau:\tau+H-1}(t)|\mathbf{u}_{\tau-1}(0)\big)$$
$$= \int \prod_{i=0}^{H-1} p\big(\mathbf{z}_{\tau+i}(t + \frac{i}{H}T)|\mathbf{u}_{\tau-1}(0), \mathbf{z}_{\tau:\tau+H-1}(0)\big) p\big(\mathbf{z}_{\tau:\tau+H-1}(0)|\mathbf{u}_{\tau-1}(0)\big) \mathrm{d}\mathbf{z}_{\tau:\tau+H-1}(0)$$
$$= \int \prod_{i=0}^{H-1} p\big(\mathbf{z}_{\tau+i}(t + \frac{i}{H}T)|\mathbf{z}_{\tau+i}(0)\big) p\big(\mathbf{z}_{\tau:\tau+H-1}(0)|\mathbf{u}_{\tau-1}(0)\big) \mathrm{d}\mathbf{z}_{\tau:\tau+H-1}(0)$$
$$= \mathbb{E}_{\mathbf{z}_{\tau:\tau+H-1}(0)}\big[\prod_{i=0}^{H-1} p\big(\mathbf{z}_{\tau+i}(t + \frac{i}{H}T)|\mathbf{z}_{\tau+i}(0)\big)\big]. \tag{49}$$

Here, the second equation is because $\mathbf{z}_{\tau+i}(t + \frac{i}{H}T)$ only depends on $\mathbf{z}_{\tau+i}(0)$ with the distribution Eq. 22 described in Section A. And we have proved the desired conclusion. □

## E  CLOSE-LOOP PROPERTY

The joint distribution of $\{\mathbf{w}_\tau(0), \mathbf{u}_{\mathrm{env},\tau}\}_{\tau=1}^N$ in Algorithm 1 satisfies the following proposition.

**Proposition 2.** *Using inference described in Algorithm 1, the following holds:*

$$p\big(\mathbf{w}_1(0), \mathbf{u}_{\mathrm{env},1}, \cdots, \mathbf{w}_N(0), \mathbf{u}_{\mathrm{env},N}|\mathbf{u}_{\mathrm{env},0}\big)$$
$$= \int p_{\mathrm{gd}}\big(\tilde{\mathbf{z}}_{1:H}(\frac{1}{H}T)|\mathbf{u}_{\mathrm{env},0}\big)$$
$$\prod_{\tau=1}^N p_{\mathrm{gd}}\big(\tilde{\mathbf{z}}_{\tau:\tau+H-1}(0)|\mathbf{u}_{\mathrm{env},\tau-1}, \tilde{\mathbf{z}}_{\tau:\tau+H-1}(\frac{1}{H}T)\big) p_{\mathrm{G}}\big(\mathbf{u}_{\mathrm{env},\tau}|\mathbf{u}_{\mathrm{env},\tau-1}, \mathbf{w}_\tau(0)\big) \tag{50}$$
$$\mathcal{N}(\mathbf{z}_{\tau+H}(T); \mathbf{0}, \sigma_T^2\mathbf{I})\mathrm{d}\{\mathbf{u}_0(0), \tilde{\mathbf{z}}_{1:H}(\frac{1}{H}T), \cdots, \mathbf{u}_N(0), \tilde{\mathbf{z}}_{N+1:N+H}(\frac{1}{H}T)\},$$

*where $p_{\mathrm{gd}}$ denotes the transition distribution of the guided sampling (Eq. 13), and $p_G$ denotes the transition distribution of the system dynamics $G$.*

According to Eq. 50, for every $\tau$, the control signal $\mathbf{w}_\tau(0)$ is conditional on the states $\mathbf{u}_{\mathrm{env},0:\tau-1}$ instead of the predicted $\mathbf{u}_{0:\tau-1}(0)$. Therefore, our method achieves closed-loop control.

*Proof of Proposition 2.* This is a direct conclusion from the process of Algorithm 1. Among variables in the history, only $\mathbf{u}_{\mathrm{env},\tau-H:\tau-1}, \{\mathbf{z}_{\tau-H+1}(\frac{i}{H}T)\}_{i=0}^1, \cdots, \{\mathbf{z}_{\tau-1}(\frac{i}{H}T)\}_{i=0}^{H-1}$ involve the generation of $\mathbf{w}_\tau(0)$. So, we get the conclusion. □

## F  DETAILS OF 1D BURGERS' EQUATION CONTROL

### F.1  DATASET

We follow instructions in Wei et al. (2024) to generate a 1D Burgers' equation dataset. Specifically, we use the finite difference method (FDM) to generate trajectories in a domain of space range $x \in [0, 1]$ and time range $\tau \in [0, 1]$, with random initial states and control sequences following certain distributions. The space is discretized into 128 cells and time into 10000 steps. We generated 90000 trajectories for the training set and 50 for the testing set. For each training sample, its target state $u_d(\tau, x)$ is randomly selected from other training samples, which means that almost all the

samples in the training set are unsuccessful as they could hardly achieve the target states under random controls. Thus it is challenging to generate control sequences with performance beyond the training dataset.

## F.2 Experimental Settings

Similar to (Wei et al., 2024), we design different settings of 1D Burgers' equation control in Section 5.2:

**Noise-free:** This is a scenario where all states $u(\tau, x), x \in [0, 1]$ for $\tau \in [0, 1]$ of the system can be observed. The system, control, and measurement do not have the noise.

**Physical constraint:** In real-world scenarios, the actuator often has the upper and lower limits due to the physical constraints. We limit the control with bounds $-2$ to $2$ in this setting, while unconstrained control ranges from $-5$ to $5$.

**System noise:** For complex engineering systems, the practical physical systems often have noise issues in the system (plant) and actuator (control). We consider such real-world scenarios and perturb the system by Gaussian noise with standard deviation $\sigma = 0.025$ following the work (Yildiz et al., 2021). It can also simulate the control noise setting as the noise added to the external force can be decomposed and considered as the system noise for the 1D Burgers' equation in Eq. 15. Specifically, for a deterministic system: Burgers' equation, $G_0 : \frac{\partial u}{\partial \tau} = -u \cdot \frac{\partial u}{\partial x} + \nu \frac{\partial^2 u}{\partial x^2} + w(\tau, x)$, where $u$ denotes the state variable and $w$ represents the control. To account for system noise, we augment the dynamics with additive Gaussian noise: $G_1 : \frac{\partial u}{\partial \tau} = -u \cdot \frac{\partial u}{\partial x} + \nu \frac{\partial^2 u}{\partial x^2} + w(\tau, x) + \xi_\tau, \xi_\tau \sim \mathcal{N}(0, \sigma)$. During numerical simulation, the perturbed system ($G_1$) supersedes the nominal system ($G_0$) when implementing the control input $w$. Consequently, the feedback state $u$ is obtained as the solution to the stochastic PDE ($G_1$).

**Measurement noise:** The measurement noise is also a common phenomenon in practical physical systems. We consider the measurement with Gaussian noise. The feedback state is characterized by the superposition of the nominal solution $u$ to system ($G_0$) and a Gaussian noise term $\xi_\tau \sim \mathcal{N}(0, \sigma)$ ($\sigma = 0.025$ in the experiments), thereby incorporating measurement noise into the feedback loop while preserving the dynamics of the nominal system.

**Partial observations and partial control:** In general, the sensors for observation and actuators for control are located in a small part of the spatial domain, unlike the full observation and full control (FOFC) noise-free setting. In this setting, we consider two observation cases: full observation and 16 sensors observation (1/8 of the spatial domain). Both cases are controlled by 16 evenly placed actuators. Although only partial observations are available, the reported results are calculated for an entire spatial domain, including unobservable parts. (As a common single-input single-output (SISO) controller, the PID controller is difficult to directly apply to multiple-input multiple-output (MIMO) systems. Its application to the MIMO system often requires additional decoupling and target planning modules, and the control effect of PID is greatly influenced by decouplers and planners, making it difficult to compare fairly. Therefore, we only apply the PID in 16 sensors for observation and 16 actuators for control.) The placement of sensors and actuators is illustrated in Figure 4.

**Half domain:** In this setting, we hide some parts of $\mathbf{u}$ and measure the $\mathcal{J}$ of model control as the white area with oblique lines shown in Figure 11. Specifically, $\mathbf{u}(\tau, x), x \in [\frac{1}{4}, \frac{3}{4}]$ is set to zero in the dataset during training and $\mathbf{u}_0(x), x \in [\frac{1}{4}, \frac{3}{4}]$ is also set to zero during testing. Only $\Omega = [0, \frac{1}{4}] \cup [\frac{3}{4}, 1]$ is observed, controlled and evaluated.

## F.3 Implementation

We use U-Net (Ronneberger et al., 2015) as architectures for the models $\epsilon_\phi$ and $\epsilon_\theta$. Two models are separately trained using the same training dataset. Note that in the partial observation settings, the unobserved data is invisible to the model during both training and testing as introduced in Appendix F.2. We simply pad zero in the corresponding locations of the model input and conditions, and also exclude these locations in the training loss. Therefore, the model only learns the correlation between the observed states and control sequences. We use the MSE loss to train the models. Both models have $T = 900$ diffusion steps. The DDIM (Song et al., 2020) sampling we use only has 30 diffusion steps with the hyperparameter $\eta = 1$. Hyperparameters of models and training are listed in Table 3.

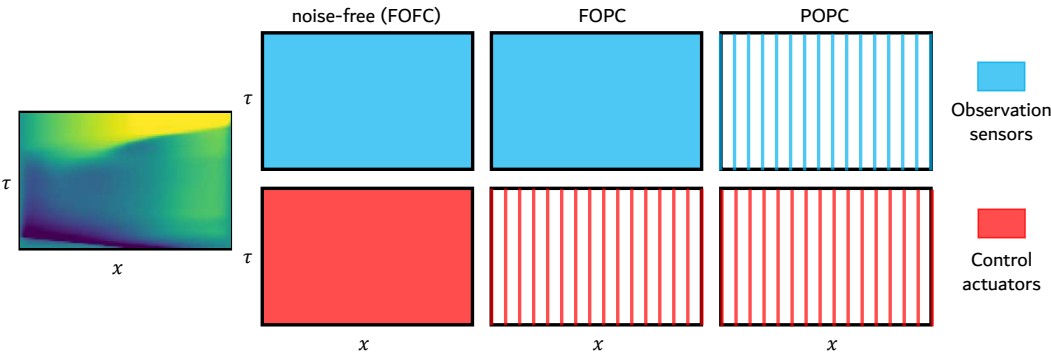

Figure 4: **Illustration of partial observation and control.** Blue represents the position observed by sensors, and red represents the position that actuators are able to control. The region filled entirely with blue and red indicates that all spatial regions can be observed and controlled, respectively. The vertical line indicates where there are sensors and actuators in that spatial position.

Table 3: **Hyperparameters of models and training for 1D Burgers' equation control.**

| Hyperparameter name | Value |
|---|---|
| U-Net $\epsilon_\phi(\mathbf{w})$ | |
| Model horizon $H$ | 16 |
| Initial dimension | 64 |
| Downsampling/Upsampling layers | 4 |
| Convolution kernel size | 3 |
| Dimension multiplier | $[1, 2, 4, 8]$ |
| Attention hidden dimension | 32 |
| Attention heads | 4 |
| U-Net $\epsilon_\theta(\mathbf{u}, \mathbf{w})$ | |
| Model horizon $H$ | 16 |
| Initial dimension | 64 |
| Downsampling/Upsampling layers | 4 |
| Convolution kernel size | 3 |
| Dimension multiplier | $[1, 2, 4, 8]$ |
| Attention hidden dimension | 32 |
| Attention heads | 4 |
| Training | |
| Training batch size | 16 |
| Optimizer | Adam |
| Learning rate | 1e-4 |
| Training steps | 190000 |
| Learning rate scheduler | cosine annealing |
| Inference | |
| Synchronously sampling steps | 900 |
| Each asynchronously sampling step | 60 |

## F.4 RESULTS OF HALF DOMAIN

In this subsection, we consider another noise-free setting that only observe, control, and evaluate on half of the spatial domain. From the results in Table 4, our conclusion is consistent with the noise-free scenario, and CL-DiffPhyCon shows significant improvements in both performance and efficiency.

Table 4: The average control objective $\mathcal{J}$ of 1D Burgers' equation control in the half domain, and inference time on a single NVIDIA A100 80GB GPU with 16 CPU cores are reported. Bold font is the best model and the runner-up is underlined.

| | $\mathcal{J}\downarrow$ | time (s)$\downarrow$ |
|---|---|---|
| BC | 0.2558 | **0.7150** |
| BPPO | 0.2033 | 0.7342 |
| PID | 0.2212 | 0.7236 |
| DiffPhyCon-1 | 0.0196 | 39.9720 |
| DiffPhyCon-5 | 0.0184 | 8.4481 |
| DiffPhyCon-15 | 0.0192 | 3.8935 |
| RDM | 0.0196 | 9.8153 |
| CL-DiffPhyCon (ours) | **0.0090** | 9.7516 |
| CL-DiffPhyCon (DDIM, ours) | 0.0104 | 0.7628 |

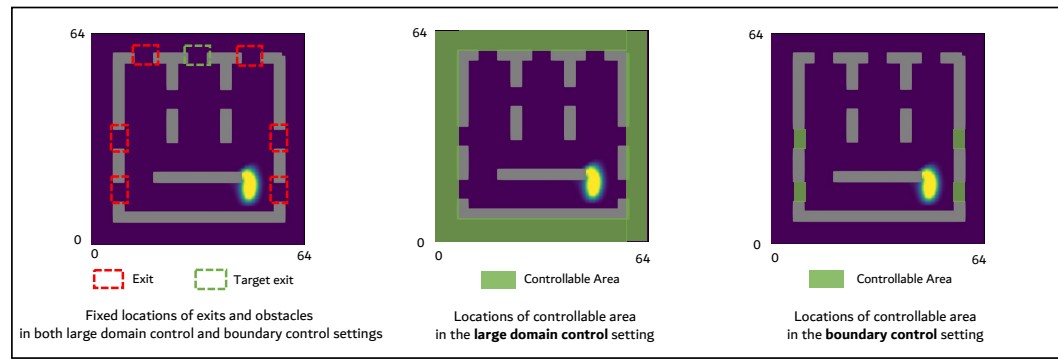

**Fixed map**
**(training data and in-distribution test)**

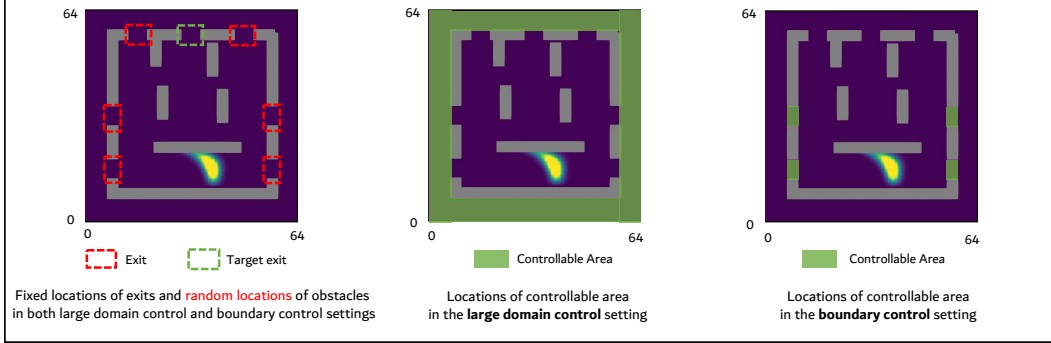

**Random map**
**(out-of-distribution test)**

Figure 5: Illustration of the 2D incompressible fluid control task settings. In the **large domain control** setting (middle), control signals are applied to the peripheral green regions surrounding the obstacles. In the **boundary control** setting (right), control signals are limited to the green cells within the four exits. Two evaluation modes are used: **fixed map** (top) for in-distribution testing and **random map** (bottom) for out-of-distribution testing.

Table 5: **Hyperparameters of 2D experiments**.

| Hyperparameter Name | Value |
|---|---|
| Number of attention heads | 4 |
| Kernel size of conv3d | (3, 3, 3) |
| Padding of conv3d | (1,1,1) |
| Stride of conv3d | (1,1,1) |
| Kernel size of downsampling | (1, 4, 4) |
| Padding of downsampling | (1, 2, 2) |
| Stride of downsampling | (0, 1, 1) |
| Kernel size of upsampling | (1, 4, 4) |
| Padding of upsampling | (1, 2, 2) |
| Stride of upsampling | (0, 1, 1) |

## G   DETAILS OF 2D INCOMPRESSIBLE FLUID CONTROL

### G.1   EXPERIMENTAL SETTING

Dynamics of 2D incompressible fluid follows the Navier-Stokes equations:

$$
\begin{cases}
\frac{\partial \mathbf{v}}{\partial \tau} + \mathbf{v} \cdot \nabla \mathbf{v} - \nu \nabla^2 \mathbf{v} + \nabla p = f, \\
\nabla \cdot \mathbf{v} = 0, \\
\mathbf{v}(0, \mathbf{x}) = \mathbf{v}_0(\mathbf{x}),
\end{cases}
\tag{51}
$$

where $f$ denotes the external force, $p$ denotes pressure, $\nu$ denotes the viscosity coefficient and $\mathbf{v}$ denotes velocity. $\mathbf{v}_0(\mathbf{x})$ is the initial condition. We follow the setup of the 2D incompressible fluid control task as described in (Wei et al., 2024), using the Phiflow solver Holl et al. (2020) to simulate fluid dynamics. The resolution of the 2D flow field is set to $64 \times 64$, and the flow field is unbounded. We consider two settings: **large domain control** and **boundary control**. In the large domain control setting, control signals are applied to all peripheral regions outside the obstacles, consisting of 1,792 cells, as highlighted in green in the middle subfigures of Figure 5. This setting is consistent with (Wei et al., 2024). In the boundary control setting, control signals are restricted to only the $4 \times 8$ cells inside the four exits, as shown in green in the right subfigures of Figure 5. This setting is newly designed in this paper. Both configurations represent indirect control, as the smoke primarily moves within the gray obstacles. The boundary control setting is more challenging due to the significantly reduced number of controllable cells. We also follow (Wei et al., 2024) to generate the training dataset. For both settings, the obstacle locations (gray cells) are shown in the top row of Figure 5 and are consistent across all training trajectories. Each trajectory contains $N = 64$ physical time steps and includes features such as horizontal and vertical velocities, smoke density, and horizontal and vertical control forces. We generated 40,000 training trajectories for the large domain control setting, and 30,000 for the boundary control setting.

During inference, following RDM (Zhou et al., 2024), we add a small level of control noise to make the task more challenging. Specifically, for a trajectory of length $N = 64$, the control signal in each physical time step is executed with probability $p = 0.1$ as a random control, where the horizontal and vertical components following the uniform distributions bounded by the minimal and maximal values of horizontal and vertical control signals in the training dataset, respectively. On each setting of large domain control and boundary control, we design two evaluation modes, fixed map (FM) and random map (RM), to test the generalization capability of each method:

**Fixed map (FM):** In this mode, all 50 test samples use the same obstacle configuration with training trajectories. However, the initial locations of test samples are different. In this mode, the testing and training initial conditions follow the same distribution (in-distribution test).

**Random map (RM):** In this mode, the 50 test samples use random obstacles' configuration in the fluid field. Specifically, the movement ranges for the five internal obstacles, whose default locations are shown in the top two rows in Figure 5, are as follows:

- The two upper obstacles can move downward, left, or right by no more than 3 grid spaces.

Table 6: Effect of the model horizon $H$ on 2D incompressible fluid control under the large domain control setting and fixed map (FM) evaluation mode. The average control objective $\mathcal{J}$ and inference time on a single NVIDIA A6000 48GB GPU with 16 CPU cores are reported. Bold font is the best model and the runner-up is underlined. The results of $H = 15$ are copied from Table 2 for comparison.

|  | $H = 6$ | | $H = 10$ | | $H = 15$ | |
|---|---|---|---|---|---|---|
|  | $\mathcal{J} \downarrow$ | time (s) $\downarrow$ | $\mathcal{J} \downarrow$ | time (s) $\downarrow$ | $\mathcal{J} \downarrow$ | time (s) $\downarrow$ |
| DiffPhyCon-1 | 0.8986 | 1022 | 0.5140 | 1216 | 0.5454 | 1677 |
| **CL-DiffPhyCon (ours)** | **0.6012** | **143.32** | **0.3367** | **136.81** | 0.3371 | **140.83** |

- The two middle obstacles can move upward, downward, left, or right by no more than 3 grid spaces.
- The one lower obstacle can move upward, left, or right by no more than 3 grid spaces.

In this mode, the testing and training initial conditions follow different distributions (out-of-distribution test). Note that although the obstacles' configuration varies across different test samples, for each test sample, its obstacles' configuration remains unchanged during the control process.

## G.2 IMPLEMENTATION

We use 3D U-net Ho et al. (2022) as architectures for the models $\boldsymbol{\epsilon}_\phi$ and $\boldsymbol{\epsilon}_\theta$. Two models are separately trained using the same training dataset. We use the MSE loss to train the models. Both models have $T = 600$ diffusion steps. The DDIM (Song et al., 2020) sampling we use has 75 diffusion steps with the hyperparameter $\eta = 0.3$ for the large domain control setting and 120 diffusion steps with the hyperparameter $\eta = 0.3$ for the boundary control setting. Hyperparameters of models and training are listed in Table 5.

## G.3 EFFECT OF MODEL HORIZON

The choice of the model horizon $H$ is determined by balancing efficiency and effectiveness. We conducted experiments on the 2D incompressible fluid control task under the fixed map (FM) setting with $H = 6$ and $H = 10$. The results are shown in Table 6, where we also copy the results of $H = 15$ from 2 together for comparison.

The results indicate that both our CL-DiffPhyCon and DiffPhyCon-1 yield similar performance when $H = 10$, compared to $H = 15$ as reported in Table 2. However, performance significantly deteriorates when $H$ decreases to 6, which is likely due to the shorter observation window leading to inaccurate future control objective $\mathcal{J}$ estimation and suboptimal guidance sampling (Line 5 in Algorithm 1). On the other hand, increasing $H$ beyond 15 significantly raises GPU memory costs during inference, thus not recommended. Across different values of $H$ (i.e., $H$=6, 10, and 15), our method consistently outperforms DiffPhyCon-1. Therefore, for this task, a horizon between 10 and 15 is appropriate. In practical applications, the optimal model horizon can be determined through multiple trials to balance performance and efficiency, similar to the approach used in diffusion policies (Chi et al., 2023) (see Figure 5 (left) in the referenced paper).

# H DOUBLE PENDULUM CONTROL EXPERIMENT

To investigate the performance of our CL-DiffPhyCon in lower-dimensional control problems, we performed experiments on controlling the system of a cart-inverted double pendulum system. The goal is to keep the double pendulum from falling down by exerting a force on the cart.

## H.1 SIMULATION ENVIRONMENT SETUP

The system consists of two point masses fixed on the end of two massless rigid rods, which are mutually connected and connected to a cart with frictionless hinges, as shown in Figure 6. The

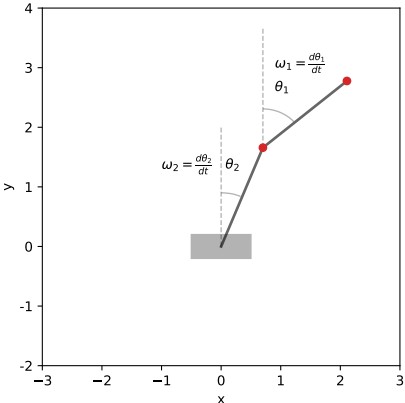

Figure 6: Illustration of the cart-double pendulum system. The angles are measured with respect to the vertical direction, and the angular velocities are defined as the time derivative of the angles.

cart-double pendulum system is simulated with the 4th order Runge-Kutta method (Ixaru & Vanden Berghe, 2004). In our experiment, the simulation time step interval is set to 5e-3 to ensure the accuracy of the simulation.

The physical quantities (unitless) are as follows: both rods have a length of 0.2, both point masses have a mass of 0.1, the mass of the cart is 1, and the gravitational acceleration is 9.8. The control force is restricted in [-10, 10]. Although the control signal and system states are relatively low-dimensional, we note the control task is non-trivial. Since no torque can be exerted on the hinges, the system can exhibit limited controllability. Besides, the rods deviate from the equilibrium point, and the non-linear or even chaotic dynamics impose significant challenges to its control. This system has been studied in the control research community (Hesse et al., 2018; Yamakita et al., 1993).

We would emphasize that this experiment provides a more easy-to-run verification of our method, as opposed to our 1D Burgers' equation experiment and 2D Navier-Stokes equation control experiments. Our cart-double pendulum environment can run at a refreshing rate of over 300 Hz on an Intel Xeon Platinum 8358 CPU, which allows real-time simulation (200 Hz refreshing rate in our setting) without high-end GPUs need.

## H.2 TRAINING AND EVALUATION DETAILS

In this experiment, we focus on an offline learning setting, where a training dataset containing expert action-state sequences is provided. The training dataset consists of 1e5 trajectories generated by an expert policy pretrained with PPO. Our CL-DiffPhyCon and baseline methods are trained with shared hyperparameters. In all diffusion models, VP-SDE training and ODE sampling are adopted (Song et al., 2021). All methods are trained for 2.9e5 steps, and 16-step Midpoint ODE solver (Karras et al., 2022) is used. The backbone is kept to an identically configured Transformer network. Other hyperparameters are also kept the same.

Our control task is defined on finite-length episodes, with a total of 1 second (200 steps) environment time. We use the metric *success rate* to reflect control performance, which is defined as the ratio of not-falling pendulums when the episode terminates. The not-falling criterion is defined as the upper point height larger than 0.9 maximum height. An ensemble of 100 cart-double pendulum systems is independently randomly initialized and controlled to fairly evaluate the performance of each task. The average and standard deviation are reported for five differently seeded runs to demonstrate the uncertainty of the evaluation.

To mimic realistic systems where the system state may be randomly perturbated, we introduce stochasticity into our cart-double pendulum system. At each step, the environment will perturb the system state (including the angle and angular velocities of two rods, the position, and the velocity of the cart) with probability. Once added, the perturbation will be sampled from a uniform

distribution in the range $[-0.02, 0.02]$. To make a comprehensive comparison, we report the results with three different rates: 0, 0.1, and 0.3.

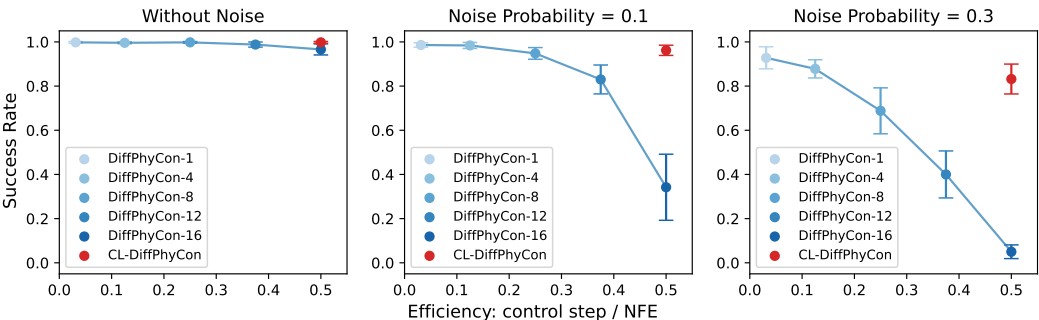

Figure 7: Pareto plot comparing our CL-DiffPhyCon and baselines. The error bar shows $\pm$ 1 standard deviation across 5 seeded runs.

### H.3 RESULTS

As shown in Figure 7, CL-DiffPhyCon achieves a remarkable balance between the control algorithm runtime and the control performance. Compared to the baselines of DiffPhyCon-15, DiffPhyCon-12, and DiffPhyCon-8, our CL-DiffPhyCon provides much higher control performance while only requiring similar neural function evaluations (NFE, as in Song et al. (2021); Karras et al. (2022)) per environment step. Although DiffPhyCon-4 and DiffPhyCon-1 show marginal performance improvement over our CL-DiffPhyCon, they are nearly one order of magnitude slower.

For a more intuitive demonstration of the performance comparison, we include a comprehensive visualization where our method is compared with two selected baselines under three different random perturbation settings, as shown in Figure 8. Compared to DiffPhyCon-15, Our CL-DiffPhyCon is able to stabilize the inverted double pendulum for a longer time (subfigure (a)), and the final upper mass state $(\theta_2, \omega_2)$ is closer to the equilibrium point $(0, 0)$ (subfigure (b)). Additionally, although the performance of our CL-DiffPhyCon is similar to that of DiffPhyCon-1, CL-DiffPhyCon costs about only $1/15$ computation time of it.

## I  1D VISUALIZATION RESULTS

We present the visualization results of our method and baselines under noise-free and half domain settings in Figure 9, 10, 11, and 12, respectively. Under each setting, we present the results of four randomly selected samples from the test set.

## J  2D VISUALIZATION RESULTS

More visualization results of our method and compared baselines are presented. For both fixed map (Figure 13 and Figure 14) and random map (Figure 15 and Figure 16) settings, we present two randomly selected test samples. Our method outperforms baselines on average. Note that the fluid dynamics is very sensitive to the generated control signals. Thus different methods may perform dramatically differently on the same test sample. However, we find that each method performs stably on each test sample through multiple evaluations.

## K  BASELINES

For RDM (Zhou et al., 2024), it involves two hyperparameters to determine when to update the previously planned control sequences or replan from scratch. Their default values do not work well on our 1D and 2D tasks. On the 1D task, they are too small, which results in very expensive

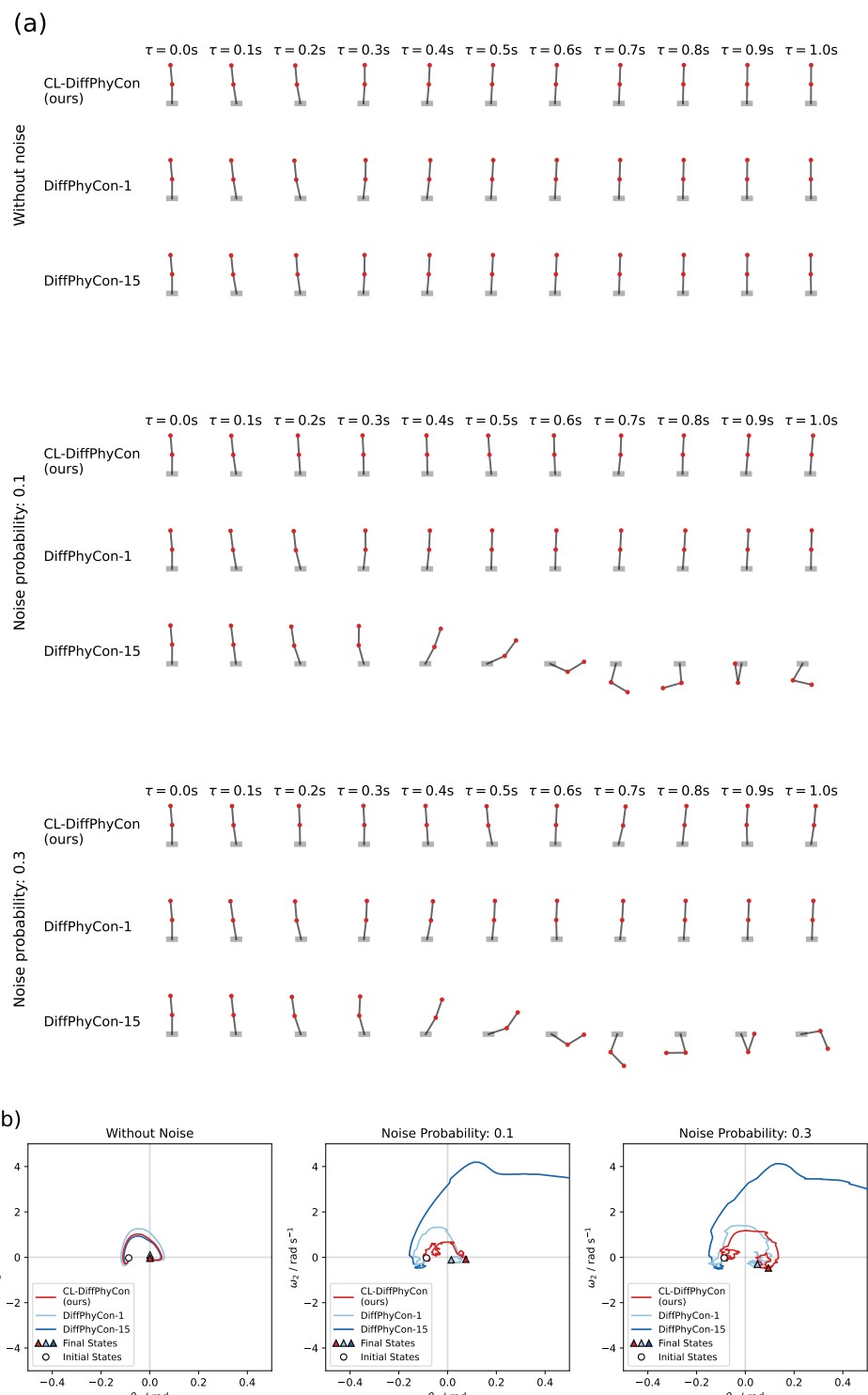

Figure 8: **Visualization of and instance of the cart-double pendulum under control**. The goal is to keep the double pendulum from falling down by exerting a force on the cart over 1 second (200 control steps) sequentially. Three control methods are applied under three different perturbation rates. In each setting, 10 snapshots of the pendulum are shown in subfigure (a), whereas the corresponding trajectories in the phase space are shown in subfigure (b).

replanning from scratch almost every physical time, although its performance is still unsatisfactory. On the 2D task, they are too large, leading to few replannings and inferior results. Therefore, we run extensive evaluations to search for their best values. On the 1D task, we find that the best pair of values for the two hyperparameters is (0.06, 0.1) in the noise-free setting. On the 2D task, we find that the best pair of values is (0.0005, 0.002).

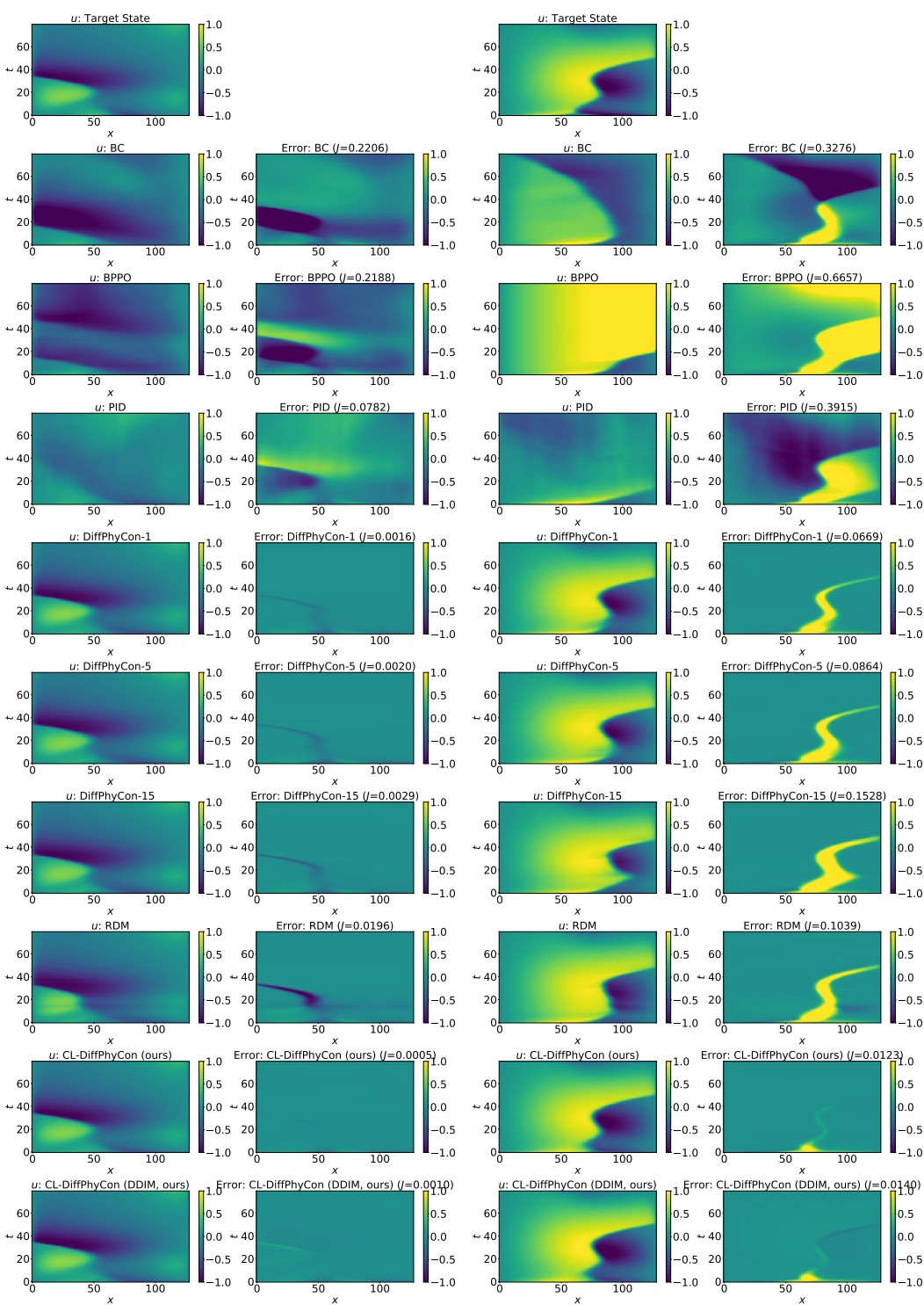

Figure 9: **Two visualizations results of 1D Burgers' equation control under the noise-free setting**. The first line is the target $u_d(t, x)$ and the error $u(t, x) - u_d(t, x)$ measures the gap between the state under control and target. The horizontal axis is the time coordinate and the vertical axis is the state.

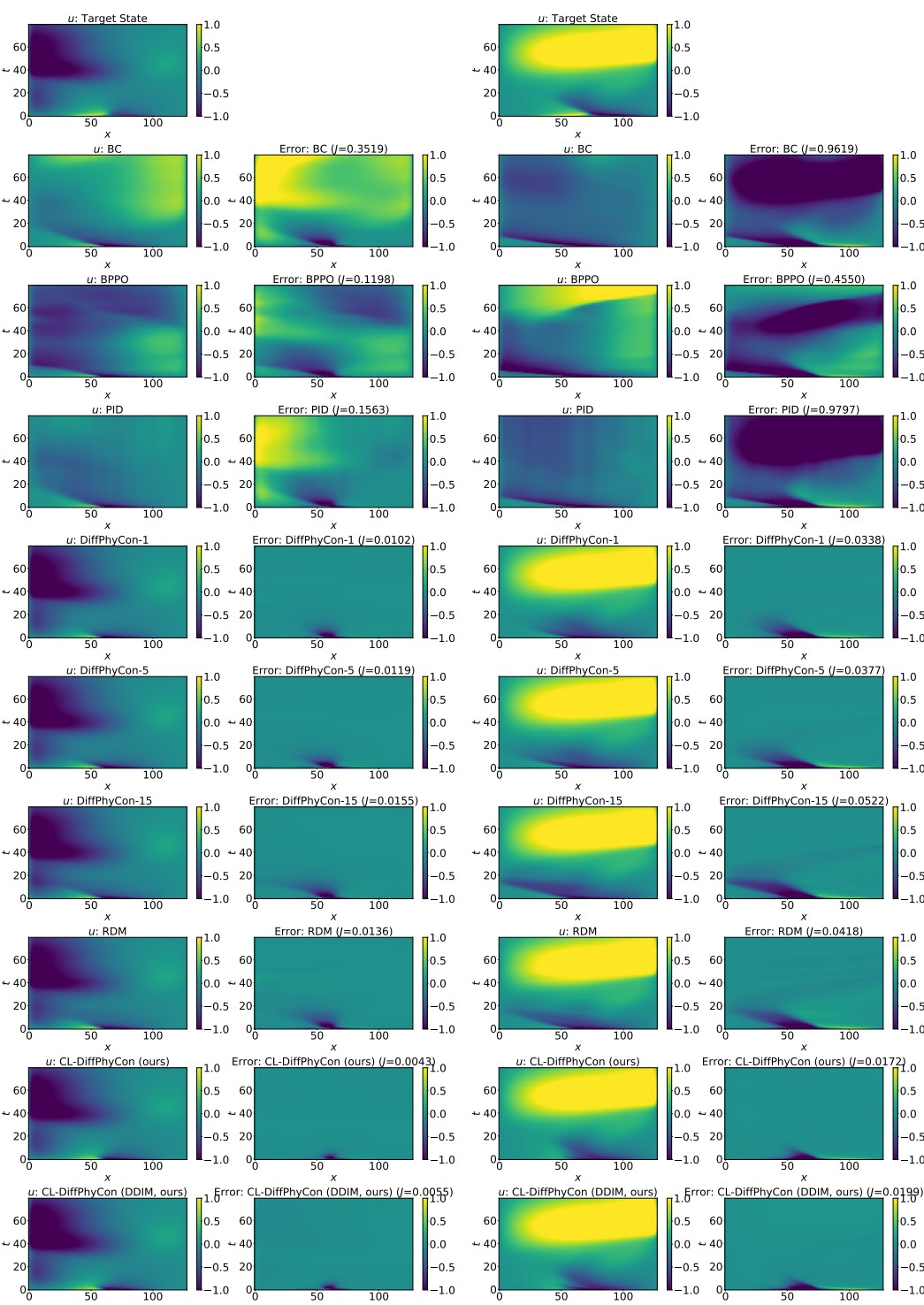

Figure 10: **Two visualizations results of 1D Burgers' equation control under the noise-free setting**. The first line is the target $u_d(t, x)$ and the error $u(t, x) - u_d(t, x)$ measures the gap between the state under control and target. The horizontal axis is the time coordinate and the vertical axis is the state.

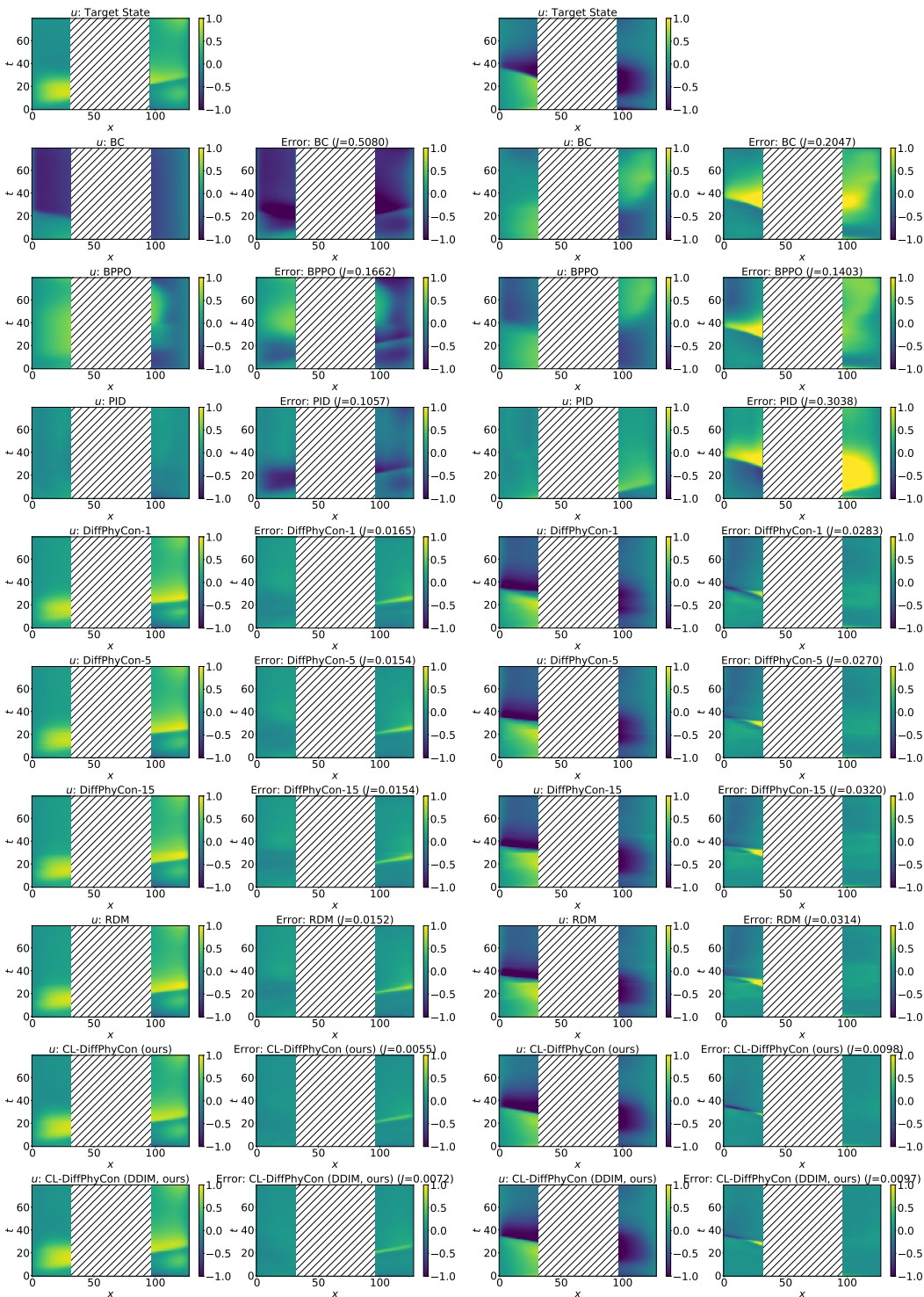

Figure 11: **Two visualizations results of 1D Burgers' equation control under the half domain setting**. The first line is the target $u_d(t,x)$ and the error $u(t,x) - u_d(t,x)$ measures the gap between the state under control and target. The horizontal axis is the time coordinate and the vertical axis is the state. The white area with oblique lines in the middle represents an unobservable area.

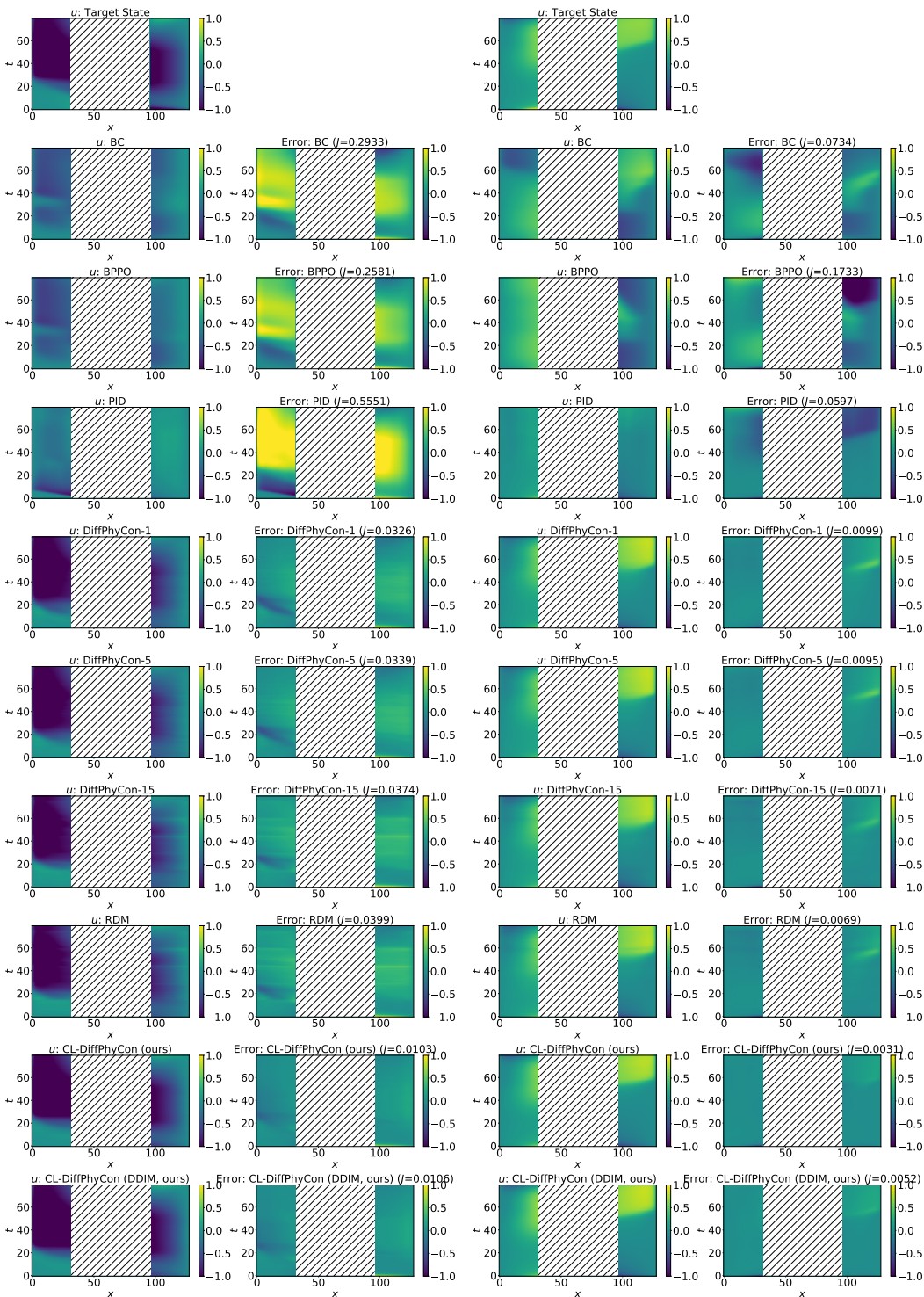

Figure 12: **Two visualizations results of 1D Burgers' equation control under the half domain setting**. The first line is the target $u_d(t, x)$ and the error $u(t, x) - u_d(t, x)$ measures the gap between the state under control and target. The horizontal axis is the time coordinate and the vertical axis is the state. The white area with oblique lines in the middle represents an unobservable area.

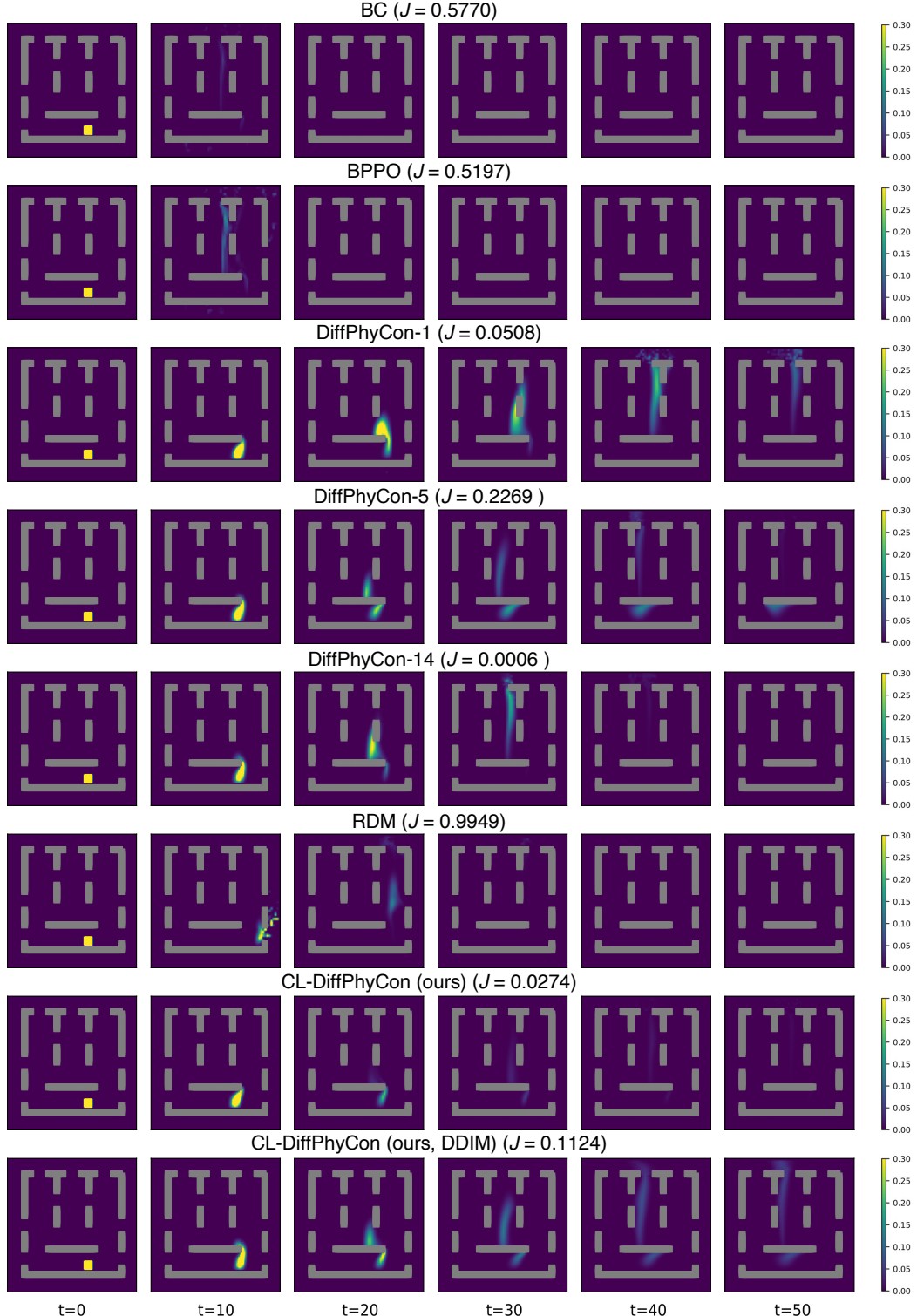

Figure 13: **Visualizations results of 2D fluid control by our CL-DiffPhyCon method and baselines for a same test sample**. This is an example of the *fixed map* setting. Each row shows six frames of smoke density. The smaller the value of $J$, the better the performance.

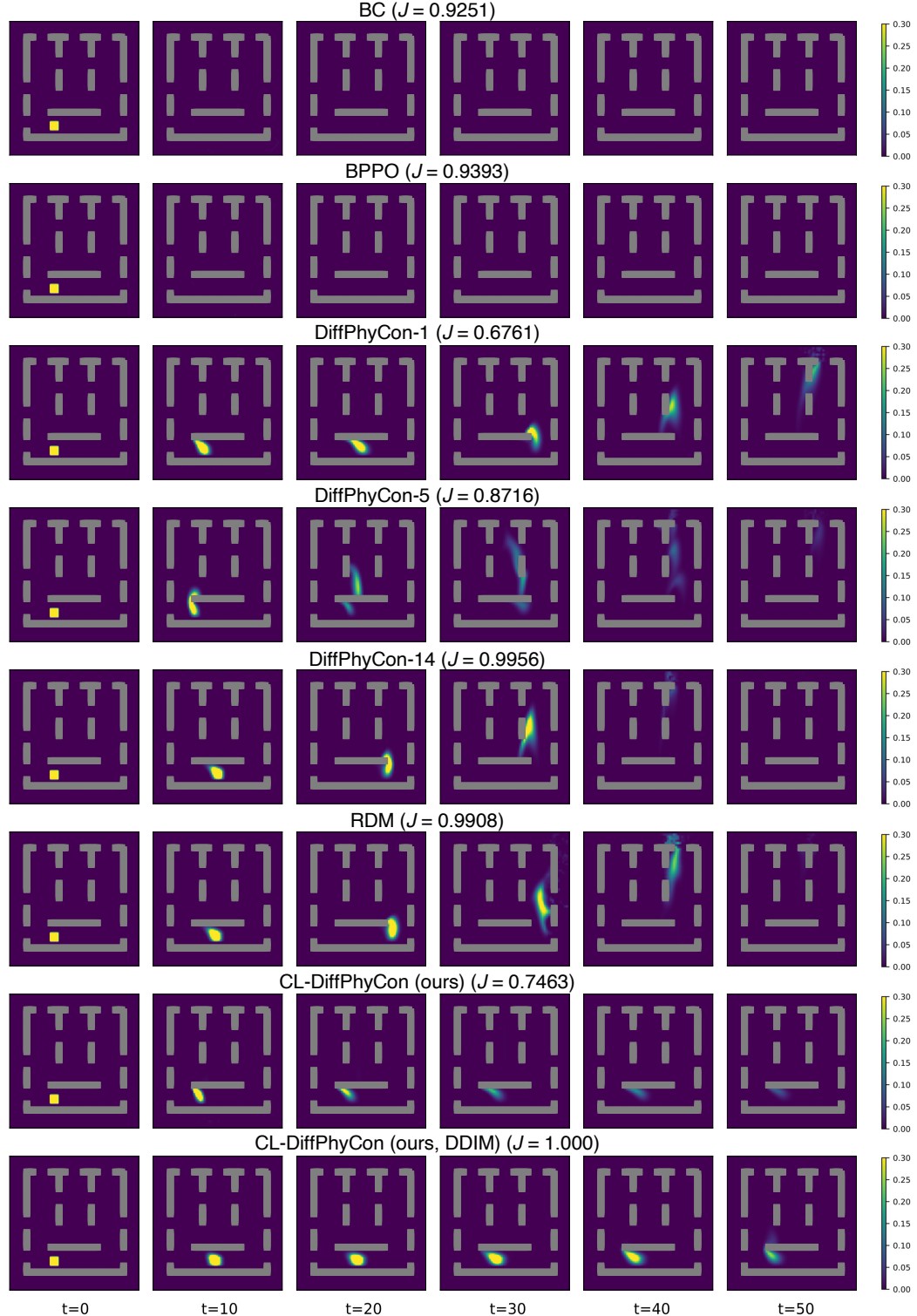

Figure 14: **Visualizations results of 2D fluid control by our CL-DiffPhyCon method and baselines for the same test sample**. This is an example of the *fixed map* setting. Each row shows six frames of smoke density. The smaller the value of $J$, the better the performance.

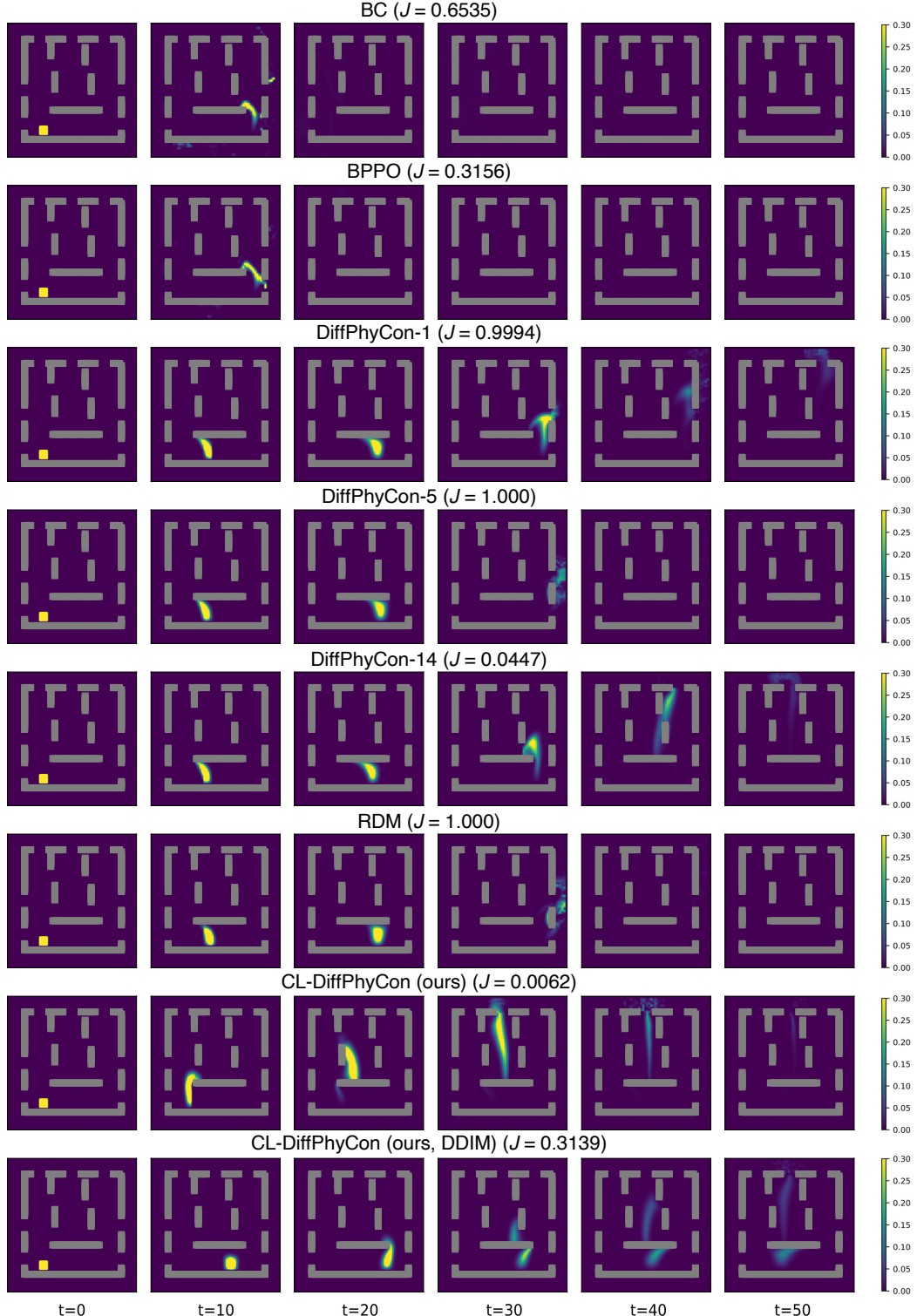

Figure 15: **Visualizations results of 2D fluid control by our CL-DiffPhyCon method and baselines for the same test sample**. This is an example of the *random map* setting. Each row shows six frames of smoke density. The smaller the value of $J$, the better the performance.

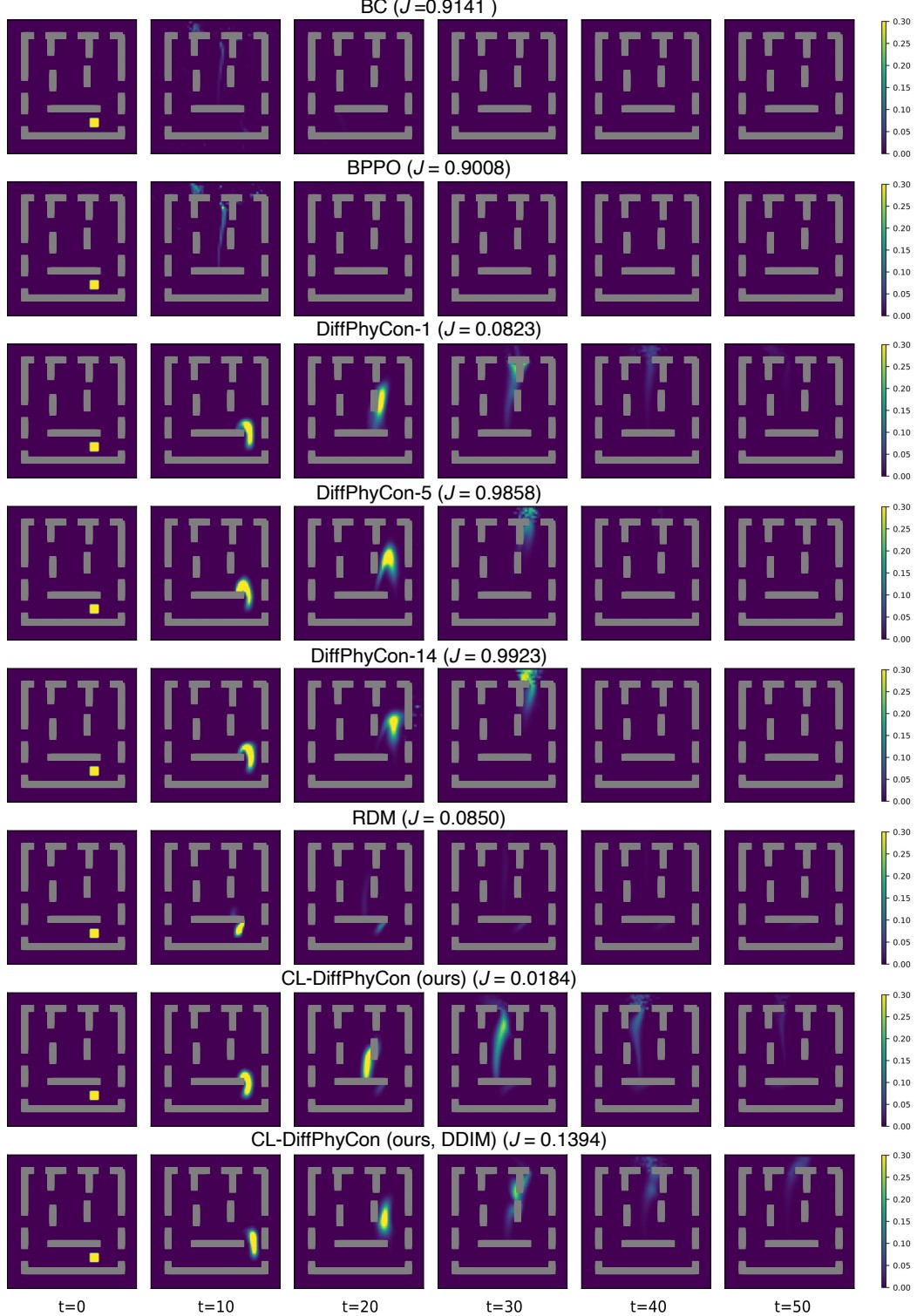

Figure 16: **Visualizations results of 2D fluid control by our CL-DiffPhyCon method and baselines for the same test sample**. This is an example of the *random map* setting. Each row shows six frames of smoke density. The smaller the value of $J$, the better the performance.

