# OpenReview forum: "CL-DiffPhyCon: Closed-loop Diffusion Control of Complex Physical Systems"
_ICLR.cc/2025/Conference — ICLR 2025 Poster_

### Official Review · Reviewer_pd6k · 2024-11-01

**Soundness:** 2
**Presentation:** 1
**Contribution:** 2
**Rating:** 3
**Confidence:** 4

**Summary:**

The paper studies the problem of designing feedback control algorithms using diffusion techniques. Differently from open-loop control strategies, which compute a pre-determined control sequence that is applied to a system over time, closed-loop methods use information about the state of the system to tune the control inputs to compensate for deviations of the system trajectories from the predicted one, due to unmodelled dynamics or noise. The paper focuses on the computational complexity of the methods, and it proposes a technique to render closed-loop diffusion-based control computationally efficient.

**Strengths:**

The overall objective of the paper is interesting and timely: to design closed-loop diffusion-based control algorithms. The issue of computational complexity is important, since diffusion-based methods may work well but require extensive computations, making them difficult to use in feedback settings.

**Weaknesses:**

The paper is poorly presented, to an extent that makes the contributions unclear and difficult to appreciate. This start with the definition of the problem (where what is known/unknown/available to the designer is unclear, where there is confusion between system and environment, where   the stated problem is inherently open-loop and the feedback structure is not evident), and continues into the description of the proposed method (the synchronous and asynchronous parts are vague and difficult to grasp). Additionally, and importantly, the paper doesn't seem to offer performance guarantees for the proposed method, which makes it difficult to advocate for the proposed method in any practical use.

**Questions:**

A careful rewriting of the manuscript seems necessary to clarify the problem setting and contribution, and to appreciate the differences between the propose method and the existing ones. A characterization of the performance of the proposed method is necessary, otherwise the  method takes the form of a heuristic that may not be desirable in controls applications.

As a side note, the problem described in equation (1) is inherently an open loop problem, since no structure is imposed on the input w. In fact, the set of open-loop sequences include the set of closed-loop sequences, but the converse may not be true. Since the system dynamics are deterministic, it is also unclear why using a closed-loop strategy would be necessary. Finally, the used notation (w for control inputs and u for the system state) is very unusual and, to the best of my knowledge, never been used in any controls paper.

---

> ### Author Response · Authors · 2024-11-23
> **Official Response to Reviewer pd6k (1)**
>
> We thank the reviewer for the detailed feedback and helpful suggestions. We appreciate the reviewer for noting that our objective of designing closed-loop diffusion-based control algorithms is interesting and timely. The reviewer also highlights the importance of addressing computational complexity of diffusion-based methods. Below, we address the reviewer's concerns one by one.
>
> > Comment 1: The paper is poorly presented. This starts with the definition of the problem (where what is known/unknown/available to the designer is unclear, where there is confusion between system and environment, where the stated problem is inherently open-loop and the feedback structure is not evident). As a side note, the problem described in equation (1) is inherently an open loop problem, since no structure is imposed on the input w. In fact, the set of open-loop sequences include the set of closed-loop sequences, but the converse may not be true. Since the system dynamics are deterministic, it is also unclear why using a closed-loop strategy would be necessary.
>
> **Answer:**   Thank you for your feedback. To reduce ambiguity, we **replace the keyword** **"environment"**  to **"system dynamics"**, still denoted by $G$, which represents the transition of states under external control at each time step, i.e., $u_{\tau+1}=G(u_{\tau}, w_{\tau+1})$ if deterministic, or $u_{\tau+1}$ follows a distribution $p(u_{\tau+1}|u_{\tau}, w_{\tau+1})$ if stochastic. Below, we clarify potential misunderstandings regarding the problem definition:
>
> 1. **Known and Unknown Aspects of the Problem:**
>
>     The designer **has access to the following:**
>
>     - **Initial state ($u_0$):** The initial condition that sets the basis for the control process, which is known to the designer.
>     -  **System dynamics ($G$):** It represents the transition of states over time in the system, typically determined by implicit PDEs. The evolution of states can only be observed through state measurement. The dynamics could be deterministic or stochastic.
>     -  **Control objective ($\mathcal{J}$):** It is defined as a function of the state trajectory ($\{u_1, \ldots, u_N\}$) and the control sequences ($\{w_1, \ldots, w_N\}$), and it drives the optimization of control decisions over physical time steps from $\tau = 1$ to $\tau = N$. $\mathcal{J}$ is specified by the designer.
>
>     The above three components, i.e., initial state, system dynamics, and control objective define a complex physical systems control problem, as clarified at the beginning of Section 3.1 of the revised manuscript.
>
> 2. **Clarification on Open-Loop vs. Closed-Loop:**
>
>     Open-loop control involves predetermined actions without feedback, making it suitable only for systems with simple or well-known dynamics. Closed-loop control, by contrast, adapts based on real-time feedback, which is crucial for complex or evolving systems. Since the evolution of states can only be observed through measurement of the state, the problem in **Equation (1) is inherently closed-loop**. Namely, each control action depends on real-time feedback, making it impossible to pre-compute all control signals simultaneously, thereby necessitating a closed-loop strategy, with the feedback structure at each time step $u_{\tau+1}$ as provided by the environment.
>
>     **To clarify these points, we have revised the manuscript:**
>
>     - In **Section 3.1, a more explicit definition** of what information is known and what is unknown to the designer. In particular, we emphasized the **property of system dynamics** and the **closed-loop nature** of our optimization problem shown in Equation (1).
>     - In Figure 1 and Figure 2, we replace "environment" with "system dynamics".

---

> > ### Comment · Reviewer_pd6k · 2024-11-25
> > **Deterministic vs stochastic**
> >
> > We thank the Authors for their careful response. However, we remain confused by this manuscript, and particularly by the control problem it claims to address.
> >
> > How does noise enter the dynamics G? The system dynamics do not show any noise and it's unclear how this happens. What are the property of the noise process? It is difficult to accept that the proposed method can solve *any* optimal control problem for *any* kind of deterministic or stochastic dynamical system with *any* kind of noise. Does the paper offer guarantees on the optimality of the resulting control policy? Are the dynamics G stable and is the resulting closed-loop system stable? Since the control input is itself stochastic (sampled from a distribution conditioned on the current stat), how is stability and performance measured and guaranteed?

---

> ### Author Response · Authors · 2024-11-23
> **Official Response to Reviewer pd6k (2)**
>
> > Comment 2: The contributions are unclear. A careful rewriting of the manuscript seems necessary to clarify contributions, and to appreciate the differences between the propose method and the existing ones.
>
> **Answer:** Thanks for your comment. **Contributions were summarized in the last paragraph of Section 1 "Introduction" of our original submission.** For clarity, we highlight them below with additional details:
>
> **Contributions:**
>
> 1. **A novel closed-loop diffusion control framework (CL-DiffPhyCon):**
>
>     We propose a closed-loop diffusion-based control framework that directly interacts with the environment to generate adaptive, real-time control actions. The primary advantage of this framework is its significant improvement in computational efficiency, marking important progress in the field of diffusion-based control. Notably, reviewer sLTC recognized that "This idea addresses one of the main shortcomings of controllers based on diffusion policies."
>
>     Technically, we introduce a novel asynchronous denoising process, enabling closed-loop operation while reducing computational complexity. The proposed asynchronous diffusion model has three notable properties:
>
>     - **Efficient asynchronous denoising:** The asynchronous diffusion model is a novel denoising model that enables parallel denoising across the direction of physical time steps, allowing earlier decision steps to be sampled sooner. By incorporating feedback into this asynchronous denoising process, closed-loop control is achieved, and redundant sampling steps, as in previous diffusion models, have been eliminated.
>     - **Theoretical derivation, not heuristic:** Our model is derived from a rigorous theoretical analysis of the target distribution. We decompose the complex joint distribution into a product of conditional distributions, which results in only two types of distributions to learn. This theoretical framework may inspire future research on analyzing such complex distributions.
>     - **Independent effect of acceleration beyond existing fast sampling methods**: Our method already significantly improves control efficiency compared to prior diffusion-based methods and can be further accelerated by incorporating fast sampling techniques, such as DDIM. This demonstrates that our method offers an independent and substantial acceleration effect beyond the current fast sampling techniques in the diffusion model community.
>
> 2. **Significant Performance on Complex Physical Systems:**
>     - We validate our method on complex physical systems, such as the 1D Burgers' equation and 2D fluid control, showing superior performance with significant sampling acceleration—even in challenging scenarios like partial observations (Table 1), new noise type experiments (Table 1), stochastic control actions (Table 2), and out-of-distribution environments (Table 2 and Figure 3). These results underscore the practical effectiveness of our approach for systems with complex, evolving dynamics.
>
>    **We have modified the last paragraph of Section 1 "Introduction" in the revised manuscript according to the above discussion.**
>
> *(Continued in next piece of response)*

---

> ### Author Response · Authors · 2024-11-23
> **Official Response to Reviewer pd6k (3)**
>
> > Comment 2: The contributions are unclear. A careful rewriting of the manuscript seems necessary to clarify contributions, and to appreciate the differences between the propose method and the existing ones.
>
> **Answer:** *(Continued from previous piece of response)*
>
> **Differences from Existing Methods:**
>
> Existing methods are discussed in **Section 2 of our original submission** and can be categorized as diffusion-based control, reinforcement learning, imitation learning, and classical control methods such as PID and MPC. Below, we further clarify the distinctions:
>
> 1. **Diffusion-based Control Methods (DiffPhyCon, Adaptive Replanning):**
>     - Unlike existing diffusion-based control methods like DiffPhyCon and adaptive replanning, our approach incorporates real-time feedback, achieving a closed-loop structure rather than an open-loop or purely replanning-based approach. If DiffPhyCon were to conduct replanning at every physical time step, it could achieve closed-loop control; however, the sampling cost would be approximately $H$ times higher ($H$ is the model horizon, i.e., the window size of physical time steps) than our method, as compared intuitively in the left and right subfigures of Figure 1.
>     - Unlike these methods, which typically require full synchronous denoising inside the diffusion model horizon at each physical time step, we use an **asynchronous denoising framework** that significantly reduces computational costs and ensures timely response, making our approach practical for real-time applications.
>     - A further comparison with adaptive replanning (RDM) demonstrates that our method is inherently closed-loop (RDM is open-loop), more efficient (as RDM involves extra likelihood computations and more sampling steps), and easier to use (RDM requires tuning two hyperparameters extensively for new tasks). These differences have been discussed in Section 2 of our original submission.
>
> 2. **Reinforcement Learning (RL) and Imitation Learning Methods:**
>     - Previous work [1] has demonstrated that diffusion-based control outperforms RL and imitation learning in controlling high-dimensional nonlinear systems due to the superior capacity of diffusion models in modeling complex system dynamics [2,3]. Our CL-DiffPhyCon extends this advantage by incorporating closed-loop inference, achieving significant gains in efficiency and performance.
>
> 3. **Classical control methods such as PID and MPC:**
>     - Classical methods face considerable challenges in controlling high-dimensional, complex systems. For instance, as a common single-input single-output (SISO) controller, the PID is difficult to directly applied to multiple-input multiple-output (MIMO) systems like our 2D incompressible fluid control task (the application of PID to the MIMO system often requires additional decoupling and planning modules, and the control effect of PID is greatly influenced by decouplers and planners, making it difficult to compare fairly). MPC depends heavily on precise models, which are difficult to obtain and often inaccurate for complex physical systems. In contrast, our method is flexible and excels at handling high-dimensional, complex systems without relying on precise models, making it particularly effective for scenarios where system dynamics are implicit or evolving and where classical control methods fall short.
>
>     **We have emphasized these differences more clearly in Section 2 "Related Work" in the revised manuscript.**
>
> [1] Long Wei et al. “DiffPhyCon: A Generative Approach to Control Complex Physical Systems”. NeurIPS 2024.
>
> [2] Aliaksandra Shysheya et al. "On conditional diffusion models for PDE simulations." NeurIPS 2024.
>
> [3] Jiahe Huang et al. "DiffusionPDE: Generative PDE-Solving under Partial Observation." NeurIPS 2024.

---

> > ### Author Response · Authors · 2024-11-23
> > **Official Response to Reviewer pd6k (4)**
> >
> > > Comment 3: The paper is poorly presented, to an extent that makes the contributions unclear and difficult to appreciate. 1. ..., 2. and continues into the description of the proposed method (the synchronous and asynchronous parts are vague and difficult to grasp).
> >
> > **Answer:** Thanks for your comment. We would like to clarify the description of the proposed method from four aspects: purpose, challenges, our solution, and our presentation.
> >
> > - **Our purpose:** The purpose of our work is to find an efficient approach to generate closed-loop control sequences by diffusion models. It means **we hope the diffusion model can utilize the feedback of the environment to generate the control action step by step.**
> > - **The challenges:** However, **there is an inherent problem in diffusion models that they sample from a joint distribution instead of autoregress from conditional distribution.** To generate h steps of actions, diffusion models can only simultaneously sample the whole h-step sequence from the joint distribution, where the feedback between these h steps could not be used in this process (This is shown in the middle part of Figure 1 of the original submission). Otherwise, due to the large number of physical time steps, it takes intolerable time costs to sample h times by using a full sampling process at each physical time step (This is shown in the left part of Figure 1 of the original submission). For example, if the denoising process needs k steps, the total number of model inferences is hk when replanning for every action step.
> > - **Our solution:** What we did is propose a new diffusion model method that suits the sequence control problem. The motivation is that only by generating action steps in order can feedback be utilized. So **our main contribution is proposing such an asynchronous diffusion model** (This is shown in the right part of Figure 1 of the original submission), which needs another ordinary (synchronous) diffusion model to help initialize the sequence generation (Left part of Figure 1 of the original submission). The asynchronous diffusion model makes the denoising process asynchronous for the control sequence, which allows parallel denoising and makes the earlier decision steps sampled earlier. Using this asynchronous diffusion model, closed-loop control can be achieved by simply plugging in feedback to asynchronous denoising (The whole method is stated in **Algorithm 1** of the original submission).
> > - **Our presentation:**
> >     -  First, our paper introduces a control method grounded in **solid theoretical support, not heuristics**. To facilitate theoretical analysis, we use conventional stochastic differential equation (SDE) formulations, widely adopted in diffusion model literature [1, 2, 3]. We **provide self-contained preliminaries** in Section 3.2 and Appendices A and B in our original submission to aid understanding. **Familiarity with these materials is essential for grasping our method**, particularly its synchronous and asynchronous components.
> >     - Second, the synchronous and asynchronous models are **derived theoretically** from the joint distribution $p(z_1, z_2, \cdots, z_N|z_0)$, which we aim to sample from. A single synchronous model, as used in DiffPhyCon, struggles to balance closed-loop control with efficient sampling for long trajectories. Our key contribution, as detailed in Theorem 1, is the decomposition of this joint distribution into sequential conditional distributions, leveraging the **Markov assumption of the physical system, making it not difficult to understand**. This decomposition enables the construction of two diffusion models (synchronous and asynchronous) **using standard SDE formulations**, with the novel asynchronous denoising schedule described in Section 4.3.
> >     - Finally, even without delving into the technical details, we believe Figure 1, Figure 2, and Algorithm 1 provide an accessible overview of our method. **Figures 1 and 2 illustrate the core intuitions and comparisons with existing methods**, while **Algorithm 1 presents the step-by-step implementation**. Together, these visual aids and procedural descriptions are designed to convey the fundamental concepts of our method intuitively.
> >
> > Therefore, we conclude that **our presentation is well-suited to balance theoretical rigor and practical clarity**. We hope that this response addresses your concerns and are happy to provide further clarifications if needed.
> >
> > [1] Yang Song et al. "Score-Based Generative Modeling through Stochastic Differential Equations", ICRL 2021.
> >
> > [2] Hyungjin Chung et al. "Diffusion Posterior Sampling for General Noisy Inverse Problems", ICLR 2023.
> >
> > [3] Tero Karras et al. "Elucidating the design space of diffusion-based generative models". NeurIPS 2022.

---

> > > ### Comment · Reviewer_pd6k · 2024-11-25
> > > **Presentation of the paper**
> > >
> > > I respect the Authors' opinion of the presentation of their work, as well as their efforts in clarifying the concerns raised by the reviewers. However, I still find the paper very hard to read and appreciate. Perhaps the paper is tailored to experts in the field and those that already know the related and cited works, but I strongly doubt that anyone else can appreciate this contribution.

---

> ### Author Response · Authors · 2024-11-23
> **Official Response to Reviewer pd6k (5)**
>
> > Comment 4: Additionally, and importantly, the paper doesn't seem to offer performance guarantees for the proposed method, which makes it difficult to advocate for the proposed method in any practical use. A characterization of the performance of the proposed method is necessary, otherwise the method takes the form of a heuristic that may not be desirable in controls applications.
>
> **Answer:** Thank you for your comment. We would like to clarify that **our method is not heuristic**. The use of synchronous and asynchronous diffusion models is theoretically derived from Theorem 1, and the role of feedback in the control process is explained in **Proposition 2 of our original submission**. Specifically, our original submission provides theoretical support for the following claims:
>
> - Our modified diffusion models can accurately learn and sample from the distribution of control sequences, providing true dynamics for control through guidance sampling.
> - Integrating feedback into the asynchronous denoising process has a qualitative impact on the control sequences.
>
> To optimize the control objective during inference, we utilize guidance sampling for action search, as originally proposed in the diffusion model literature [1, 2]. According to these works, guidance sampling is proven to be able to achieve conditional generation, allowing the sampling results to optimize specific objectives. Practically, this process involves applying gradient descent to diffusion model sampling, as demonstrated in Equations (5) and (11) of our original submission. However, while guidance sampling is theoretically validated for conditional sampling and has shown empirical success across multiple domains, establishing a formal theoretical bound for its exact optimization performance remains an open problem in the diffusion model community. We hope future research will provide further theoretical insights, but it is beyond the scope of the present work. **We have pointed out this as a limitation of our work in Section 6 "Conclusion and Limitation" of the revised manuscript.**
>
> Besides, the challenge in providing performance guarantees also stems from the inherent complexity of the physical systems under control, including non-linearity and stochasticity. Similar difficulties are faced by other control methods, such as PID, which, despite lacking formal guarantees, remain among the most widely used approaches in engineering.
>
> Empirically, we have conducted extensive experiments, including those presented in the original submission and additional results during the rebuttal, which demonstrate the capability of our method in handling complex control tasks effectively. These results highlight the potential applicability of our approach in real-world control settings, even in the absence of formal performance guarantees.
>
> [1] Yang Song et al. "Score-Based Generative Modeling through Stochastic Differential Equations", ICRL 2021.
>
> [2] Hyungjin Chung et al. "Diffusion Posterior Sampling for General Noisy Inverse Problems", ICLR 2023.
>
> > Comment 5: The used notation (w for control inputs and u for the system state) is very unusual and, to the best of my knowledge, never been used in any controls paper.
>
> **Answer:** Thank you for your feedback regarding the notations used in our manuscript. To ensure clarity, we have defined all notations in Section 3 before detailing our proposed method in Section 4 of our original submission. We believe this introduction to notations helps mitigate potential confusion and allows readers from diverse backgrounds to follow the content easily.
> We understand that selecting appropriate notations requires careful consideration, particularly in such interdisciplinary work that spans multiple research communities—physical systems (PDEs), control theory, and diffusion models—each of which has its own notational conventions. Our primary principle of choosing notations is to highlight clearly the differences from existing methods. Since our work builds on and extends the recent study [1], we chose to use consistent notations, specifically $u$ for state and $w$ for control, to facilitate continuity and ease of comparison. Additionally, in the PDE control community [2,3], $u$ is commonly used for state variables, which further supports our choice.
>
> [1] Long Wei et al. “DiffPhyCon: A Generative Approach to Control Complex Physical Systems”. NeurIPS 2024.
>
> [2] Rakhoon Hwang et al. "Solving pde-constrained control problems using operator learning". AAAI 2022.
>
> [3] Philipp Holl et al. "Learning to Control PDEs with Differentiable Physics." ICLR 2020.

---

> > ### Author Response · Authors · 2024-11-25
> > **A gentle reminder: please respond to our rebuttal**
> >
> > Dear Reviewer pd6k,
> >
> > Thank you for reviewing our work. We have carefully responded to all your comments and questions in our rebuttal.
> >
> > The deadline for the author-reviewer discussion is coming up soon. Could you please take a moment to review our responses? If you find them satisfactory, we hope you might consider adjusting your initial rating. Any additional comments or feedback would be appreciated.
> >
> > Thanks a lot!
> >
> > Best regards,
> >
> > Authors

---

> > ### Comment · Reviewer_pd6k · 2024-11-25
> > **Difficulty to obtain theoretical guarantees**
> >
> > I agree with the authors that obtaining guarantees on the performance of the proposed control action is a difficult task. However, I'm not sure about the usefulness of any control method that comes without guarantees. This would be a heuristic, in my opinion, that can be used in simulations but probably not on any real application.
> >
> > Would it be possible to obtain guarantees by considering simpler, possibly linear, system dynamics? I'm unsure what the role of PDE dynamics is, since the paper doesn't enter into any details regarding the system under consideration. Using a simpler setting would help to clarify the proposed control method, which is otherwise obscured by the complexities of the setting that are tangential to the proposed method.

---

> ### Author Response · Authors · 2024-11-28
> **Official Response to Reviewer pd6k: Deterministic vs stochastic (1)**
>
> > **Comment**: We thank the Authors for their careful response. However, we remain confused by this manuscript, and particularly by the control problem it claims to address. How does noise enter the dynamics G? The system dynamics do not show any noise and it's unclear how this happens. What are the property of the noise process? It is difficult to accept that the proposed method can solve any optimal control problem for any kind of deterministic or stochastic dynamical system with any kind of noise. Does the paper offer guarantees on the optimality of the resulting control policy? Are the dynamics G stable and is the resulting closed-loop system stable? Since the control input is itself stochastic (sampled from a distribution conditioned on the current stat), how is stability and performance measured and guaranteed?
>
> **Answer:** Thanks for your feedback. We agree that in the real-world practice, the system often encounters the issue of noise. As presented in the revised manuscript (Section 3.1), we assume that the system dynamics $G$ has the following form $u_{\tau + 1}  = G(u_{\tau}, w_{\tau + 1}, \xi_{\tau + 1})$, where $\{\xi_{\tau + 1}\}$ are independent random variables. When $G$ is stochastic, $\{\xi_{\tau + 1}\}$ are nonzero. In our revised manuscript, we have considered three kinds of noise: **system**, **measurement**, and **control** noise, as suggested by Reviewer GNga.
>
> **In 1D experiment**, the **deterministic** Burgers' equation system has the following form:
>     $G_0: \frac{\partial u}{\partial \tau} = -u\cdot \frac{\partial u}{\partial x} + \nu \frac{\partial^2u}{\partial x^2} + w(\tau,x)$,
>     where $u$ denotes the state variable and $w$ represents the control. Then, we consider the system and measurement noise:
>
> - For **system noise**, we follow the setting in previous work [1], which perturbs the system by Gaussian noise with standard deviation $\sigma=0.025$. The augmented  dynamics with additive Gaussian noise have the following form:
>     $G_1: \frac{\partial u}{\partial \tau} = -u\cdot \frac{\partial u}{\partial x} + \nu \frac{\partial^2u}{\partial x^2} + w(\tau,x) +\xi_\tau,  \xi_\tau\sim\mathcal{N}(0,\sigma)$.
>     During numerical simulation, the perturbed system $(G_1)$ supersedes the nominal system $(G_0)$ when implementing the control input $w$. Consequently, the feedback state $u$ is obtained as the solution to the stochastic PDE $(G_1)$.
> - For **measurement noise**, it evaluates the case where sensors have noise. The feedback state is characterized by the superposition of the nominal solution $u$ to system $(G_0)$ and a Gaussian noise term $\xi_\tau\sim\mathcal{N}(0,\sigma)$ ($\sigma=0.025$ in the experiments), thereby incorporating measurement noise into the feedback loop while preserving the dynamics of the nominal system.
>
> Please note that our method does not rely on any numerical simulation and can be applied in practice, and the purpose of noise cases is to simulate real-world scenarios. We have added the description of noise in Appendix D.2 of the revised manuscript.
>
> **In 2D experiments**, we consider the **control noise**, which means stochastic perturbation during the execution of the control signal in each time step, following the setting in previous work [2]. Specifically, as detailed in **Section 5.3 of the original submission**, for each trajectory of length $N=64$, the control signal in each physical time step is executed with probability $p=0.1$ as a random action, where the horizontal and vertical components following the uniform distributions bounded by the minimal and maximal values of horizontal and vertical control signals in the training dataset, respectively.
>
> The results in **Table 1 and Table 2 in the revised manuscript** demonstrate the superiority and robustness of our approach against different types of noise.
>
>
> [1] Cagatay Yildiz et al. "Continuous-time model-based reinforcement learning." ICML 2021
>
> [2] Siyuan Zhou et al. "Adaptive online replanning with diffusion models". NeurIPS 2024.
>
> (_Continued in next piece of response_)

---

> ### Author Response · Authors · 2024-11-28
> **Official Response to Reviewer pd6k: Deterministic vs stochastic (2)**
>
> > **Comment**: We thank the Authors for their careful response. However, we remain confused by this manuscript, and particularly by the control problem it claims to address. How does noise enter the dynamics G? The system dynamics do not show any noise and it's unclear how this happens. What are the property of the noise process? It is difficult to accept that the proposed method can solve any optimal control problem for any kind of deterministic or stochastic dynamical system with any kind of noise. Does the paper offer guarantees on the optimality of the resulting control policy? Are the dynamics G stable and is the resulting closed-loop system stable? Since the control input is itself stochastic (sampled from a distribution conditioned on the current stat), how is stability and performance measured and guaranteed?
>
> **Answer:** (_Continued from previous piece of response_)
>
> For the question "Since the control input is itself stochastic (sampled from a distribution conditioned on the current state), how is stability and performance measured and guaranteed?", **our answer** is: although the control input sampled by the diffusion models is stochastic, it does not affect the control performance as the guided sampling optimizes the control objective. Under ideal assumptions, the sampled control signals can be approximately optimal at an arbitrarily high probability. Please refer to the next response for details.
>
>
> Regarding the comment that "It is difficult to accept that the proposed method can solve any optimal control problem for any kind of deterministic or stochastic dynamical system with any kind of noise," we would like to offer the following clarifications.
>
> Lots of diffusion-based control methods have been proposed and show outstanding performance in control problems like robotic control and flow control [2-5], although **they did not claim they can solve any optimal control problem for any kind of system with any kind of noise**. Our proposed method moves forward and overcomes the limitation of previous diffusion-based control methods: the very long process of denoising during sampling, which is currently a major obstacle to the wider adoption of such control methods (as pointed out by the **Reviewer sLTC**). Through extensive experiments, we have observed consistent and impressive control performance improvements. Moreover, we **present a guarantee under ideal assumptions for the deterministic system**; please refer to the next response for details. Therefore, although we do not claim to meet the reviewer's high standard of solving all possible optimal control problems under all conditions, we firmly believe that our method offers a valuable contribution to existing control methods.
>
> We hope this explanation helps to address the reviewer's concern and provides a more comprehensive understanding of the capabilities of our proposed approach.
>
> [2] Siyuan Zhou et al. "Adaptive online replanning with diffusion models". NeurIPS 2024.
>
> [3] Cheng Chi, et al. "Diffusion policy: Visuomotor policy learning via action diffusion." The International Journal of Robotics Research (2023).
>
> [4] Long Wei et al. "DiffPhyCon: A Generative Approach to Control Complex Physical Systems". NeurIPS 2024.
>
> [5] Anurag Ajay et al. "Is Conditional Generative Modeling all you need for Decision Making?". ICLR 2023.

---

> ### Author Response · Authors · 2024-11-28
> **Official Response to Reviewer pd6k: A primary theoretical guarantee about the control performance (1)**
>
> > **Comment**: Would it be possible to obtain guarantees by considering simpler, possibly linear, system dynamics? I'm unsure what the role of PDE dynamics is, since the paper doesn't enter into any details regarding the system under consideration. Using a simpler setting would help to clarify the proposed control method, which is otherwise obscured by the complexities of the setting that are tangential to the proposed method.
>
> **Answer:** As we mentioned in the previous response, presenting a guarantee for the practical guided sampling is very challenging and beyond the scope of our work.
> Therefore, we will make some ideal assumptions about the guided sampling here, only focusing on the insightful part.
> We list these **ideal assumptions** as follows:
> 1. We assume that we can get the ideal classifier mentioned in the paper [1];
> 2. We don't consider the practical discretization of the SDE in the sampling;
> 3. We assume the trained diffusion models have learned the exact distribution of the system.
>
> In our practical algorithm, we use DPS[2] to estimate the classifier in the guidance, and the loss function is the greedy version.
> All the effort here is to illuminate how our method works in the very ideal theoretical setting. We use **Lemma 1** to show that the ideal guided sampling can get the approximately optimal sample. And we use another **Lemma 2** to show that the sampled sequence will reflect the true feedback states in the system when the system is deterministic. So at least in the ideal setting, our method has the performance guarantee when the system is **deterministic**.
>
> **Lemma 1** By guided sampling, $\forall \varepsilon > 0, \forall \delta > 0$, we can get the approximately optimal sample $z_{1:N}(0)$, s.t., $J(u_0, z_{1:N}(0)) - J_\min \leq \varepsilon$ with probability $1 - \delta$.
>
> _Proof._
> Given the initial state $u_0$, define an auxiliary random variable $y \in \\{0, 1\\}$, by $p(y=1|z_{1:N}(0)) =  \exp\\{- \lambda (J(u_0, z_{1:N}(0)) - J_\min)\\}$.
> According to [1], if we use guided sampling, we can get $z_{1:N}(0) \sim p(z_{1:N}(0)|y=1)$.
> We consider a target set $Q(\varepsilon) = \\{ z_{1:N}(0) | J(u_0, z_{1:N}(0)) - J_\min \leq \varepsilon \\}$.
> Using the guided sampling, we will concentrate the probability into the target set. Specifically,
>
>
> \begin{equation}
> \begin{aligned}
> P(z_{1:N}(0) \in Q(\varepsilon)) &= \int_{Q(\varepsilon)} p(z_{1:N}(0)|y=1) {\rm d} z_{1:N}(0) \\\\
> &= \int_{Q(\varepsilon)} \frac{p(y=1|z_{1:N}(0))}{p(y=1)} p(z_{1:N}(0)) {\rm d} z_{1:N}(0) \\\\
> &= \frac{E_{z_{1:N}(0) \sim p(z_{1:N}(0)|u_0)} [1_{Q(\varepsilon)} p(y=1|z_{1:N}(0))]}{E_{z_{1:N}(0) \sim p(z_{1:N}(0)|u_0)} [p(y=1|z_{1:N}(0))]} \\\\
> &= \frac{E_{z_{1:N}(0) \sim p(z_{1:N}(0)|u_0)} [1_{Q(\varepsilon)} \exp\\{- \lambda (J(u_0, z_{1:N}(0)) - J_\min)\\}]}{E_{z_{1:N}(0) \sim p(z_{1:N}(0)|u_0)} [\exp\\{- \lambda (J(u_0, z_{1:N}(0)) - J_\min)\\}]}
> \end{aligned}
> \end{equation}
>
> Note that $\lim_{\lambda \to \infty} \exp\\{- \lambda (J(u_0, z_{1:N}(0)) - J_\min) = 1_{Q(0)}$ and $1_{Q(\varepsilon)} 1_{Q(0)} = 1_{Q(0)}$ for any $\varepsilon > 0$. So, we have
>
> \begin{equation}
> \begin{aligned}
> P(z_{1:N}(0) \in Q(\varepsilon))
> &= \frac{E_{z_{1:N}(0) \sim p(z_{1:N}(0)|u_0)} [1_{Q(\varepsilon)} \exp\\{- \lambda (J(u_0, z_{1:N}(0)) - J_\min)\\}]}{E_{z_{1:N}(0) \sim p(z_{1:N}(0)|u_0)} [\exp\\{- \lambda (J(u_0, z_{1:N}(0)) - J_\min)\\}]} \\\\
> &\to \frac{E_{z_{1:N}(0) \sim p(z_{1:N}(0)|u_0)} [1_{Q(\varepsilon)} 1_{Q(0)}]}{E_{z_{1:N}(0) \sim p(z_{1:N}(0)|u_0)} [1_{Q(0)}]}
> = 1,
> \end{aligned}
> \end{equation}
> as $\lambda\to\infty$.
> For any $\varepsilon > 0$, we can make the successful probability arbitrarily close to $1$ by setting $\lambda$ large enough.
> We complete the proof.
>
> [1] Yang Song et al. "Score-Based Generative Modeling through Stochastic Differential Equations", ICRL 2021.
>
> [2] Hyungjin Chung et al. "Diffusion Posterior Sampling for General Noisy Inverse Problems", ICLR 2023.Just combine the Lemma 1 and Lemma 2.
>
> (_Continued in next piece of response_)

---

> ### Author Response · Authors · 2024-11-28
> **Official Response to Reviewer pd6k: A primary theoretical guarantee about the control performance (2)**
>
> > **Comment**: Would it be possible to obtain guarantees by considering simpler, possibly linear, system dynamics? I'm unsure what the role of PDE dynamics is, since the paper doesn't enter into any details regarding the system under consideration. Using a simpler setting would help to clarify the proposed control method, which is otherwise obscured by the complexities of the setting that are tangential to the proposed method.
>
> **Answer**: (_Continued from previous piece of response_)
>
> **Lemma 2** If the dynamics $G$ is deterministic, when following the control sequence $z_{1:N} = (w_1(0), u_1(0), \cdots, w_N(0), u_N(0))$ sampled from the synchronous and asynchronous diffusion models, we get the states $(u_{\text{env}, 1}, u_{\text{env}, 2}, \cdots, u_{\text{env}, N})$ satisfying $u_{\text{env}, \tau} = u_\tau (0)$ for every $\tau$.
>
> _Prove by contradiction._
> If we get another sequence $u_{\text{env}, 1:N} \neq u_{1:N}$, let $k$ be the first index that $u_{\text{env}, k} \neq u_k(0)$ and $u_{\text{env}, i} \neq u_i(0)$ for $i < k$.
> According to our paper, the synchronous and asynchronous diffusion models can generate samples from the truthful distribution of the system.
> Since we get the sample $z_{1:N}$, we know that the system distribution allows $p(u_k(0)|u_{k-1}(0), w_k(0)) > 0$.
> Since we get the states $u_{\text{env}, 1:N}$ from the system using $w_{1:N}$, we know that the system distribution also allows $p(u_{\text{env}, k}|u_{\text{env}, k-1}, w_k(0)) > 0$.
> But  $u_{\text{env}, k} \neq u_{k}(0)$ and $u_{\text{env}, k-1} = u_{k-1}(0)$, which is contradictary with $G$ is deterministic.
> We complete the proof.
>
> **Main conclusion** If the dynamics $G$ is deterministic, by guided sampling, we can get the approximately optimal control sequence and states.
>
> _Proof._
> Just combine the Lemma 1 and Lemma 2.

---

> ### Author Response · Authors · 2024-11-28
> **Official Response to Reviewer pd6k: Presentation of the paper**
>
> > **Comment**: I respect the Authors' opinion of the presentation of their work, as well as their efforts in clarifying the concerns raised by the reviewers. However, I still find the paper very hard to read and appreciate. Perhaps the paper is tailored to experts in the field and those that already know the related and cited works, but I strongly doubt that anyone else can appreciate this contribution.
>
> **Answer:** We understand your concern about the presentation. However, as we clarified previously, the presentation of our method is primarily based on an understanding of diffusion models. Given that diffusion models have become one of the most popular methods for AI-generated content (AIGC), they are increasingly becoming common knowledge in both academic and industrial AI communities. Beyond their successful applications in image [1], video [2], and text [3] generation, diffusion models are also experiencing huge prosperity in fields such as weather forecasting [4], molecule and material design [5], drug discovery [6], and decision-making tasks like motion planning [7] and robot control [8]. Many researchers from diverse backgrounds, including physics, biology, chemistry, materials science, medicine, robotics, and control engineering, are now familiar with this tool. Therefore, our work is aimed at a broad audience with a background in diffusion models, rather than being limited to a small number of experts.
>
> Even in the fields of control and robotics, hundreds of papers on diffusion model-based control have been published in recent years, making it a highly active and rapidly advancing research area. Diffusion models have a unique advantage in solving high-dimensional problems due to their strong generalization capabilities. As a top conference in AI and related fields, ICLR aims at publishing cutting-edge and pioneering research, especially innovative studies that differ significantly from traditional approaches. Such work is more likely to lead to disruptive technological breakthroughs.
>
> [1] William Peebles et al. "Scalable diffusion models with transformers". CVPR 2023.
>
> [2] Duygu Ceylan et al. "Pix2video: Video editing using image diffusion". CVPR 2023.
>
> [3] Tong Wu et al. "Ar-diffusion: Auto-regressive diffusion model for text generation". NeurIPS 2023.
>
> [4] Lei Chen et al. "Swinrdm: integrate swinrnn with diffusion model towards high-resolution and high-quality weather forecasting". AAAI 2023.
>
> [5] Jan-Hendrik Bastek et al. "Inverse design of nonlinear mechanical metamaterials via video denoising diffusion models". Nature Machine Intelligence (2023).
>
> [6] Denise B. Catacutan et al. "Machine learning in preclinical drug discovery". Nature Chemical Biology (2024).
>
> [7] Anurag Ajay et al. "Is Conditional Generative Modeling all you need for Decision Making?". ICLR 2023.
>
> [8] Siyuan Zhou et al. "Adaptive online replanning with diffusion models". NeurIPS 2024.

---

### Official Review · Reviewer_sLTC · 2024-11-01

**Soundness:** 4
**Presentation:** 4
**Contribution:** 4
**Rating:** 10
**Confidence:** 4

**Summary:**

The paper proposes a novel method for using diffusion policies for closed loop control that speeds up the generation of a control sequence of a length up to a chosen horizon by avoiding full denoising of the latter steps in the sequence, as they will likely not be executed by a receding horizon controller. The proposed method demonstrates superior performance in terms of achieving control objectives, due to closed loop control using a diffusion policy, with computation time much shorter than other diffusion policy baseline controllers.

**Strengths:**

The main strength of the proposed method is the idea to save computation by not fully denoising the latter steps of the control sequence, with the understanding that in receding horizon control, the controls for those latter stages will likely not be used, as new state observations will appear in the meantime and the control sequence will be replanned. This idea addresses one of the main shortcomings of controllers based on diffusion policies - the very long process of denoising during sampling, which is currently a major obstacle to the wider adoption of diffusion policies that have shown considerable advantages with respect to more traditional behavior cloning schemes.

**Weaknesses:**

I believe the proposed method is excellent and clearly applicable to very difficult control problems involving PDEs. However, most control problems are not of this kind. Why not show the performance on a simpler control problem with a handful of variables, hopefully not requiring high-end GPUs? This could potentially increase the applicability and appeal of the method significantly.

Some minor typos:

P. 3, L. 112: "It" -> "They"
P. 3, L. 140: "has" -> "have"
P. 7, L. 376: "well well" -> "perform well"?
P. 10, L. 539: "done" -> "drone"?
P. 19, L. 1009: "128 grids" -> "128 cells"

**Questions:**

I am not quite sure why the proposed method achieves a better control objective than DiffPhyCon-1 - isn't the latter replanning at each control step by means of full denoising, thus as closed-loop as the proposed method? Can the authors clarify this?

The planning horizon for the two control tasks is fixed. Why exactly to these lengths? What happens if full denoising is done over shorter horizons, replanning at each time step, using DiffPhyCon-1?

---

> ### Author Response · Authors · 2024-11-23
> **Official Response to Reviewer sLTC (1)**
>
> We appreciate the high recognition of the reviewer for our work. We are grateful that the reviewer acknowledges our method's significant contribution to addressing the low control efficiency of diffusion control methods. Additionally, we value the insight that our approach can be applied to highly challenging control tasks. Below, we address the reviewer's questions one by one.
>
> > Comment 1: I believe the proposed method is excellent and clearly applicable to very difficult control problems involving PDEs. However, most control problems are not of this kind. Why not show the performance on a simpler control problem with a handful of variables, hopefully not requiring high-end GPUs? This could potentially increase the applicability and appeal of the method significantly.
>
> **Answer:** Thank you for your positive feedback and recognition of our work!
>
> To demonstrate the applicability of our method to lower-dimensional tasks, we performed a **new experiment** on the control of the **inverted double pendulum on a cart**. The system consists of two point masses fixed on the end of two massless rigid rods, which are mutually connected and connected to a cart with frictionless hinges. The objective is to control the double pendulum from falling down by exerting a force on the cart. A detailed description and visual illustration of the problem setting can be found in **Appendix F.1 and F.2**.
>
> We would like to note that this system consists of 6 state variables and 1 action variable, which can be more efficiently simulated. In our implementation, the **environment can be simulated with a CPU in real-time**. Although the control signal and system states are relatively low-dimensional, we note the control task is non-trivial. Since no torque can be exerted on the hinges, the system can exhibit limited controllability. This system has been studied in the control research community [1,2].
>
> We evaluated the performance of our CL-DiffPhyCon under 3 different levels of random perturbation on the system state and compared the performance with different baselines. Results averaged over 5 seeded runs are reported to enhance soundness. The results of this experiment clearly demonstrate the effectiveness of our CL-DiffPhyCon in controlling this task with improved control efficiency, as can be found in **Appendix F.3**.
>
> [1] Michael Hesse et al. "A reinforcement learning strategy for the swing-up of the double pendulum on a cart." Procedia Manufacturing (2018).
>
> [2] Masaki Yamakita et al. "Swing up control of a double pendulum." ACC 1993.
>
> > Comment 2: Some minor typos: P. 3, L. 112: "It" -> "They" P. 3, L. 140: "has" -> "have" P. 7, L. 376: "well well" -> "perform well"? P. 10, L. 539: "done" -> "drone"? P. 19, L. 1009: "128 grids" -> "128 cells".
>
> **Answer :** Thanks. All these typos have been fixed in the revised manuscript.

---

> > ### Author Response · Authors · 2024-11-23
> > **Official Response to Reviewer sLTC (2)**
> >
> > > Comment 3: I am not quite sure why the proposed method achieves a better control objective than DiffPhyCon-1 - isn't the latter replanning at each control step by means of full denoising, thus as closed-loop as the proposed method? Can the authors clarify this?
> >
> > **Answer:** Thanks for your question. As explained in **Lines 48 to 49 of our original submission**, DiffPhyCon-1 replans at every physical time step using full denoising, which may disrupt the **consistency** of control signals—particularly due to the stochastic nature of diffusion models—thereby impacting overall performance. This issue has been acknowledged in previous studies [1, 2]. In contrast, at each physical time step, our method retains $(H-1)/H * T$ sampling steps accumulated from the previous $(H-1)$ physical time steps and only performs the final $1/H * T$ sampling steps based on fresh feedback from the environment. This approach achieves a better balance between incorporating new observations and maintaining control consistency.
> >
> > [1] Leslie Pack Kaelbling et al. "Hierarchical task and motion planning in the now." ICRA 2011.
> >
> > [2] Siyuan Zhou et al. "Adaptive online replanning with diffusion models". NeurIPS 2024.
> >
> > > Comment 4: The planning horizon for the two control tasks is fixed. Why exactly to these lengths? What happens if full denoising is done over shorter horizons, replanning at each time step, using DiffPhyCon-1?
> >
> > **Answer:** Thanks for your comment. Our method does not rely on a fixed planning horizon; the choice of horizon H is determined by balancing efficiency and effectiveness. To address the reviewer's concern, we conducted additional experiments on the more challenging 2D task with $H=6$ and  $H=10$. The results are shown in the following Table 6. Inference time using a single NVIDIA A6000 48GB GPU with 16 CPU cores is also reported.
> >
> > These results indicate that both our method and DiffPhyCon-1 yield similar performance when $H=10$, compared to  $H=15$, as reported in Table 2 (Fixed map column) of the original submission. However, performance significantly deteriorates when  $H$ decreases to 6, which is likely due to the shorter observation window leading to inaccurate future control objective $\mathcal{J}$ estimation and suboptimal guidance sampling (Line 5 in Algorithm 1 in the original submission). On the other hand, increasing $H$ beyond 15 significantly raises GPU memory costs during inference, thus not recommended. Across different values of  $H$ (i.e.,  $H=6,10$ and $15$), our method consistently outperforms DiffPhyCon-1. Therefore, for this task, a horizon between 10 and 15 is appropriate.
> >
> > In practical applications, the optimal $H$ can be determined through multiple trials to balance performance and efficiency, similar to the approach used in diffusion policies [1] (see Figure 5 (left) in the referenced paper). **These results and discussions have been added to Appendix E.3 of our revised manuscript.**
> >
> > **Table 6:** Effect of the model horizon H on 2D incompressible fluid control under the large domain control setting and the fixed map evaluation mode.
> >
> > |                      | $H=6, \mathcal{J}$ | $H=6$, time (s) | $H=10, \mathcal{J}$ | $H=10$, time (s) | $H=15, \mathcal{J}$ | $H=15$, time (s) |
> > |----------------------|--------------------|-----------------|---------------------|------------------|---------------------|------------------|
> > | DiffPhyCon-1         | 0.8986             | 1022            | 0.5140              | 1216             | 0.5454              | 1677             |
> > | **CL-DiffPhyCon (ours)** | **0.6012**             | **143.32**          | **0.3367**              | **136.81**           | **0.3371**              | **140.83**           |
> >
> > [1] Cheng Chi, et al. "Diffusion policy: Visuomotor policy learning via action diffusion." The International Journal of Robotics Research (2023).

---

> > > ### Author Response · Authors · 2024-11-25
> > > **A gentle reminder: please respond to our rebuttal**
> > >
> > > Dear Reviewer sLTC,
> > >
> > > Thank you for reviewing our work. We have carefully responded to all your comments and questions in our rebuttal.
> > >
> > > The deadline for the author-reviewer discussion is coming up soon. Could you please take a moment to review our responses? Any additional comments or feedback would be appreciated.
> > >
> > > Thanks a lot!
> > >
> > > Best regards,
> > >
> > > Authors

---

> > > > ### Comment · Reviewer_sLTC · 2024-11-25
> > > > **Responses reviewed, rating maintained**
> > > >
> > > > I have reviewed the authors' rebuttal and found it to be completely satisfactory. I maintain my rating and recommend acceptance.

---

> > > > > ### Author Response · Authors · 2024-11-25
> > > > > **Thanks for your recognition and recommendation for our work**
> > > > >
> > > > > Thank you for your recognition and recommendation for our work. We are pleased that our response has successfully addressed your suggestions and satisfied your expectations.

---

> > > ### Comment · Reviewer_sLTC · 2024-11-25
> > > **Clarification and investigation of horizon length**
> > >
> > > Thank you for clarifying the advantages of your method in terms of maintaining the consistency of the sequence of controls. Furthermore, the empirical investigation of the effect of different planning horizons illustrates that it is indeed important as regards control performance. I understand that the optimal planning horizon has to be selected manually to balance performance and computational cost, but because this is similar to other uses of diffusion policies for control, and optimizing the planning horizon is not a direct objective of this paper, this procedure should be acceptable in practical use.

---

> > ### Comment · Reviewer_sLTC · 2024-11-25
> > **Suggestion for additional low-dimensional control problems addressed**
> >
> > Thank you for addressing my suggestion for testing your method on additional low-dimensional control problems. Your choice to test the method on a double inverted pendulum on a cart is very suitable and illustrates well the advantages of your method.

---

### Official Review · Reviewer_GNga · 2024-11-02

**Soundness:** 3
**Presentation:** 3
**Contribution:** 3
**Rating:** 8
**Confidence:** 3

**Summary:**

This manuscript addresses a closed-loop diffusion control problem of complex physical systems. The proposed CL-DiffPhyCon (Wei et al., 2024) extends the work of DiffPhyCon (Wei et al., 2024).

**Strengths:**

One advantage is to improve the computation efficiency, which is achieved by an asynchronous denoising framework. This aspect is validated using numerical simulations. Overall, this manuscript is well-written and organized.

**Weaknesses:**

There are some technical limitations, including noise-free inputs, input constraints, and numerical validations by using full in-domain control and observation.

**Questions:**

Detailed comments are summarized below:

1. How restrictive is the consideration of noise-free control? There are many noise issues arising from complex engineering systems and applications. Can the proposed method be extended to account for plant, control, and measurement noise?

2. Another important aspect associated with complex and practical physical systems is physical constraints. For example, physical control actions are implemented via actuators which often have upper and lower limits. That is one major reason model predictive control is useful. Can the proposed method handle physical constraints associated with control actions?

3.  From Figure 2 and the limitation mentioned in the appendix, the algorithm is trained offline. In that case, it is not clear how the control is designed and employed in a real environment, as controllers are typically designed and used in practice in an interactive manner. In Figure 2, the ending time instance is T, can it be changed to T+1, as a bigger loop, to make it useful in an online/interactive manner?

4. In both examples, the controls are defined as a spatiotemporal function in the PDE model, which is too ideal to be implemented in a realistic system. For PDE system control, typically, the control is implemented as a temporal function. The corresponding actuator is located in the boundary or a small part of the entire spatial domain. It would be interesting to see the results in these more realistic settings by using the proposed method.

5. In the Burgers’ equation example, there is no difference between the full observation and partial observation cases, because the latter case only observes and controls two parts of the entire domain. In the case considered in the manuscript, one can simply treat each observed sub-domain as a total entire domain and apply the algorithm on the PDE with new spatial coordinates/ranges/boundary conditions. Partial observation typically refers to the case using partial observation to control the entire domain. The control performance results of the unobserved state should be added to fully demonstrate the effectiveness of the proposed algorithm. Similarly. practical applications often have sensors installed in the boundary of PDE systems. It would be good to also add results by using boundary observation data.

6. The limitations of the proposed method should be moved to the conclusion section for better readability.

---

> ### Author Response · Authors · 2024-11-23
> **Official Response to Reviewer GNga (1)**
>
> We thank the reviewer for the valuable and detailed feedback. We are glad that the reviewer recognizes the clarity and experimental strengths of our work. Below, we address the reviewer's questions one by one.
>
> >Comment 1: How restrictive is the consideration of noise-free control? There are many noise issues arising from complex engineering systems and applications. Can the proposed method be extended to account for plant, control, and measurement noise?
>
> **Answer:** Thanks for your question.
>
> - **Firstly,** in the 2D incompressible fluid control task in the original submission, we have added **noise in the execution of control**, following previous work [1]. Specifically, as detailed in **Section 5.3 of the original submission**, for each trajectory of length $N = 64$, the control signal in each physical time step is executed with probability $p = 0.1$ as a random noise, where the horizontal and vertical components following the uniform distributions bounded by the minimal and maximal values of horizontal and vertical control signals in the training dataset, respectively. The results reported in **Table 2 of the original submission** demonstrated the superiority and robustness of our approach against control noise.
>
>     To further evaluate the control performance with respect to control noise, we conducted **new experiments** on a new **boundary control** setting, where control signals are restricted to only 8 cells for each of the 4 side exits of the $64\times 64$ fluid field. Consistent with our original setting (renamed as **"large domain control"** in the revised manuscript), the control signal in each physical time step is still executed with probability $p = 0.1$ as a random noise. Evaluations of different methods are conducted on both fixed map and random map modes. The results are shown in the following Table 1. The average inference time of two evaluation modes on a single NVIDIA A6000 48GB GPU with 16 CPU cores is also reported.
>
>
> **Table 1:** Comparison results on 2D incompressible fluid control under control noise with probability 0.1 in the additional **boundary control** setting.
>   |                            | Fixed map | Random map | Average time (s) |
> |----------------------------|-----------|------------|------------------|
> | BC                         | 0.8861    | 0.8871     | 4.72             |
> | BPPO                       | 0.8830    | 0.8844     | **4.66**             |
> | DiffPhyCon-1               | 0.7517    | 0.7955     | 1648.52          |
> | DiffPhyCon-5               | 0.6703    | 0.7451     | 343.43           |
> | DiffPhyCon-14              | 0.6498    | 0.7221     | 139.00           |
> | RDM                        | 0.6553    | 0.7087     | 190.06           |
> | **CL-DiffPhyCon (ours)**       | **0.6169**    | **0.7003**     | 147.40           |
> | **CL-DiffPhyCon (DDIM, ours)** | 0.6671    | 0.7109     | 30.63            |
>
>
> These results indicate that even under the perturbation of control noise, our method still achieves acceptable control performance under this more challenging boundary control setting with a much smaller number of actuators. Compared with baselines, our CL-DiffPhyCon achieves the best control performance. The fast sampling DDIM method could still be applied to our method for significant acceleration of control.
>
> **We have added the above results and analysis in Section 5.3 of the revised manuscript, with further details presented in Appendix E.**
>
> [1] Siyuan Zhou et al. "Adaptive online replanning with diffusion models". NeurIPS 2024.
>
> *(Continued in next piece of response)*

---

> > ### Author Response · Authors · 2024-11-23
> > **Official Response to Reviewer GNga (2)**
> >
> > >Comment 1: How restrictive is the consideration of noise-free control? There are many noise issues arising from complex engineering systems and applications. Can the proposed method be extended to account for plant, control, and measurement noise?
> >
> > **Answer:** *(Continued from previous piece of response)*
> >
> > - **Secondly,** to account for **system and measurement noise**, we conducted **new experiments** on the 1D control problem to verify the performance of our proposed method. The system and measurement are perturbed by Gaussian noise, with the standard deviation $\sigma=0.025$ following the work [2]. The results are shown in the following Table 2, where the average time means the average computational time (on a single NVIDIA A100 80GB GPU with 16 CPU cores) over three settings in inference. The results imply the following conclusions:
> >
> >   - The results with system noise are comparable to the results of noise-free and the proposed CL-DiffPhyCon has a significant improvement compared with baselines.
> >   - Although the measurement noise results in a slight decline in all diffusion-based control methods, CL-DiffPhyCon still has superior performance.
> >   - Therefore, in these scenarios with noise, our conclusion is consistent with the original submission.
> >
> >   **We have added the above results in Table 1, Section 5.2 of the revised manuscript.**
> >
> > **Table 2:** Comparison results on 1D Burgers' equation control under (new) system and measurement noise and (original) noise-free settings.
> > |                            | Noise-free | System noise | Measurement noise | Average time (s) |
> > |----------------------------|------------|--------------|-------------------|------------------|
> > | BC                         | 0.4708     | 0.4138       | 0.3981            | 0.8053           |
> > | BPPO                       | 0.4686     | 0.4088       | 0.3979            | 0.8193           |
> > | PID                        | 0.3250     | 0.2323       | 0.2911            | **0.7433**           |
> > | DiffPhyCon-1               | 0.0210     | 0.0222       | 0.0232            | 39.0152          |
> > | DiffPhyCon-5               | 0.0252     | 0.0271       | 0.0272            | 8.8100           |
> > | DiffPhyCon-15              | 0.0361     | 0.0382       | 0.0377            | 4.1949           |
> > | RDM                        | 0.0296     | 0.0310       | 0.0336            | 6.6567           |
> > | **CL-DiffPhyCon (ours)**       | **0.0096**     | **0.0095**       | **0.0127**            | 3.7224           |
> > | **CL-DiffPhyCon (DDIM, ours)** | 0.0112     | 0.0114       | 0.0146            | 0.8350           |
> >
> > [2] Cagatay Yildiz et al. "Continuous-time model-based reinforcement learning." ICML 2021.

---

> > > ### Author Response · Authors · 2024-11-23
> > > **Official Response to Reviewer GNga (3)**
> > >
> > > > Comment 2: Another important aspect associated with complex and practical physical systems is physical constraints. For example, physical control actions are implemented via actuators which often have upper and lower limits. That is one major reason model predictive control is useful. Can the proposed method handle physical constraints associated with control actions?
> > >
> > > **Answer:** Thanks for your question. Control actions sampled by the diffusion model are **within the range of training dataset generally**, because the diffusion model **learns the distribution of the training dataset**. The training dataset should satisfy physical constraints, as these training datasets are collected in physical systems.
> > >
> > >   Moreover, **we can also clip the actions using the upper and lower bounds of physical constraints**. We conducted **new experiments** to verify the performance of our method under physical constraints. We set the lower and upper bounds to -2 and 2, less than the unconstrained actions ranging approximately from -5 to 5 (minimum and maximum values of unconstrained actions generated by all methods). From the following Table 3, we can see that the performance is slightly worse, but our proposed CL-DiffPhyCon is still superior. Average inference time is evaluated using a single NVIDIA A100 80GB GPU with 16 CPU cores.
> > >
> > > **We have added the results and analysis to Section 5.2 of the revised manuscript.**
> > >
> > > **Table 3:** Comparison results on 1D Burgers' equation control under (new) physical constraint and original settings.
> > >
> > > |                            | original results | physical constraint | Average time (s) |
> > > |----------------------------|------------------|---------------------|------------------|
> > > | BC                         | 0.4708           | 0.4704              | 0.8352           |
> > > | BPPO                       | 0.4686           | 0.4507              | 0.8184           |
> > > | PID                        | 0.3250           | 0.3585              | **0.7288**           |
> > > | DiffPhyCOn-1               | 0.0210           | 0.0214              | 38.6665          |
> > > | DiffPhyCon-5               | 0.0252           | 0.0257              | 8.6050           |
> > > | DiffPhyCon-15              | 0.0361           | 0.0365              | 3.9989           |
> > > | RDM                        | 0.0296           | 0.0296              | 6.5646           |
> > > | **CL-DiffPhyCon (ours)**       | **0.0096**           | **0.0110**              | 3.6239           |
> > > | **CL-DiffPhyCon (DDIM, ours)** | 0.0112           | 0.0123              | 0.7957           |
> > >
> > > > Comment 3: From Figure 2 and the limitation mentioned in the appendix, the algorithm is trained offline. In that case, it is not clear how the control is designed and employed in a real environment, as controllers are typically designed and used in practice in an interactive manner. In Figure 2, the ending time instance is T, can it be changed to T+1, as a bigger loop, to make it useful in an online/interactive manner?
> > >
> > > **Answer:**   Thank you for your comment. We would like to clarify that **our model is employed in the real environment in an interactive manner during inference**, which is a core contribution of our method: it enables closed-loop feedback control with asynchronous sampling in diffusion-based control.
> > >
> > >   Our model is trained offline, similar to an offline reinforcement learning setting. Specifically, training data is collected beforehand, and the model does not interact with the environment during training. During inference and evaluation, the model interacts with the environment, generating control actions based on real-time feedback. Both our method and the baseline methods in the manuscript follow this setup.
> > >
> > >   Regarding Figure 2, please note that $T$ represents the total number of diffusion steps, not the total number of physical time steps. The physical time step index is $\tau$, with a maximum value denoted as $N$, which is assumed to be much larger than the diffusion model horizon $H$, **as stated at the beginning of Section 4.1 of the original submission**. Figure 2 illustrates the interactive control process: at each physical time step $\tau$, the model generates the control action $w_{\tau}$ based on the feedback state $u_{\tau-1}$. This process is executed in a loop over $\tau$, allowing the method to continuously adapt control actions based on real-time feedback from the evolving environment.

---

> ### Author Response · Authors · 2024-11-23
> **Official Response to Reviewer GNga (4)**
>
> > Comment 4: In both examples, the controls are defined as a spatiotemporal function in the PDE model, which is too ideal to be implemented in a realistic system. For PDE system control, typically, the control is implemented as a temporal function. The corresponding actuator is located in the boundary or a small part of the entire spatial domain. It would be interesting to see the results in these more realistic settings by using the proposed method.
>
> **Answer:** Thanks for your question. The settings of our experiments follow previous works [1,2], where the control actions are the spatiotemporal functions. According to your suggestions, we have conducted new experiments when control actuators are located in a small part of the spatial domain, both in 1D and 2D experiments.
>
> - **Firstly**, in 1D experiments, we reduce the dimension of control, and actuators are evenly placed in 1/8 spatial domain (16 actuators are placed to partially control)(FOPC). **The illustration of placement is shown in Appendix D.2 of the revised manuscript**. The results are evaluated on the entire spatial domain, which is shown in the following Table 4. Average inference time is evaluated using a single NVIDIA A100 80GB GPU with 16 CPU cores.
>
>     We can see that the reduced controllable range generally diminishes the effectiveness of diffusion-based control methods, yet our method consistently performs better than the baselines. BC, BPPO, and PID demonstrate improved performance because only a small part of actions need to be optimized, making it easier to fit compared to high-dimensional actions, while they still show significantly lower performance compared to diffusion-based methods. **We have added the results and analysis in Section 5.2 of the revised manuscript**. We also considered partial observation (POPC). (As a common SISO controller, the PID is difficult to directly apply to MIMO systems. Its application to the MIMO system often requires additional decoupling and planning modules. Thus, we only apply the PID in 16 sensors for observation and 16 actuators for control (POPC).)
>
> **Table 4:** Comparison results on 1D Burgers' equation control under FOPC and POPC settings.
>
> |                            | FOPC | POPC | Average time (s) |
> |----------------------------|----------------------------------|-------------------------------------|------------------|
> | BC                         | 0.1093                           | 0.0987                              | 0.8257           |
> | BPPO                       | 0.1079                           | 0.0984                              | 0.7982           |
> | PID                        | -                                | 0.0827                              | **0.7334**           |
> | DiffPhyCOn-1               | 0.0330                           | 0.0332                              | 69.1113          |
> | DiffPhyCon-5               | 0.0482                           | 0.0484                              | 14.7506          |
> | DiffPhyCon-15              | 0.1128                           | 0.1132                              | 6.5274           |
> | RDM                        | 0.1124                           | 0.1121                              | 7.4914           |
> | **CL-DiffPhyCon (ours)**       | **0.0291**                           | **0.0295**                              | 6.1809           |
> | **CL-DiffPhyCon (DDIM, ours)** | 0.0311                           | 0.0313                              | 0.8080           |
>
> [1] Rakhoon Hwang et al. "Solving pde-constrained control problems using operator learning". AAAI 2022.
>
> [2] Philipp Holl et al. "Learning to Control PDEs with Differentiable Physics." ICLR 2020.
>
> *(Continued in next piece of response)*

---

> > ### Author Response · Authors · 2024-11-23
> > **Official Response to Reviewer GNga (5)**
> >
> > > Comment 4: In both examples, the controls are defined as a spatiotemporal function in the PDE model, which is too ideal to be implemented in a realistic system. For PDE system control, typically, the control is implemented as a temporal function. The corresponding actuator is located in the boundary or a small part of the entire spatial domain. It would be interesting to see the results in these more realistic settings by using the proposed method.
> >
> > **Answer:** *(Continued from previous piece of response)*
> >
> > - **Secondly,** in 2D incompressible fluid control task, we conducted new experiments of **boundary control**, where control signals are restricted to only 8 cells for each of the 4 side exits of the $64\times64$ fluid field. Thus the total number of control actuators is $8\times4=32$. We generate 30,000 additional training trajectories to train two diffusion models of this setting. Our original setting is renamed as **large domain control** with 1,792 actuators. The boundary control setting is more challenging compared to the large domain control setting due to the significantly reduced number of actuators. For both settings, evaluations are conducted on fixed map and random map modes, consistent with the original submission. Please refer to **Appendix E.1 of the revised manuscript for a detailed description of experimental settings**. The results of both large domain control and boundary control are shown in the following Table 5, where we copy the results of large domain control settings of Table 2 in our original submission for comparison. The average inference time using a single NVIDIA A6000 48GB GPU with 16 CPU cores shown in the last column is averaged over time costs of settings in the first four columns.
> >
> >     The results indicate that our CL-DiffPhyCon outperforms the baselines not only in the original large domain control setting but also in the new boundary control setting, under both fixed map and random map evaluation modes. By applying DDIM for fast sampling, our method is significantly accelerated at least 5x faster than strong diffusion control baselines RDM and DiffPhyCon-14, with comparable results.
> >
> >     **We have added the results and analysis to Section 5.3 of the revised manuscript, with further details presented in Appendix E.**
> >
> > **Table 5:** Comparison results on 2D incompressible fluid control control under (new) boundary control and (original) large domain control settings.
> >
> > |                            | Large domain control - fixed map | Large domain control - random map | Boundary control - fixed map | Boundary control - random map | Average time (s)
> > |----------------------------|----------------------------------|-----------------------------------|------------------------------|-------------------------------|-------------------------------|
> > | BC                         | 0.6722                           | 0.7046                            | 0.8861                       | 0.8871                        | 4.67 |
> > | BPPO                       | 0.6343                           | 0.6524                            | 0.8830                       | 0.8844                        | **4.65** |
> > | DiffPhyCon-1               | 0.5454                           | 0.3754                            | 0.7517                       | 0.7955                        | 1666.50 |
> > | DiffPhyCon-5               | 0.5051                           | 0.5458                            | 0.6703                       | 0.7451                        | 357.66 |
> > | DiffPhyCon-14              | 0.5823                           | 0.5621                            | 0.6498                       | 0.7221                        | 141.88 |
> > | RDM                        | 0.4074                           | 0.4356                            | 0.6553                       | 0.7087                        | 238.43 |
> > | **CL-DiffPhyCon (ours)**       | **0.3371**                           | **0.3485**                            | **0.6169**                       | **0.7003**                        | 144.04 |
> > | **CL-DiffPhyCon (DDIM, ours)** | 0.4100                           | 0.4254                            | 0.6671                       | 0.7109                        | 26.01 |
> >
> > *(Continued in next piece of response)*

---

> ### Author Response · Authors · 2024-11-23
> **Official Response to Reviewer GNga (6)**
>
> > Comment 4: In both examples, the controls are defined as a spatiotemporal function in the PDE model, which is too ideal to be implemented in a realistic system. For PDE system control, typically, the control is implemented as a temporal function. The corresponding actuator is located in the boundary or a small part of the entire spatial domain. It would be interesting to see the results in these more realistic settings by using the proposed method.
>
> **Answer:** *(Continued in next piece of response)*
>
> - **Thirdly,** we also conducted **new experiments** on a challenging control problem of the **inverted double pendulum on a cart, which is also a chaotic system**. The system consists of two point masses fixed on the end of two massless rigid rods, which are mutually connected and connected to a cart with frictionless hinges. The objective is to control the double pendulum from falling down. A detailed description and visual illustration of the problem setting can be found in **Appendix F.1 and F.2 of the revised manuscript**.
>
>     This setting is more realistic in terms of its partial controllability. The control is restricted to the cart, reflecting the spatially limited control in realistic systems. Besides, only the force exerted on the cart can be directly controlled instead of the velocity to reflect a more realistic control setting. This system has been studied in the control research community [3, 4].
>
>     We evaluated the performance of CL-DiffPhyCon by comparing its control performance to different diffusion control baselines under 3 different levels of random perturbation on the state. Results averaged over 5 seeded runs are reported to enhance soundness. They clearly demonstrate the effectiveness of our CL-DiffPhyCon in controlling this task with improved control efficiency, as can be found in **Appendix F.3 of the revised manuscript**.
>
> [3] Michael Hesse et al. "A reinforcement learning strategy for the swing-up of the double pendulum on a cart." Procedia Manufacturing (2018).
>
> [4] Masaki Yamakita et al. "Swing up control of a double pendulum." ACC 1993.
>
> > Comment 5: In the Burgers’ equation example, there is no difference between the full observation and partial observation cases, because the latter case only observes and controls two parts of the entire domain. In the case considered in the manuscript, one can simply treat each observed sub-domain as a total entire domain and apply the algorithm on the PDE with new spatial coordinates/ranges/boundary conditions. Partial observation typically refers to the case using partial observation to control the entire domain. The control performance results of the unobserved state should be added to fully demonstrate the effectiveness of the proposed algorithm. Similarly. practical applications often have sensors installed in the boundary of PDE systems. It would be good to also add results by using boundary observation data.
>
> **Answer:** Thanks for your question. We conducted **new experiments** to show the control capacity on partial observation. In this setting, sensors only measure 1/8 of the spatial domain (16 sensors are evenly placed to partially observe). The POPC results are shown in **Table 4 of the previous response (Official Response to Reviewer GNga (4))**. We can see that performance slightly decreases due to the reduction of observation, but our method still has a significant improvement over the baselines.
>
> **We have added the experiments in Table 1 of Section 5.2, of the revised manuscript.**
>
> > Comment 6: The limitations of the proposed method should be moved to the conclusion section for better readability.
>
> **Answer:** Thank you for the suggestion. **In the revised manuscript, we have moved the discussion of limitations from the appendix to Section 6 "Conclusion and Limitation".**

---

> > ### Author Response · Authors · 2024-11-25
> > **A gentle reminder: please respond to our rebuttal**
> >
> > Dear Reviewer GNga,
> >
> > Thank you for reviewing our work. We have carefully responded to all your comments and questions in our rebuttal.
> >
> > The deadline for the author-reviewer discussion is coming up soon. Could you please take a moment to review our responses? If you find them satisfactory, we hope you might consider adjusting your initial rating. Any additional comments or feedback would be appreciated.
> >
> > Thanks a lot!
> >
> > Best regards,
> >
> > Authors

---

> > > ### Comment · Reviewer_GNga · 2024-11-25
> > > **Response to the revised manuscript**
> > >
> > > The revised manuscript has addressed all my comments, so an acceptance is recommended.

---

> ### Author Response · Authors · 2024-11-26
> **Thanks for your recognition and recommendation for our work**
>
> Thank you for your recognition and recommendation for our work. We are pleased that our response has successfully addressed all your comments.
>
> Moreover, we gently remind you that in ICLR's scoring system, a score of 6 indicates a "marginally above the acceptance threshold", while a score of 8 represents "accept". If you believe our work merits acceptance, we hope you might consider giving us an "accept", as it would greatly support our work.

---

### Author Response · Authors · 2024-11-23
**General response**

We sincerely thank reviewers for their dedicated time and constructive comments. We are pleased that the reviewers found our work to be **novel** (Reviewer sLTC), **timely** (Reviewer pd6k), **well-written** (Reviewers GNga and sLTC), and **applicable to very difficult control problems** (Reviewer sLTC). Reviewers GNga and sLTC particularly pointed out that our proposed asynchronous denoising method **effectively addresses the computational efficiency challenge** in diffusion-based controllers, which is currently a major obstacle to the wider adoption of these methods.

Based on the reviewers' insightful comments, we have conducted new experiments and made several clarifications to address their concerns. In this revised manuscript, we have updated both the main text and appendices, with changes highlighted in blue. The major improvements and new experiments are summarized as follows:
- **Experiments on system, measurement, and control noise:** We add experiments to analyze the effect of system and measurement noise on the 1D task, and control noise on the 2D task, as suggested by Reviewer GNga. The results are added to Table 1 and Table 2 in our revised manuscript. The results further confirm the robustness and superiority of our method. For more details, see the responses to Reviewer GNga.
- **Experiments on physical constraints:** We add experiments on the effect of physical constraints on control actions, as suggested by Reviewer GNga. Our method still shows robust and steady performance even with limited control action ranges. The results are added to Table 1 in our revised manuscript. For more details, see the responses to Reviewer GNga.
- **Experiments on more realistic scenarios with partial/boundary control and new double pendulum task:** We add experiments of more realistic scenarios where control actions are restricted to boundaries or a small part of the domain, as concerned by Reviewer GNga. These additional scenarios include 1D and 2D tasks, as well as a new double pendulum task, which is classically challenging yet computationally efficient, in response to Reviewer sLTC. The effectiveness of our method is confirmed in these scenarios. The results are added to Table 1, Table 2, and Appendix F in our revised manuscript. For more details, see the responses to Reviewer GNga and sLTC.
- **Clarification on contributions and differences:** We further emphasized our contributions and clarified the differences from existing methods in detail, in response to Reviewer pd6k. We make modifications accordingly in the final paragraph of Section 1, and Section 2. For more details, see the responses to Reviewer pd6k.
- **Clarified problem setup:** In response to Reviewer pd6k, we refined our problem setup and replaced the term "environment" with "system dynamics" for better clarity. These improvements are reflected in Section 3.1. For more details, see the responses to Reviewer pd6k.
- **Description of the proposed method:** We present clarification of the description of the proposed method, as concerned by Reviewer pd6k. In particular, we justify that our method is clear to follow, given a solid understanding of the prerequisite materials on SDEs provided in Section 3.2 and Appendices A and B of the original submission, which are consistent with conventions in the diffusion model literature. Additionally, we emphasize that our method is not heuristic, based on our established theoretical analysis. For more details, see the responses to Reviewer pd6k.
- **Analysis of the model horizon $H$:** We add the analysis of the effect of the model horizon $H$, as raised by Reviewer sLTC. We perform additional evaluations for $H=6$ and $H=10$, and provided practical guidelines for selecting $H$. The results and analysis are shown in Appendix E.3. For more details, see the responses to Reviewer sLTC.

---

### Meta-Review · Area_Chair_cCBD · 2024-12-19

**Metareview:**

The paper proposes a novel method for closed loop control using a diffusion policy which preferentially denoises early steps of the control sequence over later ones. The proposed method demonstrates superior performance in terms of achieving control objectives, with computation time much shorter than other diffusion policy baseline controllers.

The insight of the proposed method is simple (avoid denoising later steps of a controller since they are unlikely to be executed in a closed-loop system), but insightful. The result is an approach which is likely to be widely applicable to many control problems, as the authors demonstrate in several low and high-dimensional control settings. The proposed method achieves better performance results than prior work, while also having faster computation times.

The biggest weakness of the paper is in the presentation. The reviewers were split on the quality of the presentation, but in my own reading, the method *is* hard to follow. One reviewer also declined to provide a review at all because they could not understand the paper's contents despite reading it several times. Several variables are not defined, and Figure 2 (demonstrating the method) is not sufficiently self-encapsulating as an explanation on its own. For those already very familiar with diffusion models (and specifically how they might be used in a control setting), the math might be easier to follow, but it seems unlikely that a reader less well-versed in diffusion modeling (or control) could figure out how the method works without spending a week reading the paper in depth. In either case, the method is likely not described clearly enough for someone to reimplement it.

However, given that there were very positive reviews on the novelty and success of the method, I lean towards acceptance (poster). The authors are encouraged to take the concerns about presentation quality seriously for the camera-ready version however.

**Additional Comments On Reviewer Discussion:**

During the rebuttal period, the authors added new results testing their method on an inverted double pendulum setup as requested by reviewer sLTC. Theoretical guarantees for the method were also provided to address the concerns of reviewer pd6k.

Additional experiments on different types of noise were added to address concerns from GNga, as well as additional experiments when physical constraints are present.

Reviewer pd6k brought up concerns about presentation, and these were not addressed in the revision.

---

### Decision · Program_Chairs · 2025-01-22

Accept (Poster)